# Nearly Space-Optimal Graph and Hypergraph Sparsification in Insertion-Only Data Streams

**Vincent Cohen-Addad**[*]    **David P. Woodruff**[†]    **Shenghao Xie**[‡]    **Samson Zhou**[§]

## Abstract

We study the problem of graph and hypergraph sparsification in insertion-only data streams. The input is a hypergraph $H = (V, E, w)$ with $n$ nodes, $m$ hyperedges, and rank $r$, and the goal is to compute a hypergraph $\widehat{H}$ that preserves the energy of each vector $x \in \mathbb{R}^n$ in $H$, up to a small multiplicative error. In this paper, we give a streaming algorithm that achieves a $(1+\varepsilon)$-approximation, using $\mathcal{O}\left(\frac{rn}{\varepsilon^2} \log^2 n \log r\right) \cdot \text{poly}\left(\log \log m\right)$ bits of space, matching the sample complexity of the best known offline algorithm up to $\text{poly}\left(\log \log m\right)$ factors. Our approach also provides a streaming algorithm for graph sparsification that achieves a $(1 + \varepsilon)$-approximation, using $\mathcal{O}\left(\frac{n}{\varepsilon^2} \log n\right) \cdot \text{poly}(\log \log n)$ bits of space, improving the current bound by $\log n$ factors. Furthermore, we give a space-efficient streaming algorithm for min-cut approximation. Along the way, we present an online algorithm for $(1 + \varepsilon)$-hypergraph sparsification, which is optimal up to polylogarithmic factors. Hence, we achieve $(1 + \varepsilon)$-hypergraph sparsification in the sliding window model, with space optimal up to poly-logarithmic factors. Lastly, we give an adversarially robust algorithm for hypergraph sparsification using $\frac{n}{\varepsilon^2} \cdot \text{poly}\left(r, \log n, \log r, \log \log m\right)$ bits of space.

## 1 Introduction

Graphs are fundamental structures that naturally model complex relationships in real-world data, from social networks and transportation systems to knowledge graphs and human brains. Because of their great expressive power, these relational models are fundamental to research in computer science, data science, and machine learning, in addition to many other fields. *Graph cuts* are used to partition graphs into distinct regions by minimizing a cost function, thereby providing insightful information. For example, in image segmentation, graph cuts help separate objects from the background by modeling pixels as nodes in a graph and optimizing the partitioning based on intensity or color differences (Boykov & Kolmogorov, 2004). In network clustering, graph cuts are used to detect communities within social networks by partitioning nodes into groups with strong internal connections while minimizing connections between different groups (Newman, 2006). In hierarchical clustering, sparse graph cuts are used to increasingly refine subgraphs to achieve better performance for Dasgupta's objective (Dasgupta, 2016; Braverman et al., 2025; Deng et al., 2025). More generally, spectral methods not only preserve key quantities such as cut sizes, but also accommodate more complex partitioning, e.g., multi-way cuts, and consider other concepts such as the graph Laplacian and its eigenvalues.

With the explosive growth of large-scale databases and the increasing demand for scalable machine learning and AI systems, graphs have become more complex and massive than ever before. In many modern applications, the sheer volume of graphs create significant computational bottlenecks, making it crucial to obtain smaller approximate graphs while preserving key features. *Cut sparsifiers* and *spectral sparsifiers* are smaller, efficient representations of large-scale graphs that approximately preserve cut sizes and spectral properties, respectively. These techniques are therefore often used to accelerate graph-related algorithms and reduce computation overhead. For instance, graph spar-

[*]Google Research. E-mail: `cohenaddad@google.com`.

[†]Carnegie Mellon University and Google Research. E-mail: `dwoodruf@cs.cmu.edu`.

[‡]Texas A&M University. E-mail: `xsh1302@gmail.com`.

[§]Texas A&M University. E-mail: `samsonzhou@gmail.com`.

sifiers have been used to enhance the performance of graph representation learning methods by reducing irrelevant information and preserving important structural features in the graph (Calandriello et al., 2018; Zhu et al., 2019; Zeng et al., 2020; Zheng et al., 2020; Braverman et al., 2021a; Chunduru et al., 2022; Zhang et al., 2024), spectral sparsifiers have been used to construct lightweight graph neural networks while maintaining their overall effectiveness (Li et al., 2023; Xie et al., 2024), and hypergraph sparsifiers have been used to produce efficient algorithms related to cuts and flows (Chekuri & Xu, 2017; Chandrasekaran et al., 2018; Chekuri & Xu, 2018; Veldt et al., 2023), which have been applied to hypergraph partitioning and clustering (Akbudak et al., 2013; Deveci et al., 2013; Ballard et al., 2016) and machine learning tasks (Li & Milenkovic, 2017; 2018; Veldt et al., 2020). In addition, graph sparsification is a key algorithmic component in theoretical computer science, e.g., computing the eigenvalues of the Laplacian matrix (Soma & Yoshida, 2019) and solving the Laplacian system, which produces fast symmetric diagonally dominant (SDD) system solvers (Koutis et al., 2014; Spielman & Teng, 2014; Soma & Yoshida, 2019).

**Data streams.** As datasets have grown dramatically in size in recent years, there is an increasing interest in large-scale computational models that avoid storing the entire dataset or making multiple passes over it. This has led to the development of the streaming model, which processes data sequentially under strict memory constraints, making it well-suited for handling massive datasets efficiently, e.g., network activity logs, IoT device streams, financial transactions, database event records, and real-time scientific observations. In this paper, we ask: *Do graph sparsification problems require additional space complexity in the streaming model, compared to the offline setting?*

## 1.1 OUR CONTRIBUTIONS

In this paper, we resolve the above question, providing algorithms for cut and spectral sparsification problems, losing only poly-iterated logarithmic factors in space, i.e., $\mathrm{polylog}(\log n)$ factors for a graph with $n$ vertices.

**Graph sparsification.** The graph sparsification problem has been extensively studied (Spielman & Srivastava, 2008; Batson et al., 2014; Lee & Sun, 2017; Jambulapati & Sidford, 2018), and there are well-known constructions for spectral sparsifiers with $\mathcal{O}\left(\frac{n}{\varepsilon^2}\right)$ edges (Batson et al., 2014). We provide efficient algorithms for graph spectral sparsification in insertion-only streams that have optimal space complexity compared to the best sample complexity of an offline algorithm, up to poly-iterated logarithmic factors in $n$. Here, we represent a graph as a three-tuple $G = (V, E, w)$, where $V$ is the set of vertices with size $n$, $E$ is the set of edges with size $m = \mathrm{poly}(n)$, and $w : E \to \mathbb{Z}^+$ is the weight assignment function, where the weights are positive integers upper bounded by $\mathrm{poly}(n)$, i.e., a fixed polynomial in $n$. We give our formal statement as follows.

**Theorem 1.1** (Streaming graph sparsification, informal version of Theorem E.3). *Given a graph $G = (V, E, w)$ with $n$ vertices defined by an insertion-only stream, there is an algorithm that gives a $(1 + \varepsilon)$-spectral sparsifier with probability $1 - \frac{1}{\mathrm{poly}(n)}$, storing $\frac{n}{\varepsilon^2} \mathrm{poly}(\log \log n)$ edges, i.e., $\frac{n}{\varepsilon^2} \log n \, \mathrm{poly}(\log \log n)$ bits, and using $\mathrm{poly}(n)$ update time.*

By comparison, offline constructions for graph spectral sparsifiers can be combined with the merge-and-reduce framework (see Section 3) for coreset constructions to achieve algorithms with $\mathcal{O}\left(\frac{n}{\varepsilon^2} \log^4 n\right)$ edges in the insertion-only setting. Additionally, Cohen et al. (2020) produced a spectral sparsifier with $\mathcal{O}\left(\frac{n}{\varepsilon^2} \log^2 n\right)$ edges in the online model, and similarly, Kapralov et al. (2020) achieved a spectral sparsifier with $\mathcal{O}\left(\frac{n}{\varepsilon^2} \log n\right)$ edges in the dynamic streaming model, where both insertions and deletions of edges are permitted; both of these results can also be applied to the insertion-only setting. Here, we note that storing each edge in the sparsifier uses $\mathcal{O}(\log n)$ bits of space. Our result avoids the additional multiplicative $\log n$ factors in previous results, achieving a nearly-optimal space. We visualize the above comparison in Figure 1.

**Graph min-cut approximation.** In addition, we construct a space-efficient streaming algorithm that approximates the graph min-cut in insertion-only data streams.

**Theorem 1.2** (Streaming min-cut approximation, informal version of Theorem E.18). *Given a graph $G = (V, E, w)$ with $n$ vertices defined by an insertion-only stream, there is an algorithm which outputs a $(1 + \varepsilon)$-approximation to the size of the min-cut of $G$ with probability $1 - \frac{1}{\mathrm{poly}(n)}$, storing $\frac{n}{\varepsilon} \cdot \mathrm{polylog}(n, \frac{1}{\varepsilon})$ edges, i.e., $\frac{n}{\varepsilon} \cdot \mathrm{polylog}(n, \frac{1}{\varepsilon})$ bits, and using $\mathrm{poly}(n)$ update time.*

| Setting | Reference | Space (number of bits) |
|---------|-----------|------------------------|
| Offline | Batson et al. (2014) | $\mathcal{O}\left(\frac{n}{\varepsilon^2}\log n\right)$ |
| Streaming | Kapralov et al. (2020) | $\mathcal{O}\left(\frac{n}{\varepsilon^2}\log^2 n\right)$ |
| Streaming | Theorem 1.1 | $\mathcal{O}\left(\frac{n}{\varepsilon^2}\log n\,\text{poly}(\log\log n)\right)$ |

Fig. 1: Comparison of our graph spectral sparsifier and the previous optimal results in different settings. Our sparsifier is tight up to $\text{poly}(\log\log n)$ factors.

We remark that our result improves on the logarithmic dependence in $n$ compared to the previous work (Ding et al., 2024).That is, our algorithm uses $\mathcal{O}\left(\frac{n}{\varepsilon}\cdot\log^c(\frac{n}{\varepsilon})\log\frac{1}{\varepsilon}\right)$ bits of space, while the algorithm of Ding et al. (2024) uses $\mathcal{O}\left(\frac{n}{\varepsilon}\cdot\log^{c+1}(\frac{n}{\varepsilon})\right)$ bits of space. We illustrate this comparison in Figure 2.

| Setting | Reference | Space (number of bits) |
|---------|-----------|------------------------|
| Offline | Ding et al. (2024) | $\mathcal{O}\left(\frac{n}{\varepsilon}\log^c n\right)$ |
| Streaming | Ding et al. (2024) | $\mathcal{O}\left(\frac{n}{\varepsilon}\log^{c+1} n\right)$ |
| Streaming | Theorem 1.2 | $\mathcal{O}\left(\frac{n}{\varepsilon}\log^c n\log\frac{1}{\varepsilon}\right)$ |

Fig. 2: Comparison of our graph min-cut approximation algorithm and the previous optimal results in different settings.

**Hypergraph sparsification.** Hypergraphs are a generalization of graphs where each hyperedge can connect more than 2 nodes, enabling the representation of multi-way relationships beyond pairwise connections. This ability to capture higher-order correlations makes hypergraphs particularly valuable in various applications, such as hypergraph neural networks (Feng et al., 2019; Gao et al., 2023; Feng et al., 2024) and hypergraph clustering (Takai et al., 2020). Soma & Yoshida (2019) formalized the definition of a $(1+\varepsilon)$-hypergraph spectral sparsifier and gave a construction with $\mathcal{O}\left(\frac{n^3}{\varepsilon^2}\log n\right)$ hyperedges. Bansal et al. (2019) achieved an upper bound of $\mathcal{O}\left(\frac{nr^3}{\varepsilon^2}\log n\right)$, where $r = \max_{e\in E}|e|$ is the rank of the hypergraph. Kapralov et al. (2021) gave an upper bound of $\mathcal{O}\left(\frac{n}{\varepsilon^4}\log^3 n\right)$. Jambulapati et al. (2023) and Lee (2023) simultaneously improved the upper bound to $\mathcal{O}\left(\frac{n}{\varepsilon^2}\log n\log r\right)$.

Note that the above results are in the offline setting and it is uncertain whether they can be adapted to our streaming setting. In contrast, existing algorithms use $\mathcal{O}\left(\frac{nr}{\varepsilon^2}\cdot\log^2 n\log r\right)$ hyperedges (Soma et al., 2024) or $\frac{n}{\varepsilon^2}\cdot\text{polylog}(m)$ hyperedges (Khanna et al., 2025a). However, they are not tight in $r$ and $\log m$ factors, which is *prohibitively large* for streaming large-scale hypergraphs, since $r$ could be $\Omega(n)$ and $m$ could be $\Omega(2^n)$ in the worst case. Therefore, the goal of our work is to improve both the $r$ and $\log m$ factors in the streaming setting. We give the first streaming algorithm that is optimal up to poly-iterated logarithmic factors in $m$, compared to the current best offline sample complexity.

**Theorem 1.3** (Streaming hypergraph sparsification, informal version of Theorem E.11)**.** *Given a hypergraph $H = (V, E, w)$ with $n$ vertices, $m$ hyperedges, and rank $r$ defined by an insertion-only stream, there is an algorithm that gives a $(1 + \varepsilon)$-spectral sparsifier with probability $1 - \frac{1}{\text{poly}(m)}$ storing $\frac{n}{\varepsilon^2}\log n\cdot\text{poly}(\log r, \log\log m)$ hyperedges, i.e., $\frac{rn}{\varepsilon^2}\log^2 n\cdot\text{poly}(\log r, \log\log m)$ bits, and using $\text{poly}(n)$ update time.*

Here, we note that storing each hyperedge requires $\mathcal{O}(r\log n)$ bits of space (see Remark D.7). Our algorithm only loses $\text{polylog}\,r$ and $\text{poly}(\log\log m)$ factors, which are at most $\text{polylog}\,n$ in the worst case, removing off the undesirable $\text{poly}(n)$ factors when $m = \Omega(2^n)$.

We note that the update times of our streaming algorithms are $\text{poly}(n)$, which could be potentially improved if we relax the space constraint. The main bottleneck is the calculation of the online sampling probability, which requires matrix-multiplication time $\tilde{\mathcal{O}}(n^\omega)$ (omitting $\varepsilon$ and polylogarithmic factors), where $\omega \approx 2.37$ is the exponent for matrix multiplication runtime (Alman et al., 2025). While applying the standard merge-and-reduce directly offers faster $\tilde{\mathcal{O}}(r)$ update time (where $r$

is the rank), it incurs a prohibitive $\mathrm{polylog}(m)$ space dependency. Our focus is optimizing space, which is often the primary bottleneck for processing massive datasets in streaming.

In addition, we give an online sampling scheme for hypergraph spectral sparsification with efficient space. The online setting is a more restrictive setting that must immediately and irrevocably decide whether each arriving edge should be sampled or discarded, i.e., online algorithms cannot retract decisions. Online algorithms are particularly useful in settings where intermediate results *must* be reported as the stream continues and thus are beneficial in practice because downstream computation can begin immediately when an item is sampled, as the item will not be removed at a later time.

**Theorem 1.4** (Online hypergraph sparsification, informal version of Theorem E.8). *Given a hypergraph $H = (V, E, w)$ with $n$ vertices, $m$ hyperedges, and rank $r$ defined by an insertion-only stream, there is an online algorithm that uses $n \log n \, \mathrm{poly}(\log \log m)$ bits of working memory and gives a $(1+\varepsilon)$-spectral sparsifier with probability $1 - \frac{1}{\mathrm{poly}(m)}$ with $\mathcal{O}\left(\frac{n}{\varepsilon^2} \log n \log m \log r\right)$ hyperedges.*

For comparison, Soma et al. (2024) provided an online algorithm that outputs a $(1+\varepsilon)$-hypergraph sparsifier using $\mathcal{O}\left(\frac{nr}{\varepsilon^2} \cdot \log^2 n \log r\right)$ hyperedges. However, they require $\mathcal{O}\left(n^2\right)$ working memory to store the sketch that defines their sampling probability. Our algorithm improves this bound to $n \log n \, \mathrm{poly}(\log \log m)$ bits, significantly reducing memory consumption. This result resolves Question 6.2 of Soma et al. (2024), which asks for improvements in space complexity. We summarize our results in Figure 3.

| Setting | Reference | Space (number of bits) |
|---|---|---|
| Offline | Jambulapati et al. (2023); Lee (2023) | $\frac{rn}{\varepsilon^2} \log^2 n \log r$ |
| Online | Khanna et al. (2025a) | $\frac{rn}{\varepsilon^2} \mathrm{polylog}(m)$ |
| Online | Theorem 1.4 | $\frac{rn}{\varepsilon^2} \log^2 n \log m \log r$ |
| Streaming | Theorem 1.3 | $\frac{rn}{\varepsilon^2} \mathrm{poly}(\log n, \log \log m)$ |

Fig. 3: Comparison of our hypergraph sparsifiers and the optimal results in different settings. Our streaming algorithm is tight in both $r$ and $\mathrm{polylog}(m)$ factors.

**Adversarial robustness.** We also obtain hypergraph sparsifiers in the adversarially robust streaming model. This model is framed as a two-player game between an adaptive adversary, who generates the input stream, and a randomized algorithm, which processes the inputs and outputs an estimate. Unlike the standard streaming setting, future inputs may depend on previous interactions between the adversary and the algorithm. Specifically, the game proceeds in $m$ rounds: in each round the adversary selects an input based on past inputs and outputs and sends to the algorithm, then the algorithm responds with a new estimate. This formulation captures the adaptive nature of real-world data streams and parallels adversarial robustness research in machine learning, where inputs are strategically chosen to exploit vulnerabilities in learning systems, such as adversarial examples in neural networks (Goodfellow et al., 2015), optimization-based evasion attacks (Carlini & Wagner, 2017; Athalye et al., 2018), adversarial training (Madry et al., 2018), and trade-offs between accuracy and robustness (Tsipras et al., 2019; Chen et al., 2022). Importantly, adaptivity need not arise from malicious attacks: it can also result from interactive queries to a large database or from iterative queries designed to optimize an underlying objective (Hassidim et al., 2020), settings where standard streaming algorithms fail. Given these perspectives, the adversarially robust streaming model has attracted significant attention (Ben-Eliezer et al., 2020; Ben-Eliezer & Yogev, 2020; Hassidim et al., 2020; Alon et al., 2021; Braverman et al., 2021a; Kaplan et al., 2021; Woodruff & Zhou, 2021; Beimel et al., 2022; Ben-Eliezer et al., 2022a;b; Chakrabarti et al., 2022; Avdiukhin et al., 2019; Assadi et al., 2023; Attias et al., 2023; Cherapanamjeri et al., 2023; Dinur et al., 2023; Woodruff et al., 2023a; Gribelyuk et al., 2024; Woodruff & Zhou, 2024; Gribelyuk et al., 2025; Lin et al., 2026).

We first extend our streaming algorithm results to a high-probability regime, obtaining success probability $1 - \delta$ with a space overhead of at most $\mathcal{O}\left(\log \log \frac{1}{\delta}\right)$, and then use this result to achieve an adversarially robust algorithm for robust hypergraph sparsification.

**Theorem 1.5** (Robust hypergraph sparsification, informal version of Theorem E.15). *Given a graph $H = (V, E, w)$ with $n$ vertices, $m$ edges, and rank $r$ defined by an insertion-only stream, there exists an adversarially robust algorithm that constructs a $(1+\varepsilon)$-spectral sparsifier with probability*

$1 - \frac{\delta}{\text{poly}(m)}$ *storing* $\frac{n}{\varepsilon^2} \text{poly}(\log n, \log r, \log \log \frac{m}{\delta})$ *hyperedges and* $\frac{n}{\varepsilon^2} \cdot \text{poly}(r, \log n, \log \log \frac{m}{\delta})$ *edges in the associated graph, i.e.,* $\frac{n}{\varepsilon^2} \cdot \text{poly}(r, \log n, \log r, \log \log \frac{m}{\delta})$ *bits in total.*

**The sliding window model.** An additional application of our online algorithms is to the sliding window model. Since the streaming model remains oblivious to the times at which specific data points arrive, it is unable to prioritize recent data over older data. By comparison, the more general sliding window model considers the most recent $W$ updates $\{x_{n-W+1}, \dots, x_n\}$ to be the active data, which is crucial in situations where newer information is more relevant or accurate, such as tracking trends in financial markets or analyzing recent Census data. Indeed, the sliding window model outperforms the streaming model in various applications (Babcock et al., 2002; Datar et al., 2002; Papapetrou et al., 2015; Wei et al., 2016), especially for time-sensitive settings such as data summarization (Chen et al., 2016; Epasto et al., 2017), social media data (Osborne et al., 2014), and network monitoring (Cormode & Muthukrishnan, 2005; Cormode & Garofalakis, 2008; Cormode, 2013). The sliding window model is particularly useful when computations *must* focus only on the most recent data. This is important in cases where data retention is limited by regulations. For example, Facebook stores user search histories for up to six months (Facebook, 2021), Google stores browser data for up to nine months (Google, 2021), and OpenAI retains API inputs and outputs for up to 30 days (OpenAI). The sliding window model captures the ability to expire data, in alignment with these time-based retention policies, and consequently has received significant attention in various problems (Lee & Ting, 2006; Braverman & Ostrovsky, 2007; Braverman et al., 2012; 2018; 2021b; Woodruff & Zhou, 2021; Ajtai et al., 2022; Jayaram et al., 2022; Blocki et al., 2023; Cohen-Addad et al., 2025). Applying our online algorithm, we utilize a framework for the sliding window model to achieve an algorithm for hypergraph sparsification:

**Theorem 1.6** (Streaming hypergraph sparsification in the sliding window model, informal version of Theorem F.2)**.** *Given a hypergraph* $H = (V, E, w)$ *with* $n$ *vertices,* $m$ *hyperedges, and rank* $r$ *defined by the sliding window model, there is an algorithm that gives a* $(1 + \varepsilon)$-*spectral sparsifier with probability* $1 - \frac{1}{\text{poly}(n)}$, *storing* $\frac{n}{\varepsilon^2} \text{polylog}(m, r)$ *hyperedges, i.e.,* $\frac{rn}{\varepsilon^2} \log n \, \text{polylog}(m, r)$ *bits.*

We remark that the above space upper bound is worse than that of our streaming algorithm. This is because the sliding window model is more general than the standard streaming model, as the algorithm needs to maintain an accurate approximation of the most recent $W$ items. In the standard streaming model, a newly-arrived edge with low overall contribution can be discarded. By comparison, in the sliding window model, the same edge may be critical to the most recent $W$ items and must therefore be retained, which leads to a higher space complexity. Moreover, one can set the window size $W$ as the stream length, in which case the entire stream is in the active window. Thus, the sliding window model can be parameterized to subsume the streaming model.

**Concurrent and independent work.** We remark that Goranci & Momeni (2025); Khanna et al. (2025a) recently also achieved online and dynamic algorithms for hypergraph sparsification that samples $\frac{n}{\varepsilon^2} \cdot \text{polylog}(m)$ hyperedges. Our algorithm has a suboptimal update time, but we achieve better space complexity, which is *prioritized* in the streaming setting. In addition, Khanna et al. (2025b) recently showed that in an insertion-only stream, a hypergraph cut sparsifier can be computed using $\frac{nr}{\varepsilon^2} \cdot \text{polylog}(n)$ bits of space, avoiding the additional multiplicative $\log m$ factors. By comparison, our streaming algorithm for spectral sparsification, which generalizes a cut sparsifier, also avoids the $\log m$ dependency. We refer the reader to Appendix A for a more detailed discussion.

## 1.2 Preliminaries

In this section, we introduce several important definitions and techniques related to graphs and hypergraphs. We refer the reader to Appendix B for a comprehensive discussion. For an integer $n > 0$, we use the notation $[n] = \{1, \dots, n\}$. We use $\text{poly}(n)$ to denote a fixed polynomial of $n$, whose degree is determined by setting parameters in the algorithm accordingly. We use $\text{polylog}(n)$ to denote $\text{poly}(\log n)$. We use $\tilde{\mathcal{O}}(F) = F \, \text{polylog} \, F$ to hide the polylogarithmic factors.

**Hypergraphs.** For a hypergraph $H = (V, E, w)$, its energy function $Q_H : \mathbb{R}^n \to \mathbb{R}$ is $Q_H(x) := \sum_{e \in E} Q_e(x) = \sum_{e \in E} w(e) \cdot \max_{(u,v) \in e}(x_u - x_v)^2$, where $Q_e(x)$ is the energy of a hyperedge $e$. A weighted hypergraph $\widehat{H}$ is a $(1 + \varepsilon)$-multiplicative spectral sparsifier for $H$ if it preserves the energy of all vectors $x$ up to an $\varepsilon$-multiplicative error: $|Q_{\widehat{H}}(x) - Q_H(x)| \leq \varepsilon \cdot Q_H(x), \forall x \in \mathbb{R}^n$. The definition of the associated graph relates a hypergraph to a graph with the same set of

vertices. Given a hypergraph $H$, its associated graph $G = (V, E)$ is by replacing each hyperedge $e = (u_1, \ldots, u_r) \in H$ with the $\binom{r}{2}$ edges $(u_i, u_j)$, where $1 \le i < j \le r$, each with weight $w(e)$.

**Graph Laplacian.** We review the graph Laplacian that encodes the structure of a graph into a matrix. The graph Laplacian is an $n \times n$ matrix where its diagonal stores the weighted degree for each node, and the other entries store the weight of each edge $(i, j)$ in $G$. The graph Laplacian provides a way to view the graph spectral sparsifier problem as a matrix spectral approximation problem so that one can implement numerical linear algebra methods. The graph Laplacian $L_G$ can be written as the Gram matrix $\mathbf{A}^\top \mathbf{A}$, where the incidence matrix $\mathbf{A}$ consists of binary row vectors representing each weighted row $uv$ in graph $G$. That is, $\mathbf{a}_i := \mathbf{a}_{uv} = \sqrt{w(e)} \cdot (\chi_u - \chi_v)$, where $\chi_i$ denotes the elementary row vector with a single nonzero entry in the $i$-th coordinate. In the graph spectral sparsification problem, the energy is defined to be $x^\top L_G x$, which is $x^\top \mathbf{A}^\top \mathbf{A} x$. This relates the graph sparsification problem to the matrix spectral approximation problem as follows.

Spielman & Srivastava (2008) introduced the notion of the effective resistance $r_e = w(e) \cdot (\chi_u - \chi_v) L_G^{-1} (\chi_u - \chi_v)^\top$ of an edge $e$ in the graph, encoding the importance of $e$. They construct a sparsifier with $\mathcal{O}\left(\frac{n}{\varepsilon^2}\right)$ edges by sampling each edge with probability proportional to $r_e$. The effective resistance $r_e$ turns out to be the leverage score of the row $\mathbf{a}_i$ in the incidence matrix $\mathbf{A}$, defined by $\tau_i = \mathbf{a}_i (\mathbf{A}^\top \mathbf{A})^{-1} \mathbf{a}_i^\top$, where $\mathbf{a}_i$ is the row representing $e$. Thus, sampling edges in the graph is equivalent to sampling rows from the incidence matrix $\mathbf{A}$.

**Online leverage score.** Another core technical tool is online row sampling (Cohen et al., 2020). In the online setting, we only have access to the prefix matrix $\mathbf{A}_i$ that arrives before the current row $\mathbf{a}_i$. Therefore, we consider the online variation of leverage scores: $\tau_i^{\mathrm{OL}}(\mathbf{A}) := \mathbf{a}_i (\mathbf{A}_i^\top \mathbf{A}_i)^{-1} \mathbf{a}_i^\top$. Then, sampling with probability proportional to $\tau_i^{\mathrm{OL}}(\mathbf{A})$ gives us an online spectral approximation.

## 2    ONLINE HYPERGRAPH SPECTRAL SPARSIFIER

We start with our online hypergraph sparsifier, which is a fundamental subroutine in our streaming hypergraph sparsifier. We first give a brief overview of previous methods in the offline setting in Section 2.1, where the entire hypergraph is given in advance. Then in Section 2.2, we explain the main technical challenges in extending them to the online setting and how we solve them. Our approach involves using a novel local weight assignment subroutine, whose discussion is deferred to Section 2.3. Next, we briefly summarize our technical novelties.

**Technical overview.** Let $\mathbf{A}$ be the incidence matrix of the hypergraph's associated graph. For each hyperedge $e$, prior offline algorithms assigned a weight $z_{uv}$ to each edge $(u, v)$ in the clique defined by $e$. Let $\mathbf{Z}$ denote the diagonal weight matrix, where its $(i, i)$-th entry is the weight $z_i$ of the edge represented by the $i$-th row in matrix $\mathbf{A}$. The previous works then defined the sampling probability of $e$ based on the leverage scores of $\mathbf{Z}^{1/2}\mathbf{A}$ and chose $\mathbf{Z}$ wisely to reduce the sample complexity. However, this approach cannot be immediately applied since we do not have the entire hypergraph in the online setting. We introduce a novel local weight assignment scheme to decide the weights $\mathbf{Z}_t$ at each time $t$ (see Section 2.3) and combine it with online row sampling to maintain a sketch matrix for $\mathbf{Z}_t^{1/2}\mathbf{A}_t$, to define a valid set of sampling probabilities with efficient memory (see Section 2.2). In particular, when the local weights $\mathbf{Z}_t$ are defined, we fix these weights in the following updates. When the next hyperedge $e_{t+1}$ arrives, we use $\mathbf{Z}_t^{1/2}\mathbf{A}_t$ to compute the weights for each $(u, v) \in e_{t+1}$, and then pass the re-weighted edges to an online row sampling subroutine and add the sampled edges to $\mathbf{Z}_t^{1/2}\mathbf{A}_t$ to define $\mathbf{Z}_{t+1}^{1/2}\mathbf{A}_{t+1}$. Importantly, we decouple the randomness of these procedures to ensure that the algorithm outputs a valid sparsifier.

### 2.1    OFFLINE ALGORITHMS

In this section, we summarize the previous techniques in the offline setting. We begin with the method in (Kapralov et al., 2021) which achieves a small sample complexity via the weight assignment scheme. For a hyperedge $e$ with weight $w(e)$ in the hypergraph, we assign a weight $z_{uv}$ to each edge $(u, v) \in e$ in the associated graph so that the weights satisfy $\sum_{(u,v) \in e} z_{uv} = w(e)$, that is, the edges in the clique defined by $e$ sum up to $w(e)$. Here, we abuse the notation $(u, v) \in e$ to denote that $(u, v)$ is in the clique defined by $e$. We define the effective resistance of an edge $i$ in

the re-weighted associated graph to be $\frac{\tau_i(\mathbf{Z}^{1/2}\mathbf{A})}{z_i}$, where $\tau_i(\mathbf{Z}^{1/2}\mathbf{A})$ is the leverage score of the $i$-th row in the weighted incidence matrix $\mathbf{Z}^{1/2}\mathbf{A}$. Now, the weight matrix $\mathbf{Z}$ is chosen to satisfy that for all hyperedges $e$ and edges $i$ in the clique defined by $e$, the ratios $\frac{\tau_i(\mathbf{Z}^{1/2}\mathbf{A})}{z_i}$ are within a constant fraction of each other. The intuition is to "balance" the effective resistance of each edge in the clique to reduce the sample complexity (see the discussion in Appendix C.2). Let $p_e$ be proportional to the balanced effective resistance of hyperedge $e$, then sampling each $e$ with probability $p_e$ and rescaling the sampled hyperedges by $\frac{1}{p_e}$ gives a valid sparsifier.

**Sufficient conditions for sampling hyperedges.** Jambulapati et al. (2023) stated a more general definition of sampling probability called the "group leverage score overestimate", which still gives a valid spectral sparsifier. Indeed, for any weight assignment with $\sum_{(u,v)\in e} z_{uv} = w(e)$, it suffices to sample each hyperedge $e$ with probability higher than the maximum of all ratios $\frac{\tau_i(\mathbf{Z}^{1/2}\mathbf{A})}{z_i}$ of edges $i$ in the clique of $e$, as stated in the following definition. Then, we are left to choose a proper $\mathbf{Z}$ that gives the desired sample complexity.

**Definition 2.1** (Sampling probability for each hyperedge). *Given a hypergraph $H = (V, E, w)$ and its associated graph $G = (V, F, z)$, where the weight assignment $z$ satisfies $\sum_{(u,v)\in e} z_{uv} = w(e)$ for all $e \in E$, let $\mathbf{A}$ be the incidence matrix for $G$, and let $\mathbf{Z}$ be the weight matrix. Let $\rho = \mathcal{O}\left(\frac{1}{\varepsilon^2}\log m \log r\right)$ be the amplification factor. We set the sampling probability of $e$ to satisfy $p_e \geq \rho \cdot w(e) \cdot \max_{(u,v)\in e} \mathbf{a}_{uv}(\mathbf{A}^\top \mathbf{Z}\mathbf{A})^{-1}\mathbf{a}_{uv}^\top$. Here, $\mathbf{a}_{uv}(\mathbf{A}^\top \mathbf{Z}\mathbf{A})^{-1}\mathbf{a}_{uv}^\top = \frac{\tau_{uv}(\mathbf{Z}^{1/2}\mathbf{A})}{z_{uv}}$.*

Jambulapati et al. (2023) applied a refined chaining argument by leveraging Talagrand's growth function framework to achieve a tighter bound. Their analysis can be applied as a black-box if the sampling probabilities satisfy the sufficient conditions stated in Definition 2.1: (1) They are an overestimate; (2) Their values are defined independently of whether other hyperedges are sampled.

## 2.2 OUR ONLINE ALGORITHMS

The above methods do not generalize to the online setting because at time $t$, we can only define the weights $\mathbf{Z}_t$ based on the sub-hypergraph that arrived before $t$ and its corresponding incidence matrix $\mathbf{A}_t$, as the entire hypergraph is not given. However, the sampling probabilities must satisfy Definition 2.1 for the entire hypergraph and its corresponding incidence matrix $\mathbf{A}$. Thus, the challenge here is to compute the correct weights under restricted information. We summarize our novel techniques that solve this problem as follows and defer the full analysis to Appendix C.1.

**Online sampling probabilities.** Notice that the re-weighted matrix $\mathbf{Z}_t^{1/2}\mathbf{A}_t$ is consistent along the way, and hence $\frac{\tau_i(\mathbf{Z}_t^{1/2}\mathbf{A}_t)}{z_i}$ is an overestimate of $\frac{\tau_i(\mathbf{Z}^{1/2}\mathbf{A})}{z_i}$, where $\mathbf{Z}^{1/2}\mathbf{A}$ is the re-weighted matrix of the entire hypergraph, due to the monotonicity of online leverage scores. Thus, it suffices to define the sampling probability $p_{e_t}$ as $\rho \cdot w(e_t) \cdot \max_{(u,v)\in e_t} \mathbf{a}_{uv}(\mathbf{A}_t^\top \mathbf{Z}_t \mathbf{A}_t)^{-1}\mathbf{a}_{uv}^\top$, which gives an overestimate of the offline sampling probability. Next, we state the guaranties of our local weight assignment scheme, which assigns the weights to a newly-arrived hyperedge $e_{t+1}$ based on the prefix matrix $\mathbf{Z}_t^{1/2}\mathbf{A}_t$, and we defer its discussion to Section 2.3. The first point shows that the weights of $e_{t+1}$ sum up to $w(e_{t+1})$, ensuring that Definition 2.1 is satisfied. The second point upper bounds the sampling probabilities, which is used to bound the number of sampled hyperedges.

**Theorem 2.2** (Local weight assignment). *Given a graph $G = (V, F)$ with weight $\mathbf{Z}_t$ and incidence matrix $\mathbf{A}_t$ and a newly-arrived hyperedge $e_{t+1} \subset V$ with weight $w(e_{t+1})$, then there is a procedure* GETWEIGHTASSIGNMENT$(\mathbf{Z}_t^{1/2}\mathbf{A}_t, e_{t+1})$ *that assigns a weight $z_{uv}$ to each edge $(u,v) \in e_{t+1}$ such that it satisfies (1) $\sum_{(u,v)\in e_{t+1}} z_{uv} = w(e)$, and (2) $\max_{(u,v)\in e} \mathbf{a}_{uv}(\mathbf{A}_t^\top \mathbf{Z}_t \mathbf{A}_t)^{-1}\mathbf{a}_{uv}^\top = \mathcal{O}\left(\sum_{(u,v)\in e} \tau_{uv}(\mathbf{Z}_t^{1/2}\mathbf{A}_t)\right)$, where $\tau$ is the leverage score function.*

**Optimizing the working memory.** If we sample each hyperedge based on the sampling probabilities defined by the weight assignments in Theorem 2.2, we would get a valid sparsifier with a desirable sample complexity. However, we cannot store the entire matrix $\mathbf{Z}^{1/2}\mathbf{A}$, i.e., $\mathbf{Z}_t^{1/2}\mathbf{A}_t$ at each $t$, which uses a prohibitively large working memory and exceeds the sample complexity. To solve this problem, we combine the local weight-assignment subroutine with an online row sampling

subroutine (Cohen et al., 2020) to reduce working memory. Let $\mathbf{B}$ be a matrix, and let $\mathbf{B}_i$ denote the matrix formed by the first $i$ rows of $\mathbf{B}$. In online row sampling, we sample the newly arrived row $\mathbf{b}_i$ with probability $\bar{p}_i \propto \mathbf{b}_i (\widetilde{\mathbf{B}_{i-1}}^\top \widetilde{\mathbf{B}_{i-1}})^{-1} \mathbf{b}_i^\top$, where the prefix matrix $\widetilde{\mathbf{B}_{i-1}}$ consists of the rows sampled from previous steps, and we add $\mathbf{b}_i / \bar{p}_i$ to $\widetilde{\mathbf{B}_{i-1}}$ if it is sampled. We adapt this idea to our online sampling procedure in an iterative way. When a new hyperedge $e_{t+1}$ arrives: (1) We compute its weights $z_{t+1,uv}$ using our local weight-assignment subroutine implementing on the current sketch $\mathbf{M}$, which is a 2-approximate spectral approximation of the prefix weighted matrix $\mathbf{Z}_t^{1/2} \mathbf{A}_t$; (2) For each $(u,v) \in e_{t+1}$, we feed the weighted row $\sqrt{z_{t+1,uv}} \, \mathbf{a}_{uv}$ into the online row-sampling procedure to update $\mathbf{M}$. Thus, the resulting matrix is still a 2-approximate spectral approximation to the matrix $\mathbf{Z}_{t+1}^{1/2} \mathbf{A}_{t+1}$, which suffices for our purpose. Our algorithm is given in Algorithm 1.

---

**Algorithm 1** Online hyperedge spectral sparsifier

---

1: **Require:** Stream of $m$ hyperedges for hypergraph $H$ with rank $r$
2: **Ensure:** Spectral sparsifier $\widehat{H}$ for $H$
3: $\widehat{H} \leftarrow \emptyset$, $\rho \leftarrow \mathcal{O}\left(\frac{1}{\varepsilon^2} \log m \log r\right)$, $\mathbf{M} \leftarrow \emptyset$
4: **for** hyperedge $e_t$ **do**
5: $\quad z_t \leftarrow z_{t-1} \cup \text{GETWEIGHTASSIGNMENT}(\mathbf{M}, e_t)$
6: $\quad$ **for** $(u,v) \in e_t$ **do**
7: $\quad\quad$ Sample weighted row $\mathbf{a}_{uv} \cdot \sqrt{z_{t,uv}}$ to $\mathbf{M}$ by online row sampling (Cohen et al., 2020)
8: $\quad\quad$ {$\mathbf{M}$ is a 2-spectral approximation to $\mathbf{Z}_t^{1/2} \mathbf{A}_t$ at all times $t$, i.e., $\frac{1}{2} \cdot \mathbf{M}^\top \mathbf{M} \preceq \mathbf{A}_t^\top \mathbf{Z}_t \mathbf{A}_t \preceq 2 \cdot \mathbf{M}^\top \mathbf{M}$}
9: $\quad$ **end for**
10: $\quad$ **for** $(u,v) \in e_t$ **do**
11: $\quad\quad q_{uv} \leftarrow \mathbf{a}_{uv} \cdot w(e_t) \cdot (\mathbf{M}^\top \mathbf{M})^{-1} \cdot \mathbf{a}_{uv}^\top$
12: $\quad$ **end for**
13: $\quad p_{e_t} \leftarrow \min\{1, 2\rho \cdot \max_{(u,v) \in e_t} q_{uv}\}$
14: $\quad$ With probability $p_{e_t}$, $\widehat{H} \leftarrow \widehat{H} \cup \frac{1}{p_{e_t}} \cdot e_t$
15: $\quad$ **Return** $\widehat{H}$
16: **end for**

---

**Decoupling technique.** The second sufficient condition stated below Definition 2.1 requires the sampling probabilities to not depend on which hyperedges are sampled before. We show that the online sampling subroutine does not violate this requirement because the sketch $\mathbf{M}$ (built by online row sampling) and the sampling of hyperedges use independent randomness.

**Lemma 2.3** (Decoupling). *For each time $t$, the sampling probability $p_{e_t}$ defined by Algorithm 1 is independent of the hyperedges that have been sampled previously.*

Specifically, consider a fixed stream $e_1, \cdots, e_m$, and let $\mathbf{M}_t$ denote the matrix $\mathbf{M}$ at time $t$. When the inner randomness of online row sampling is fixed, the sequence of matrices $\mathbf{M}_1, \cdots, \mathbf{M}_m$ is also fixed. Then, the sampling probabilities $p_{e_t}$ are defined independently, each based on the value of $\max_{(u,v) \in e} \mathbf{a}_{uv} \cdot w(e_t) \cdot (\mathbf{M}^\top \mathbf{M})^{-1} \cdot \mathbf{a}_{uv}^\top$. Thus, the sampling of each $e_t$ is independent of whether the previous hyperdges are sampled. Applying the chaining argument in (Jambulapati et al., 2023) and a careful analysis on the sample complexity, we show Theorem 1.4.

## 2.3 OUR LOCAL WEIGHT ASSIGNMENT SCHEME

In this section, we show how to construct the local weight assignment in Theorem 2.2. The full analysis is deferred to Appendix C.2. Given a graph $G = (V, F)$ with weight $\mathbf{Z}$ and incidence matrix $\mathbf{A}$ and a newly-arrived hyperedge $e \subset V$ with weight $w(e)$, our goal is to assign weight $z_{uv}$ to each $(u,v) \in e$ such that $\gamma \cdot \min_{(u,v) \in e: z_{uv} > 0} q_{uv} \geq \max_{(u',v') \in e} q_{u'v'}$ for some constant $\gamma$, where $q_{uv}$ is defined as the ratio $\frac{\tau_{uv}(\mathbf{Z}^{1/2}\mathbf{A})}{z_{uv}}$. That is, the ratios of the clique defined by $e$ are within a $\gamma$-fraction of each other. We remark that this property gives the upper bound on the sampling probability stated in the second point of Theorem 2.2 (see the discussion at the beginning of Appendix C.2).

Kapralov et al. (2021) provided a greedy algorithm that shifts the weights from the edges with a higher ratio to the edges with a lower ratio, as long as they are not $\gamma$-balanced. We propose a

localized version of their algorithm, i.e., we only conduct the weight shift operation on the newly-arrived edge $e$ without changing the previous assigned weights in $G$, to ensure the consistency of our weight matrix. We show that local weight shift operations always increase the spanning tree potential of $G$ by a certain amount, and since the potential is upper bounded, this process terminates in finite steps and results in a $\gamma$-balanced weight assignment. The algorithm is given in Algorithm 2.

---

**Algorithm 2** Online $(\gamma, e)$-balanced weight assignment

---

1: **Require:** Given weighted graph $G = (V, F, w)$, hyperedge $e \subset V$ with weight $w(e)$
2: **Ensure:** Online $(\gamma, e)$-balanced weight assignment
3: **Initialize:** for all $(u, v) \in e$, set $z_{uv} = w(e)/|e|$
4: $G \leftarrow (V, F \cup \bigcup_{(u,v) \in e} (u, v), z)$
5: **While** G is not online $(\gamma, e)$-balanced weight assignment **do**
6:     Select $(u, v) \neq u', v' \in e$ such that $q_{uv} > \gamma \cdot q_{u'v'}$ and $z_{u'v'} > 0$ {$q_{uv}$ is the ratio in Definition 2.1}
7:     $\lambda \leftarrow \min\{z_{uv}, \frac{\gamma - 1}{2\gamma \cdot q_{uv}}\}$
8:     $z_{uv} \leftarrow z_{uv} + \lambda$
9:     $z_{u'v'} \leftarrow z_{u'v'} - \lambda$
10: **Return** $G$

---

## 3 STREAMING MODEL

In this section, we present our space-optimal streaming algorithms for both graph and hypergraph. We first introduce the classic *merge-and-reduce* method. Then we explain how we achieve space-optimal algorithms in Section 3.1. Next, we briefly summarize our technical novelties.

**Technical overview.** Cohen-Addad et al. (2023) gave a streaming framework that achieved efficient space by filtering the stream using an online sampling subroutine and then passing the sampled edges to the merge-and-reduce method. The main challenge of applying this framework is that we cannot afford to store the edges sampled by the online subroutine, which are used to define the next online sampling probability in online row sampling (Cohen et al., 2020). We revisit the analysis of Cohen et al. (2020) and show that the output of the merge-and-reduce method suffices to accurately estimate the online sampling probabilities, ensuring the validity of the framework. In addition, we show that by carefully controlling the number of times we change the output of our algorithm, we achieve an adversarially robust algorithm.

**Merge-and-reduce framework.** Let $S(\cdot) \to \mathbb{R}_{>0}$ denote a mapping from input parameters to the sample complexity of an online algorithm. The online graph sparsifier samples $S(n, \log m, \varepsilon, \delta)$ edges for an input stream of length $m$ on a graph with $n$ nodes, accuracy $\varepsilon \in (0, 1)$, and failure probability $\delta$, with high probability. We call a re-weighted subgraph $\widehat{G} \subset G$ an $(1 + \varepsilon)$-*coreset* if it is a $(1 + \varepsilon)$-sparsifier of $G$. The merge-and-reduce approach partitions the stream into blocks of size $S\left(n, \log m, \frac{\varepsilon}{2\log(mn)}, \frac{\delta}{\text{poly}(mn)}\right)$ and builds a $\left(1 + \frac{\varepsilon}{2\log(mn)}\right)$-coreset for each block, so that each coreset can be interpreted as the leaves of a binary tree with height at most $\log(mn)$. For each node in the binary tree, a coreset of size $S\left(n, \log m, \frac{\varepsilon}{2\log(mn)}, \frac{\delta}{\text{poly}(mn)}\right)$ is built from the coresets representing the two children of the node. Assuming that the coreset construction admits a merging procedure, i.e., by taking the graph consisting of the union of the weighted edges in each of the coresets, then the root of the tree represents a coreset for the entire stream with distortion at most $\left(1 + \frac{\varepsilon}{2\log(mn)}\right)^{\log(mn)} \leq (1 + \varepsilon)$ and failure probability $\delta$. A figure illustrating the merge-and-reduce approach is shown in Figure 4. Due to our choice of accuracy $\varepsilon' = \frac{\varepsilon}{2\log(mn)}$ in each node, directly applying the merge-and-reduce approach to the input stream loses $\log m$ factors. Cohen-Addad et al. (2023) improved on these factors by adding an online sampling procedure ahead of the merge-and-reduce approach. Suppose that the online sampling procedure is nearly optimal, then the input stream of the merge-and-reduce approach is significantly shorter. For instance, in the graph sparsification problem, there is an online algorithm that samples $\mathcal{O}\left(\frac{n}{\varepsilon^2}\log n\right)$ edges, so the coresets only have size $S\left(n, \log \log n, \frac{\varepsilon}{2\log \log(n)}, \frac{\delta}{\text{polylog}(n)}\right)$. This turns $\text{polylog}(n)$ factors into $\text{polylog}\log(n)$ factors, which is more space-efficient for huge graphs. Since we directly pass on

the online coreset of $\mathcal{O}\left(\frac{n}{\varepsilon^2}\log n\right)$ edges to the merge-and-reduce framework and never store them entirely, we do not lose the $\log n$ from online sampling and achieve efficient space.

## 3.1 Our Streaming Algorithms

Unfortunately, the above streaming framework does not apply immediately, since in online row sampling (Cohen et al., 2020), we use the online coreset from the previous step to compute the sampling probability of the next edge. However, we cannot afford to store the online coreset in our streaming algorithms, and it remains to obtain an efficient sketch matrix to define those probabilities. Instead, we use the coreset obtained in the last step of our streaming algorithm to compute the next online sampling probability. The algorithm is given in Algorithm 3.

---

**Algorithm 3** Streaming framework inspired by (Cohen-Addad et al., 2023)

---

1: **Require:** Stream $\mathcal{S}$, online sampling procedure for $\mathcal{S}$, merge-and-reduce procedure
2: **Ensure:** Coreset on $\mathcal{S}$
3: **for** each edge $e_t$ arrived in the stream $\mathcal{S}$ **do**
4:     Sample $e_t$ based on the online sampling probability defined by $\widehat{\mathcal{S}}$ $\{\widehat{\mathcal{S}}$ is the output sparsifier$\}$
5:     If $s_t$ is sampled by online sampling, add the corresponding update to $\mathcal{S}'$
6:     Run merge-and-reduce on $\mathcal{S}'$ and return $\widehat{\mathcal{S}}$ on the root node
7: **end for**

---

We apply a refined analysis based on Cohen et al. (2020) to show that our approach to define the online sampling probability still gives a valid online coreset, stated as follows. Using this result, we obtain the streaming algorithm in Theorem 1.1. The full analysis is deferred to Appendix E.1.

**Theorem 3.1** (Online Coreset). *Let graph $G = (V, E, w)$ with $n$ vertices and $m$ edges be defined by an insertion-only stream, let $\mathbf{A}$ be the incidence matrix of $G$, and let $\mathbf{a}_i$ be the $i$-th row in $\mathbf{A}$. We define an online sampling scheme as follows. Let $\widetilde{\ell}_i$ be the ridge-leverage score for online row sampling: $\widetilde{\ell}_i = \mathcal{O}(1) \cdot \mathbf{a}_i^T \left(\widehat{\mathbf{A}}_{i-1}^T \widehat{\mathbf{A}}_{i-1} + \lambda\mathbf{I}\right)^{-1} \mathbf{a}_i$, where $\lambda = \frac{\varepsilon}{\mathrm{poly}(n)} \leq \frac{\varepsilon}{\sigma_{\min}^2(\mathbf{A})}$ and $\widehat{\mathbf{A}}_{i-1}$ is the output of our streaming algorithm the last round (see Algorithm 3). We set the sampling probability of row $\mathbf{a}_i$ as $p_i = \min\{c\widetilde{\ell}_i, 1\}$, where $c = \mathcal{O}\left(\frac{\log m}{\varepsilon^2}\right)$, and if $\mathbf{a}_i$ is sampled, we store the re-weighted row $\frac{\mathbf{a}_i}{\sqrt{p_i}}$. Then, let $\widetilde{\mathbf{A}}_i$ be the matrix outputted by our sampling scheme for each $i \in [m]$, with probability at least $\frac{1}{\mathrm{poly}(m)}$ we have $(1 - \varepsilon)\mathbf{A}_i^\top \mathbf{A}_i \preceq \widetilde{\mathbf{A}}_i^T \widetilde{\mathbf{A}}_i \preceq (1 + \varepsilon)\mathbf{A}_i^\top \mathbf{A}_i$. The algorithm samples $\mathcal{O}\left(\frac{n}{\varepsilon^2}\log m \log n\right)$ edges and uses $\mathrm{poly}(n)$ update time.*

For our hypergraph sparsifier, recall that our online algorithm keeps a sketch $\mathbf{M}$ of the weighted matrix $\mathbf{Z}_t^{1/2}\mathbf{A}_t$, which is obtained by online row sampling. To optimize the working memory of this procedure such that it satisfies the streaming memory constraint, we further pass $\mathbf{M}$ to the merge-and-reduce method as in Algorithm 3 and use the resulted sketch matrix in the following subroutines. We show that this modification does not violate the decoupling argument in Section 2.2. The full analysis is deferred to Appendix E.2 and Appendix E.3.

**Adversarial robustness.** We observe that the streaming framework provides a hypergraph sparsification algorithm with probability $1 - \frac{\delta}{\mathrm{poly}(m)}$ with only an additional $\log\log\frac{m}{\delta}$ factor in space (see Appendix E.4), which further gives a space-efficient adversarially robust algorithm. We generalize the computational path framework in Ben-Eliezer et al. (2020) that transforms a high-probability streaming algorithm into an adversarially robust algorithm as long as it has a small $\varepsilon$-flip number. Suppose that an algorithm estimates a target function $f$ on input stream $\{x_t\}_{t=1}^T$, the $\varepsilon$-flip number is defined as the number of times $f(x_1, \ldots, x_t)$ increases by an $\varepsilon$-fraction. In the stream, the algorithm fixes its output when $f$ is within an $\varepsilon$-fraction, and so we can union bound across all possible inputs given by the adversary if the failure probability $\delta$ is sufficiently small. In our hypergraph sparsification algorithm, we use the Laplacian matrix of the input hypergraph's associated graph to control when we switch the output, i.e., we switch only if its eigenvalues increase by an $\varepsilon$-fraction. Since its eigenvalues can increase by an $\varepsilon$-fraction at most $\frac{n\log m}{\varepsilon}$ times, the $\varepsilon$-flip number is $\frac{n\log m}{\varepsilon}$. Then our high-probability result gives an adversarially robust algorithm with efficient space complexity; we defer the complete discussion to Appendix E.5.

ACKNOWLEDGMENTS

David P. Woodruff is supported in part by Office of Naval Research award number N000142112647 and a Simons Investigator Award. Shenghao Xie is supported in part by NSF CCF-2335411. Samson Zhou is supported in part by NSF CCF-2335411. Samson Zhou gratefully acknowledges funding provided by the Oak Ridge Associated Universities (ORAU) Ralph E. Powe Junior Faculty Enhancement Award.

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

## A  CONCURRENT AND INDEPENDENT WORK

We summarize several relevant results from concurrent and independent work.

**Online algorithms.**  We remark that Khanna et al. (2025a) recently also achieved an online algorithm for hypergraph sparsification that samples $\frac{n}{\varepsilon^2} \cdot \text{polylog}(m)$ hyperedges. Their approach is based on maintaining a logarithmic number of spanners and adding arriving hyperedges to a spanner based on connectivity within the spanner and a geometrically decreasing probability. In fact, due to efficient subroutines for these algorithms, their update time is near-linear. Our algorithm has a suboptimal update time, but we achieve better sample complexity and efficient working memory. By comparison, our approach is based on computing a balanced solution to an optimization problem, and it is not clear whether there exists an efficient algorithm to achieve a fast update time. We also achieve an algorithm for online spectral sparsification with fast update time while losing $\text{poly}(r)$ factors, which suffices for the remainder of our applications.

**Dynamic algorithms.**  The framework in (Khanna et al., 2025a) also solves hypergraph sparsification in the fully-dynamic setting, where both insertions and deletions are allowed and time efficiency is prioritized. Their algorithm maintains a sparsifier with $\frac{n}{\varepsilon^2} \cdot \text{polylog}(m)$ hyperedges while requiring sublinear update time in $m$. Similarly, Goranci & Momeni (2025) provided dynamic algorithms for hypergraph sparsification using $nr^3 \, \text{poly}(\log n, \frac{1}{\varepsilon})$ space and $r^4 \, \text{poly}(\log n, \frac{1}{\varepsilon})$ update time. These results naturally generalize the insertion-only streaming setting. Although they achieve better time efficiency, their space bounds lose polynomial factors in $r$ and $\log m$. As we mentioned earlier, their sub-optimal space usage hinders the applications in the streaming setting, where the space efficiency is prioritized. On the contrary, our algorithm reduces the $\text{polylog}(m)$ factors to $\text{poly}(\log \log m)$ factors while maintaining a $\text{poly}(n)$ update time, making significant improvements for dense hypergraphs (e.g., $m = \Omega(2^n)$).

**Hypergraph cut sparsifier.**  In addition, we note that Khanna et al. (2025b) recently showed that in an insertion-only stream, a hypergraph cut sparsifier can be computed using $\frac{rn}{\varepsilon^2} \cdot \text{polylog}(n)$ bits of space, establishing an $\Omega(\log m)$ separation from hypergraph cut sparsification in dynamic streams. Their algorithm estimates the strength of components in the hypergraph, and whenever a component gets sufficiently large strength, it contracts the component to a single vertex to save space. This idea extends their result to a more general bounded-deletion setting: if the stream has at most $k$

hyperedge deletions, then $\frac{rn}{\varepsilon^2} \log k \operatorname{polylog}(n)$ bits of space suffice for hypergraph cut sparsification. By comparison, our approach is based on a streaming framework that implements merge-and-reduce coreset construction to the output of an online algorithm. We achieve an algorithm that solves hypergraph spectral sparsification in insertion-only streams using $\frac{rn}{\varepsilon^2} \log^2 n \log r \operatorname{poly}(\log \log(m))$ bits of space, which generalizes the hypergraph cut sparsification result. Assuming there are no multi-hyperedges in the hypergraph, i.e., $m = \mathcal{O}(2^n)$, our algorithm uses $\frac{rn}{\varepsilon^2} \log r \operatorname{polylog}(n)$ bits of space, and hence it avoids extraneous dependence on $\operatorname{polylog}(m)$ factors, i.e., we have the same dependencies as Khanna et al. (2025b) in the $\log m$ factors. In addition, in terms of the dependence on $\log n$, our space bound has a factor of $\tilde{\mathcal{O}}(\log^2 n)$, as opposed to $\operatorname{polylog}(n)$ in Khanna et al. (2025b), and thus we achieve a better space complexity when $m$ is relatively small.

## B  PRELIMINARIES

In this section, we introduce several important definitions and techniques related to graphs and hypergraphs. For an integer $n > 0$, we use the notation $[n]$ to denote the set $\{1, \ldots, n\}$. We use $\operatorname{poly}(n)$ to denote a fixed polynomial of $n$, whose degree can be determined by setting parameters in the algorithm accordingly. We use $\operatorname{polylog}(n)$ to denote $\operatorname{poly}(\log n)$. We use $\tilde{\mathcal{O}}(F) = F \operatorname{polylog} F$ to hide the polylogarithmic factors.

**Hypergraphs.**  We state the formal definition of the hypergraph energy function and the hypergraph spectral sparsifier.

**Definition B.1** (Energy). *For a hypergraph $H = (V, E, w)$, we define $Q_H : \mathbb{R}^n \to \mathbb{R}$ to be the following analog of the Laplacian quadratic form $Q_H(x) = \sum_{e \in E} w(e) \cdot \max_{(u,v) \in e} (x_u - x_v)^2$. We define the energy $Q_e(x)$ of a hyperedge $e$ to be $Q_e(x) = w(e) \cdot \max_{(u,v) \in e} (x_u - x_v)^2$, such that $Q_H(x) = \sum_{e \in E} Q_e(x)$.*

The above definition carries a physical interpretation as the potential energy of a family of rubber bands. One can consider a hyperedge $e$ as a rubber band stretched to encircle several locations $\{x_v, v \in e\}$, so $\max_{(u,v) \in e} (x_u - x_v)^2$ is proportional to its potential energy Lee (2023). Soma & Yoshida (2019) introduced the following notion of the hypergraph spectral sparsifier, generalizing the definition for graphs (Spielman & Srivastava, 2008). The hypergraph spectral sparsifier is defined as a re-weighted subset $\widehat{H}$ of the original hypergraph $H$, which preserves the energy of $x$ on $H$ up to a $(1 + \varepsilon)$-approximation for all vectors $x \in \mathbb{R}^n$.

**Definition B.2** (Hypergraph spectral sparsifier). *A weighted hypergraph $\widehat{H}$ is a $(1+\varepsilon)$-multiplicative spectral sparsifier for $H$ if*

$$|Q_{\widehat{H}}(x) - Q_H(x)| \le \varepsilon \cdot Q_H(x), \quad \forall x \in \mathbb{R}^n \tag{1}$$

A natural way to relate a hypergraph to a multi-graph is to replace a hyperedge with $r$ nodes by a clique of $\binom{r}{2}$ edges formed by the nodes in that hyperedge, which is stated in the following definition of associated graph.

**Definition B.3** (Associated graph). *Given a hypergraph $H$, we define a multi-graph $G = (V, E)$ to be the associated graph of $H$ by replacing each hyperedge $e = (u_1, \ldots, u_r) \in H$ with the $\binom{r}{2}$ edges $(u_i, u_j)$, where $1 \le i < j \le r$, each with weight $w(e)$.*

**Graph Laplacian.**  We review the graph Laplacian that encodes the structure of a graph into a matrix. The graph Laplacian is an $n \times n$ matrix where its diagonal stores the weighted degree for each node, and the other entries store the weight of each edge $(i, j)$ in $G$. The graph Laplacian provides a way to view the graph spectral sparsifier problem as a matrix spectral approximation problem so that one can implement numerical linear algebra methods to solve it.

**Definition B.4** (Graph Laplacian). *Given a weighted graph $G = (V, E, w)$ with $n$ vertices, the graph Laplacian matrix $L_G \in \mathbb{R}^{n \times n}$ is defined as*

$$L_G = \sum_{e=(u,v) \in E} w(e) \cdot (\chi_u - \chi_v)^\top (\chi_u - \chi_v),$$

*where $\chi_i$ denotes the elementary row vector with a single nonzero entry in the $i$-th coordinate.*

The graph Laplacian can be written as the Gram matrix $\mathbf{A}^\top \mathbf{A}$, where the incidence matrix $\mathbf{A}$ consists of binary vectors representing each weighted row in graph $G$.

**Fact B.5.** *The graph Laplacian $L_G$ has the following properties:*

*(1) Let $L_{uv}$ be the graph Laplacian for an edge $uv$, so that*

$$L_{uv} = w(e) \cdot (\chi_u - \chi_v)^\top (\chi_u - \chi_v).$$

*Then $L_G = \sum_{e=(u,v)\in E} L_{uv}$.*

*(2) For each $i,j \in [n]$, the $(i,j)$-th entry of $L_G$ is the weighted degree $\deg(i) = \sum_{(i,j)\in E: u\in V} w(ij)$ for $i = j$ and $-w(ij)$ for $i \neq j$, where $w(ij)$ is the weight of edge $(i,j)$.*

*(3) Let $\mathbf{A} \in \mathbb{R}^{|E|\times n}$ be the incidence matrix of $G$, where each row $\mathbf{a}_i$ of $\mathbf{A}$ corresponds to an edge $e = (u,v) \in E$, so that $\mathbf{a}_{uv} := \mathbf{a}_i = \sqrt{w(e)} \cdot (\chi_u - \chi_v)$. Then $L_G = \mathbf{A}^\top \mathbf{A}$.*

We remark that the matrix $A$ sometimes denotes the adjacency matrix, an $n \times n$ square matrix with $A_{uv} = w(e)$ if $(u,v) \in E$, in some other definitions of the graph Laplacian, e.g., $L = D - A$. Here, we stick to the notion of the incidence matrix $\mathbf{A}$ in (Jambulapati et al., 2023), which is *different* from the definition of the adjacency matrix. In addition, we sometimes abuse the subscript in this paper, e.g., $\mathbf{a}_i = \mathbf{a}_{uv}$ if $e = (u,v)$ is represented by the $i$-th row in the incidence matrix $\mathbf{A}$.

In the graph spectral sparsification problem, the energy is defined to be $x^\top L_G x$, which is $x^\top \mathbf{A}^\top \mathbf{A} x$. Thus, sampling the edges of the graph is equivalent to sampling rows from the matrix $\mathbf{A}$. In the hypergraph spectral sparsification problem, the definition of energy is slightly different. However, we can still relate it to the graph Laplacian of its associated graph as follows.

**Fact B.6.** *Let $H = (V, E, w)$ be a hypergraph, let $G = (V, F, w)$ be its associated graph, and let $L_{uv}$ be the graph Laplacian for edge $(u,v)$ in graph $G$. The energy $Q_e(x)$ of a hyperedge $e$ satisfies*

$$Q_e(x) = \max_{(u,v)\in e} x^\top L_{uv} x$$

Spielman & Srivastava (2008) introduced the following definition of the effective resistance of an edge in the graph, and they sample the edge with probability proportional to its effective resistance, which results in sample complexity $\mathcal{O}\left(\frac{n}{\varepsilon^2}\log n\right)$. Due to the equivalence mentioned above between spectral sparsification and matrix spectral approximation, the effective resistance of an edge $e$ turns out to be the leverage score of the row $\mathbf{a}_i$ in the incidence matrix $\mathbf{A}$, where $\mathbf{a}_i$ is the row representing $e$.

**Lemma B.7** (Effective resistance and leverage score). *For a graph $G = (V, E, w)$, the effective resistance of an edge $e = (u,v) \in E$ is the quantity $r_e = w(e) \cdot (\chi_u - \chi_v) L_G^{-1}(\chi_u - \chi_v)^\top$, where $\chi_i$ denotes the elementary row vector with a single non-zero entry in the $i$-th coordinate. Given a matrix $\mathbf{A} \in \mathbb{R}^{n\times d}$, the leverage score of row $\mathbf{a}_i$ is the quantity $\tau_i = \mathbf{a}_i(\mathbf{A}^\top \mathbf{A})^{-1}\mathbf{a}_i^\top$. Let $\mathbf{A}$ be the incidence matrix of $G$. Let $\mathbf{a}_i$ be the row that represents the edge $e$. Then, we have $r_e = \tau_i$.*

**Online leverage scores.** We introduce a core technical tool given by (Cohen et al., 2020), which gives the matrix spectral approximation in the online setting, i.e., the rows arrive sequentially in a stream. When a row $\mathbf{a}_i$ arrives, we only have access to matrix $\mathbf{A}_i$, which is a part of the matrix $\mathbf{A}$ that arrives before $\mathbf{a}_i$. Therefore, we consider the following online variation of leverage scores to approximate the real leverage scores.

**Definition B.8** (Online leverage scores). *Given a matrix $\mathbf{A} \in \mathbb{R}^{n\times d}$, let $\mathbf{A}_i$ denote the first $i$ rows of $\mathbf{A}$, i.e., $\mathbf{A}_i = \mathbf{a}_1 \circ \ldots \circ \mathbf{a}_i$, where $\circ$ denotes the row-wise concatenation operator. The online leverage score of row $\mathbf{a}_i$ is the quantity $\mathbf{a}_i(\mathbf{A}_i^\top \mathbf{A}_i)^{-1}\mathbf{a}_i^\top$.*

The next statement shows that the online leverage score is in fact an overestimate of the leverage score of $\mathbf{a}_i$ in the whole matrix $\mathbf{A}$.

**Lemma B.9** (Monotonicity of leverage scores). *(Braverman et al., 2020; Cohen et al., 2020; Woodruff & Yasuda, 2023) Given a matrix $\mathbf{A} \in \mathbb{R}^{m\times n}$, let $\tau(\mathbf{a}_i)$ denote the leverage score of $\mathbf{a}_i$ and let $\tau^{\mathsf{OL}}(\mathbf{a}_i)$ denote the online leverage score of row $\mathbf{a}_i$. Then $\tau^{\mathsf{OL}}(\mathbf{a}_i) \geq \tau(\mathbf{a}_i)$.*

The next statement bounds the sum of online leverage scores, which bounds the sample complexity.

**Theorem B.10** (Sum of online leverage scores). *(Cohen et al., 2020) Given a matrix $\mathbf{A} \in \mathbb{R}^{m \times n}$, let $\tau^{\mathsf{OL}}(\mathbf{a}_i)$ denote the online leverage score of row $\mathbf{a}_i$. Then $\sum_{i=1}^{m} \tau^{\mathsf{OL}}(\mathbf{a}_i) = \mathcal{O}(n \log \kappa)$, where $\kappa = \|\mathbf{A}\|_2 \cdot \max_{i \in [n]} \|\mathbf{A}_i^{-1}\|_2$ is the online condition number of $\mathbf{A}$.*

With the above properties of monotonicity and bounded sum, one can generalize the following result for the online matrix spectral approximation.

**Theorem B.11** (Online row sampling). *(Cohen et al., 2020) Given a matrix $\mathbf{A} \in \mathbb{R}^{m \times n}$ defined by an insertion-only stream, let $\tau^{\mathsf{OL}}(\mathbf{a}_i)$ denote the online leverage score of row $\mathbf{a}_i$. Then sampling $\tilde{\mathcal{O}}(n \log n \log \kappa)$ rows with probability proportional to $\tau^{\mathsf{OL}}(\mathbf{a}_i)$ gives a 2-spectral approximation to $\mathbf{A}$ at all times.*

## C  ONLINE HYPERGRAPH SPECTRAL SPARSIFIER

In this section, we present the complete analysis for our online sparsifier.

### C.1  MISSING PROOFS IN SECTION 2.2

We provide the missing proofs in Section 2.2. For simplicity, we paste our algorithm in Algorithm 4.

---

**Algorithm 4** Online hyperedge spectral sparsifier (restatement of Algorithm 1)

---

1: **Require:** Stream of $m$ hyperedges for hypergraph $H$ with rank $r$
2: **Ensure:** Spectral sparsifier $\widehat{H}$ for $H$
3: $\widehat{H} \leftarrow \emptyset, \rho \leftarrow \mathcal{O}\left(\frac{1}{\varepsilon^2} \log m \log r\right), \mathbf{M} \leftarrow \emptyset$
4: **for** hyperedge $e_t$ **do**
5:     $z_t \leftarrow z_{t-1} \cup \text{GETWEIGHTASSIGNMENT}(\mathbf{M}, e_t)$ {See Theorem 2.2}
6:     **for** $(u, v) \in e_t$ **do**
7:         Sample weighted row $\mathbf{a}_{uv} \cdot \sqrt{z_{t,uv}}$ to $\mathbf{M}$ by online row sampling (Cohen et al., 2020)
8:         {$\mathbf{M}$ is a 2-spectral approximation to $\mathbf{Z}_t^{1/2} \mathbf{A}_t$ at all times $t$, i.e., $\frac{1}{2} \cdot \mathbf{M}^\top \mathbf{M} \preceq \mathbf{A}_t^\top \mathbf{Z}_t \mathbf{A}_t \preceq 2 \cdot \mathbf{M}^\top \mathbf{M}$}
9:     **end for**
10:     **for** $(u, v) \in e_t$ **do**
11:         $q_{uv} \leftarrow \mathbf{a}_{uv} \cdot w(e_t) \cdot (\mathbf{M}^\top \mathbf{M})^{-1} \cdot \mathbf{a}_{uv}^\top$
12:     **end for**
13:     $p_{e_t} \leftarrow \min\{1, 2\rho \cdot \max_{(u,v) \in e_t} q_{uv}\}$
14:     With probability $p_{e_t}$, $\widehat{H} \leftarrow \widehat{H} \cup \frac{1}{p_{e_t}} \cdot e_t$
15:     **Return** $\widehat{H}$
16: **end for**

---

We next show the correctness and bound the sample complexity of our algorithm. To begin with, we state the result of the chaining argument from (Jambulapati et al., 2023).

**Theorem C.1** (Correctness, see Theorem 10 in (Jambulapati et al., 2023)). *Given a hypergraph $H = (V, E, w)$ and its associated graph $G = (V, F)$ with valid weight assignment $z$, let $p_e$ be chosen independently according to Definition 2.1. Let $\rho = \mathcal{O}\left(\frac{1}{\varepsilon^2} \log m \log r\right)$. Suppose that we sample each hyperedge $e$ in $H$ with probability $\widehat{p}_e = \min\{1, p_e \cdot \rho\}$ and scale by $\frac{1}{\widehat{p}_e}$ if $e$ is sampled. Then, the resulting hypergraph $\mathcal{H}$ is a $(1+\varepsilon)$-spectral sparsifier for $H$ with probability $1 - \frac{1}{\text{poly}(m)}$.*

Next, we assume that there is a way to define the weight matrix $\mathbf{Z}_t$ locally such that the sampling probability $p_{e_t}$ defined by $p_{e_t} \geq w(e) \cdot \max_{(u,v) \in e} \mathbf{a}_{uv} (\mathbf{A}_t^\top \mathbf{Z}_t \mathbf{A}_t)^{-1} \mathbf{a}_{uv}^\top$ satisfies Definition 2.1 and is smaller than the sum of online leverage scores of $\mathbf{Z}^{1/2}\mathbf{A}$. We state the result as follows and defer the discussion to Appendix C.2.

**Theorem C.2.** *[Restatement of Theorem 2.2] Given a graph $G = (V, F)$ with weight $\mathbf{Z}_t$ and incidence matrix $\mathbf{A}_t$ and a newly-arrived hyperedge $e_{t+1} \subset V$ with weight $w(e_{t+1})$, then there is a procedure* GETWEIGHTASSIGNMENT$(\mathbf{Z}_t^{1/2}\mathbf{A}_t, e_{t+1})$ *that assigns a weight $z_{uv}$ to each edge $(u, v) \in e_{t+1}$ such that it satisfies (1) $\sum_{(u,v)\in e_{t+1}} z_{uv} = w(e)$, and (2) $\max_{(u,v)\in e} \mathbf{a}_{uv}(\mathbf{A}_t^\top \mathbf{Z}_t \mathbf{A}_t)^{-1}\mathbf{a}_{uv}^\top = \mathcal{O}\left(\sum_{(u,v)\in e} \tau_{uv}(\mathbf{Z}_t^{1/2}\mathbf{A}_t)\right)$, where $\tau$ is the leverage score function.*

Then, by Theorem C.1 and Theorem 2.2, sampling each hyperedge $e$ with probability defined by $p_{e_t} \geq w(e) \cdot \max_{(u,v)\in e} \mathbf{a}_{uv}(\mathbf{A}_t^\top \mathbf{Z}_t \mathbf{A}_t)^{-1}\mathbf{a}_{uv}^\top$ gives a valid sparsifier with desired sample complexity. Recall that this approach implies a suboptimal working memory. We apply the online row sampling scheme to sample from $\mathbf{Z}^{1/2}\mathbf{A}$, reducing the rows we store.

Notice that the chaining argument in (Jambulapati et al., 2023) requires that the sampling probabilities be assigned independently. The following statement shows that $p_{e_t}$ is defined independently of whether the previous hyperedges are sampled.

**Lemma C.3.** *[Restatement of Lemma 2.3] For each time $t$, the sampling probability $p_{e_t}$ is independent of the hyperedges that have been sampled previously.*

*Proof.* First, we stress that the construction of $\mathbf{M}$ by online row sampling from $\mathbf{Z}^{1/2}\mathbf{A}$ and sampling the hyperedges are separate procedures with independent inner randomness. Consider a fixed stream of hyperedges $e_1, \cdots, e_m$. Let $\mathbf{M}_t$ denote the matrix $\mathbf{M}$ at time $t$. When the inner randomness of online row sampling procedure is fixed, the sequence of matrices $\mathbf{M}_1, \cdots, \mathbf{M}_m$ is also fixed. Then, the sampling probabilities $p_{e_t}$ are defined independently, each based on the value of $\max_{(u,v)\in e} \mathbf{a}_{uv} \cdot w(e_t) \cdot (\mathbf{M}^\top \mathbf{M})^{-1} \cdot \mathbf{a}_{uv}^\top$. Thus, it is independent of the hyperedges that have been sampled previously. $\qquad\square$

Next, we show the correctness of Algorithm 4.

**Lemma C.4.** *Algorithm 4 outputs a $(1 + \varepsilon)$-hypergraph spectral sparsifier with probability $1 - \frac{1}{\text{poly}(m)}$.*

*Proof.* By the guarantee of Theorem B.11, $\mathbf{M}$ is a 2-spectral approximation to $\mathbf{Z}_t^{1/2}\mathbf{A}_t$ at all times $t$. Then, the estimated quadratic form $q_{uv}$ satisfies $q_{uv} \geq 2\mathbf{a}_{uv}(\mathbf{A}^\top \mathbf{Z}\mathbf{A})^{-1}\mathbf{a}_{uv}^\top$. In addition, by Theorem 2.2, our weight assignment satisfies $\sum_{(u,v)\in e} z_{uv} = w(e)$ for each $e$, so the sampling probabilities $p_{e_t}$ satisfy Definition 2.1. Moreover, by Lemma C.3, the sampling probabilities are defined independently of previously sampled hyperedges. Thus, we can apply Theorem C.1 directly to show that $\widehat{H}$ is a $(1 + \varepsilon)$-hypergraph spectral sparsifier with probability $1 - \frac{1}{\text{poly}(m)}$. $\qquad\square$

Next, we upper bound the sample complexity and the working memory, where the latter is the number of rows that we store in $\mathbf{M}$ to compute the sampling probabilities of the hyperedges.

**Lemma C.5.** *With probability $1 - \frac{1}{\text{poly}(m)}$, Algorithm 4 samples $\mathcal{O}\left(\frac{n}{\varepsilon^2}\log n \log m \log r\right)$ hyperedges. Moreover, it uses $\mathcal{O}\left(n \log^2 n \log m\right)$ bits of working memory.*

*Proof.* To bound the sample complexity, we only need to bound $\sum_{t=1}^m p_{e_t}$, where $p_{e_t} \leq \rho \cdot \max_{(u,v)\in e_t} q_{uv}$ due to our definition. Notice that the local weight assignment $\mathbf{Z}_{t+1}$ is defined by the sparsified graph with weighted incident matrix $\mathbf{M}$ at time $t$, then by Theorem 2.2, our weight assignment guarantees that $\max_{(u,v)\in e_t} q_{uv} = \mathcal{O}\left(\sum_{(u,v)\in e} \tau_{uv}(\mathbf{M})\right)$. Since $\mathbf{M}$ is a 2-spectral approximation to $\mathbf{Z}_t^{1/2}\mathbf{A}_t$ at all times $t$, we have

$$\sum_{t=1}^m \max_{(u,v)\in e_t} q_{uv} = \mathcal{O}\left(\sum_{t=1}^m \sum_{(u,v)\in e} \tau_{uv}(\mathbf{M}_t)\right)$$

$$= \mathcal{O}\left(\sum_{i=1}^{|F|} \tau_i^{\mathsf{OL}}(\mathbf{Z}^{1/2}\mathbf{A})\right),$$

where $\mathbf{M}_t$ is the matrix $\mathbf{M}$ defined at time $t$, $|F|$ is the number of rows in the incidence matrix $\mathbf{A}$, and $\tau^{\mathsf{OL}}$ denotes the online leverage score operator. Utilizing Theorem B.10, we bound the sum of online leverage scores on the RHS of the above equation by $\mathcal{O}\left(n \log \kappa\right)$. Therefore, we have

$$\sum_{t=1}^{m} \max_{(u,v) \in e_t} q_{uv} = \mathcal{O}\left(n \log \kappa\right),$$

where $\kappa = \|\mathbf{A}\|_2 \cdot \max_{i \in [n]} \|\mathbf{A}_i^{-1}\|_2$ is the online condition number of $\mathbf{A}$. We state an upper bound on the online condition number.

**Fact C.6** (see Corollary 2.4 in (Cohen et al., 2020)). *Suppose that all hyperedge weights are integers from* $[\mathrm{poly}(n)]$*, then we have* $\log \kappa = \mathcal{O}\left(\log n\right)$*.*

Suppose that the condition in Fact C.6 is satisfied, we have

$$\sum_{t=1}^{m} \max_{(u,v) \in e_t} q_{uv} = \mathcal{O}\left(n \log n\right).$$

Then, due to our choice of overestimate parameter $\rho$, we have

$$\sum_{t=1}^{m} p_{e_t} = \sum_{t=1}^{m} \rho \cdot \max_{(u,v) \in e_t} q_{uv} = \mathcal{O}\left(\frac{n}{\varepsilon^2} \log n \log m \log r\right).$$

Thus, we sample $\mathcal{O}\left(\frac{n}{\varepsilon^2} \log n \log m \log r\right)$ hyperedges in expectation. Note that the inner randomness of sampling each hyperedge with probability $p_e$ is independent, then by standard concentration inequalities, the number of sampled hyperedges is $\mathcal{O}\left(\frac{n}{\varepsilon^2} \log n \log m \log r\right)$ with probability $1 - \frac{1}{\mathrm{poly}(n)}$.

In addition, by the guarantee of online row sampling in Theorem B.11, it suffices to sample $\mathcal{O}\left(n \log n \log m\right)$ rows in $\mathbf{Z}^{1/2}\mathbf{A}$ to construct the 2-approximation $\mathbf{M}$. Since we only need to store $\mathbf{M}$ to compute the sampling probabilities, Algorithm 4 uses $\mathcal{O}\left(n \log^2 n \log m\right)$ bits of working memory. □

Combining Lemma C.4 and Lemma C.5, we have the following result for hypergraph spectral sparsification.

**Theorem C.7** (Online hypergraph spectral sparsifier). *Given a hypergraph $H = (V, E, w)$ with $n$ vertices, $m$ hyperedges, and rank $r$, there exists an online algorithm with $\mathcal{O}\left(n \log n \log^2 m\right)$ bits of working memory that constructs a $(1 + \varepsilon)$-spectral sparsifier with probability $1 - \frac{1}{\mathrm{poly}(m)}$ by sampling $\mathcal{O}\left(\frac{n}{\varepsilon^2} \log n \log m \log r\right)$ hyperedges.*

## C.2 EXISTENCE OF ONLINE $(\gamma, e)$ BALANCED WEIGHT ASSIGNMENT

In this section, we show how to construct the local weight assignment in Theorem C.2. In the offline setting, the "balancing weight" method mentioned previously is used to upper bound the number of samples. Suppose that for a hyperedge $e$, all edges $(u, v) \in e$ have the same ratio $\frac{\tau_i(\mathbf{Z}^{1/2}\mathbf{A})}{z_i}$. Then,

$$p_e = w(e) \cdot \mathbf{a}_{uv} (\mathbf{A}^\top \mathbf{Z} \mathbf{A})^{-1} \mathbf{a}_{uv}^\top = w(e) \cdot \frac{\tau_{uv}(\mathbf{Z}^{1/2}\mathbf{A})}{z_{uv}}.$$

Here, $(u, v)$ can be any edge in $e$. Hence, we have $z_{uv} \cdot p_e = w(e) \cdot \tau_{uv}$. Recall that our weight assignment satisfies $\sum_{(u,v) \in e} z_{uv} = w(e)$, then summing up the previous equation for all $(u, v) \in e$ gives $p_e = \sum_{(u,v) \in e} \tau_{uv}(\mathbf{Z}^{1/2}\mathbf{A})$, which is exactly the sum of the leverage scores as we desired. Since exact balanced weights are hard to find, we turn to the definition of $\gamma$-balanced weight assignment, where the ratios $\frac{\tau_i(\mathbf{Z}^{1/2}\mathbf{A})}{z_i}$ are within a $\gamma$ fraction of each other. This only loses a constant factor to the sample complexity.

We extend this definition to the online setting. Now, we have a weight matrix $\mathbf{Z}$, an incidence matrix $\mathbf{A}$, and an incoming hyperedge $e$. Here, we omit the subscript $t$ in the previous sections for simplicity. We assign weights to each edge in $e$ such that they satisfy the definition of $\gamma$-balanced weights.

**Definition C.8** (Online $(\gamma, e)$-balanced weight assignment). *Given a weighted graph $G = (V, F, z)$ and a hyperedge $e \subset V$ with weight $w(e)$, we assign a weight $z_{uv}$ to each edge $(u, v) \in e$ such that $\sum_{(u,v) \in e} z_{uv} = w(e)$ and add $(u, v)$ to $G$. We call it an online $(\gamma, e)$-balanced weight assignment if it satisfies*

$$\gamma \cdot \min_{(u,v) \in e: z_{uv} > 0} \frac{\tau_{uv}(\mathbf{Z}^{1/2}\mathbf{A})}{z_{uv}} \geq \max_{u', v' \in e} \frac{\tau_{u'v'}(\mathbf{Z}^{1/2}\mathbf{A})}{z_{u'v'}}.$$

Kapralov et al. (2021) provided a greedy algorithm that shifts the weights from the edges with a higher ratio to the edges with a lower ratio. They prove that such weight shift operations always increase the spanning tree potential (ST-potential) of the graph by a certain amount, which is defined as follows.

$$\Psi(G) = \log \left[ \sum_{T \in \mathbb{T}(G)} \prod_{(u,v) \in T} z_{uv} \right].$$

Here, $\mathbb{T}(G)$ is the set of all spanning trees of $G$. Since the ST-potential is upper bounded, this process terminates in finite steps and results in a valid set of weights. We demonstrate that the greedy algorithm can be applied in the online setting by showing that weight shift operations still increase the ST-potential. Now, we formalize the definition of the weight shift operation.

**Definition C.9** (Weight shift). *Given a weighted graph $G = (V, F, z)$, an edge $(u, v) \in F$, and a weight shift factor $\lambda \in \mathbb{R}$, the graph $G + \lambda \cdot uv$ is the weighted graph $G' = (V, F, z')$ such that $z'_{uv} = \max\{0, z_{uv} + \lambda\}$ and $z'_{u'v'} = z_{u'v'}$ for all $(u', v') \in F \backslash \{(u, v)\}$.*

---

**Algorithm 5** Online $(\gamma, e)$-balanced weight assignment (restatement of Algorithm 2)

1: **Require:** Given weighted graph $G = (V, F, w)$, hyperedge $e \subset V$ with weight $w(e)$
2: **Ensure:** Online $(\gamma, e)$-balanced weight assignment
3: **Initialize:** for all $(u, v) \in e$, set $z_{uv} = w(e)/|e|$
4: $G \leftarrow (V, F \cup \bigcup_{(u,v) \in e}(u, v), z)$
5: **While** $G$ is not online $(\gamma, e)$-balanced weight assignment **do**
6:     Select $(u, v) \neq u', v' \in e$ such that $q_{uv} > \gamma \cdot q_{u'v'}$ and $z_{u'v'} > 0$ {$q_{uv}$ is the ratio in Definition 2.1}
7:     $\lambda \leftarrow \min\{z_{uv}, \frac{\gamma - 1}{2\gamma \cdot q_{uv}}\}$
8:     $z_{uv} \leftarrow z_{uv} + \lambda$
9:     $z_{u'v'} \leftarrow z_{u'v'} - \lambda$
10: **Return** $G$

---

The following lemma upper bounds the ratio $q_{uv}$ (see Definition 2.1) of a bridge in a graph, which is later used to prove that we never set the weight of a bridge to $0$; thus, the connectivity ensures that the ST-potential is always well-defined.

**Lemma C.10** (Upper bound, see Fact 2.8 in (Kapralov et al., 2021)). *For any weighted graph $G = (V, F, z)$ and any edge $(u, v) \in F$, we have $z_{uv} \cdot q_{uv} \leq 1$, with equality if and only if $(u, v)$ is a bridge.*

The following lemma states the increment in the ST-potential when we operate a weight shift.

**Lemma C.11** (Reduction lemma, see Lemma 5.7 in (Kapralov et al., 2021)). *Given a weighted graph $G = (V, F, z)$ and real number $\gamma > 1$, let $(u, v) \neq (u', v')$ be two edges in $F$ such that $q_{uv} > \gamma \cdot q_{u'v'}$. Then for any $\lambda \leq z_{uv}$, shifting $\lambda$ weight from $(u, v)$ to $(u', v')$ results in an increment of at least $\log \left(1 + \lambda\gamma \cdot q_{uv} - \lambda \cdot q_{uv} - \lambda^2 \gamma \cdot q_{uv}^2\right)$ of the ST-potential of $G$.*

With the above results, we show that the greedy algorithm terminates in a finite number of steps.

**Theorem C.12.** *Algorithm 5 terminates in a finite number of steps.*

*Proof.* This proof follows closely from the proof of Theorem 5.8 in (Kapralov et al., 2021), except that we have a prefix-graph $G$ and we only assign new weights to the newly arrived hyperedge $e$.

First, we note that $G$ is never disconnected. Otherwise, we need to set $\lambda = z_{uv}$ for some bridge $(u, v) \in e$ so that it has zero weight after the weight shift. By Lemma C.10, we have $z_{uv} = \frac{1}{q_{uv}} >$

$\frac{\gamma-1}{2\gamma \cdot q_{uv}}$. So, we would set $\lambda$ to $\frac{\gamma-1}{2\gamma \cdot q_{uv}}$ instead, which is a contradiction. Therefore, the ST-potential $\Psi(G)$ is always well-defined.

From Lemma C.11, whenever we make a weight shift in an unbalanced pair, the ST-potential of $G$ increases by

$$\log\left(1 + \lambda\gamma \cdot q_{uv} - \lambda \cdot q_{uv} - \lambda^2\gamma \cdot q_{uv}^2\right).$$

Next, we classify our weight shift operation into two types. When $\lambda = \frac{\gamma-1}{2\gamma \cdot q_{uv}}$, the increment is at least $c_\gamma := \log\frac{1+(\gamma-1)^2}{4\gamma} > 0$. When $\lambda = z_{uv}$, the increment is positive, and $z_{u'v'}$ is set to zero. We specify these two cases as follows:

- 1. $\Psi(G)$ at least increases by a constant $c_\gamma > 0$.

- 2. $\Psi(G)$ increases by a positive amount, and $z_{u'v'}$ is set to zero.

Now, we define $G_0$ to be the weighted graph at the initialization stage of Algorithm 5, and we define $G_\infty$ to be a complete graph with node set $V$ with uniform weight $w(e) + \sum_{(u,v)\in F} z_{uv}$, where $F$ is the edge set before adding hyperedge $e$. Note that $\sum_{(u,v)\in e} z_{uv} = w(e)$ by definition, then by the monotonicity of $\Psi$, $\Psi(G) \leq \Psi(G_\infty)$ for all $G$ obtained in Algorithm 5. Therefore, there can be at most $\frac{\Psi(G_\infty)-\Psi(G)}{c_\gamma}$ steps of weight shifts of the first type.

For the second type, if $z_{uv} \neq 0$, then the number of zeros in the weight assignment for $e$ increases by 1, which only happens a finite number of times. Therefore, we only need to consider $z_{uv} = 0$. In this case, we switch the weight of $(u,v)$ and $(u',v')$, and set $\cup_{(u,v)\in e}z_{uv}$ remains the same. Then, the weight assignment $z$ can only be in finite stages without reverse. Thus, the number of weight shifts of type two is finite, and hence Algorithm 5 terminates in finite steps. □

Due to the algorithm's construction, it must output an online $(\gamma, e)$-balanced weight assignment when it terminates. Therefore, we show the existence of such weight assignment.

**Theorem C.13.** *Given a weighted graph $G = (V, F, z)$ and a hyperedge $e \subset V$ with weight $w(e)$, there exists an online $(\gamma, e)$-balanced weight assignment.*

## D  FAST ONLINE HYPERGRAPH SPECTRAL SPARSIFIER

In this section, we give an online algorithm with a faster update time based on (Bansal et al., 2019). We also include a result with success probability $1 - \frac{\delta}{\text{poly}(m)}$, which is used to give a streaming algorithm later, and show that the dependence on $\delta$ only increases the space by a $\log\log\frac{1}{\delta}$ factor.

As we mentioned previously, Bansal et al. (2019) constructed the associated graph by assigning weight $w(e)$ to each edge of the hyperedge $e$, and they sample $e$ with probability

$$p_e \propto \max_{(u,v)\in e} r_{uv}, \quad \text{where } r_{uv} = w(e) \cdot (\chi_u - \chi_v)^\top \cdot L_G \cdot (\chi_u - \chi_v).$$

We extend this procedure to the online setting by maintaining a 2-spectral approximation $\widehat{L_G(t)}$ to the graph Laplacian $L_G(t)$ at all times using online row sampling (Cohen et al., 2020). Then, we use $\widehat{L_G(t)}$ to define the sampling probabilities. Our algorithm is displayed in Algorithm 6.

The energy of the hypergraph reported by Algorithm 6 can be written as a random variable $\sum_{e\in E} X_e Q_e(x)$, where $X_e$ is $1/p_e$ with probability $p_e$ and 0 otherwise. Then, the error of our approximation is $\sum_{e\in E}(X_e - 1) \cdot Q_e(x)$. Bansal et al. (2019) simplified this term to a sub-Gaussian random process $V_x$. Then, they bound its increment by a simpler Gaussian process $U_x$, which can be further bounded by Talagrand's chaining theorems. We show that their bound on the supremum of $V_x$ can be directly applied to prove our results. Intuitively, this is because the approximation $\widehat{L_G(t)}$ always gives an overestimate of the effective resistance, which means that we over-sample the hyperedges.

We remark that to show that the output $\widehat{H}$ of Algorithm 6 satisfies the spectral sparsification guarantee in Eq. (1), it suffices to show its correctness for hypergraphs where all hyperedges have size

---

**Algorithm 6** Online hyperedge spectral sparsifier with a faster update time

---

1: **Require:** Stream of $m$ hyperedges for hypergraph $H$ with rank $r$
2: **Ensure:** Spectral sparsifier $\widehat{H}$ for $H$
3: $\widehat{H} \leftarrow \emptyset$, $\rho \leftarrow \mathcal{O}\left(\frac{r^4}{\varepsilon^2} \log \frac{m}{\delta}\right)$
4: Let $G$ be the associated graph of $H$. Let $\frac{1}{2} \cdot L_G(t) \preceq \widehat{L_G(t)} \preceq 2 \cdot L_G(t)$ for all $t \in [m]$ {Use online row sampling in Theorem B.11}
5: **for** hyperedge $e_t$ **do**
6:     **for** $(u, v) \in e_t$ **do**
7:         $\widehat{r_{(u,v)}} \leftarrow w(e_t) \cdot (\chi_u - \chi_v)^\top \cdot \widehat{L_G(t)}^{-1} \cdot (\chi_u - \chi_v)$
8:     **end for**
9:     $p_{e_t} \leftarrow \min\{1, \rho \cdot \max_{(u,v) \in e_t} \widehat{r_{(u,v)}}\}$
10:     With probability $p_{e_t}$, $\widehat{H} \leftarrow \widehat{H} \cup \frac{1}{p_{e_t}} \cdot e_t$
11: **end for**

---

between $[r/2, r]$ (see Lemma 5.2 of (Bansal et al., 2019)). In addition, we have (see Lemma 5.5 of (Bansal et al., 2019))

$$\frac{2}{r(r-1)} x^\top L_G x \le Q_H(x) \le \frac{4}{r} x^\top L_G x \quad \text{for all } x \in \mathbb{R}^n,$$

if all hyperedges have size between $[r/2, r]$. Thus, we only need to show that

$$|Q_{\widehat{H}}(x) - Q_H(x)| \le \frac{\varepsilon}{r^2} x^\top L_G x \quad \text{for all } x \in \mathbb{R}^n.$$

Next, we introduce a normalized version of $Q_e(x)$.

**Definition D.1** (Normalized energy, implicitly defined in Section 5.2 of (Bansal et al., 2019)). *For a hyperedge $e$ in $H$, let $p_e$ denote its sampling probability, and let $X_e$ be a random variable that is $1/p_e$ with probability $p_e$ and 0 otherwise. Setting $z = L_G^{1/2} x$ and $Y_{uv} = L_G^{-1/2} L_{uv} L_G^{-1/2}$ gives*

$$Q_e(x) = \max_{(u,v) \in e} x^\top L_{uv} x = \max_{(u,v) \in e} z^\top Y_{uv} z$$

*We define $W_e(z) = \max_{(u,v) \in e} z^\top Y_{uv} z$ and $W_H(z) = \sum_{e \in E} W_e(z)$. Let $\widehat{H}$ be constructed with sampling distribution $p_e$ and rescaled factor $X_e$, then $W_{\widehat{H}}(z) = \sum_{e \in E} X_e W_e(z)$.*

With the normalized definition, our desired equation becomes

$$|W_{\widehat{H}}(z) - W_H(z)| \le \frac{\varepsilon}{r^2} \qquad \text{for all } z \in B_2 \tag{2}$$

where $B_2$ is the unit-$\ell_2$-ball in the subspace restricted to the image of $L_G$. We prove Eq. (2) in the following statements. First, we draw an equivalence between $\|Y_{uv}\|$ and the effective resistance $r_{uv}$.

**Fact D.2.** *Let $\|\cdot\|$ denote the spectral norm of a matrix, we have $\|Y_{uv}\| = r_{uv}$.*

*Proof.* Notice that $Y_{uv} = L_G^{-1/2} L_{uv} L_G^{-1/2}$, so we have

$$Y_{uv} = w(e) \cdot L_G^{-1/2} \cdot (\chi_u - \chi_v)^\top (\chi_u - \chi_v) \cdot L_G^{-1/2}$$
$$= w(e) \cdot \left((\chi_u - \chi_v) \cdot L_G^{-1/2}\right)^\top \cdot (\chi_u - \chi_v) \cdot L_G^{-1/2},$$

where $e$ is the weight of the corresponding hyperedge in $H$. Notice $Y_{uv}$ is a rank-1 matrix spanned by vector $(\chi_u - \chi_v) \cdot L_G^{-1/2}$, thus, we have

$$\|Y_{uv}\| = \lambda_{\max}(Y_{uv}) = w(e) \cdot (\chi_u - \chi_v) \cdot L_G^{-1/2} \cdot \left((\chi_u - \chi_v) \cdot L_G^{-1/2}\right)^\top$$
$$= w(e) \cdot (\chi_u - \chi_v) \cdot L_G^{-1} \cdot (\chi_u - \chi_v)^\top = r_{uv}.$$

$\square$

Now, we state the bound on the supremum of the random process $V_z$ defined in (Bansal et al., 2019).

**Theorem D.3** (Supremum of random process, see Theorem 5.15 in (Bansal et al., 2019))**.** *For a hyperedge $e$ in $H$, let $G$ be the associated graph of $H$, we define the effective resistance of $e$ as $r_e = \max_{(u,v) \in e} r_{uv}$, where $r_{uv}$ is measured in graph $G$. Let $S \subset E(H)$ be a subset of hyperedges such that $\|Y_{uv}\| \leq b$ for all $(u,v) \in e$ and $e \in S$, where $l$ is some constant. For independent Rademacher variables $\varepsilon_e$ and vector $z \in \mathbb{R}^n$, let*

$$V_z = \sum_{e \in S} \varepsilon_e W_e(Z).$$

*Then, we have $\mathbb{E}\left[\sup_{z \in B_2} V_z\right] = \mathcal{O}\left(\sqrt{b \log n}\right)$, and for all $u \geq 0$, we have*

$$\mathbf{Pr}\left[\sup_{z \in B_2} V_z \geq \mathcal{O}\left(\sqrt{b \log n} + 2u\sqrt{b}\right)\right] \leq 2e^{-u^2}.$$

With the above lemmas, we show the correctness of our online algorithm.

**Lemma D.4.** *Let $\widehat{H}$ be the output of [Algorithm 6](#), it is a $(1 + \varepsilon)$-multiplicative spectral sparsifier for $H$ with probability $1 - \frac{\delta}{\text{poly}(m)}$.*

*Proof.* Fix a time $t$ in the stream, let $H$ be the hypergraph, and let $\widehat{H}$ be the sparsifier. Notice that the sampling probability of each hyperedge is solely determined by the calculation of the online leverage scores $\tau^{\text{OL}}$, which is independent of the hyperedges sampled from previous arrivals. Therefore, we can view the sampling procedure as a re-ordered procedure.

We state an iterative sampling process introduced by (Bansal et al., 2019). Let $\tau_e^{\text{OL}} = \max_{(u,v) \in e} \tau_{uv}^{\text{OL}}$, where $\tau_{uv}^{\text{OL}}$ is calculated at the time that $e$ is sampled. We round each sampling probability $p_e$ up to the nearest integer powers of 2. This ensures $p_e \geq \min\{1, \tau_e^{\text{OL}} \cdot \rho\}$, while at most doubling the expected sample complexity. Notice that hyperedges with $p_e = 1$ do not contribute to the sampling error, so we can assume $\tau_e^{\text{OL}} \cdot \rho$ for all $e \in E(H)$. Let $C_j = \{e \in E(H) \mid p_e = 2^{-j}\}$. Now, we view the process of sampling in the following way. Let $H_0 = H$, let $l = \mathcal{O}\left(\log n\right)$, and for $i \in [l]$, $H_i$ is obtained from $H_{i-1}$ by sampling each hyperedge $e$ from the set $\bigcup_{j \in \{l-i+1, l-i, \ldots, l\}} C_j$ independently with probability $1/2$ and doubling the weight of $e$ if it is sampled. Thus, an edge $e \in C_j$ that survives in $H_l$ is sampled with probability $p_e = 2^{-j}$ and has weight $X_e = 1/p_e$.

Now, for each $i \in [l]$, we define

$$W_{H_i}(z) = \sum_{j=0}^{l} \sum_{e \in C_j \cap E(H_i)} \max\left(1, 2^{i+j-l}\right) W_e(z).$$

Since $H_0 = H$ and $H_l = \widehat{H}$, by triangle inequality we have

$$\left|W_{\widehat{H}}(z) - W_H(z)\right| \leq \sum_{i=1}^{l} \left|W_{H_i}(z) - W_{H_{i-1}}(z)\right|.$$

Taking the supremum over all $z$ in $B_2$ gives

$$\sup_{z \in B_2} \left|W_{\widehat{H}}(z) - W_H(z)\right| \leq \sum_{i=1}^{l} \sup_{z \in B_2} \left|W_{H_i}(z) - W_{H_{i-1}}(z)\right|.$$

From our definition of the iterative sampling procedure, we have

$$W_{H_i}(z) - W_{H_{i-1}}(z) = \sum_{j=\ell-i+1}^{l} \sum_{e \in C_j \cap E(H_{i-1})} \varepsilon_e 2^{i+j-\ell-1} W_e(z),$$

where $\varepsilon_e$'s are independent Rademacher variables. Recall that we define $W_e(z) = \max_{(u,v) \in e} z^\top Y_{uv} z$. Then, for any $e \in \bigcup_{j \in \{l-i+1, l-i, \ldots, l\}} C_j$, we have

$$\|Y_{uv}\| = r_e = \max_{(u,v) \in e} \tau_{uv} \leq \tau_e^{\text{OL}} \leq 2^{-j}/\rho$$

where the first step follows from Fact D.2, the second step follows from the equivalence between effective resistance and leverage score (see Lemma B.7), and the third step follows from the over-sampling property of online leverage score (see Lemma B.9). So, $\|2^{i+j-1}Y_{uv}\| \leq 2^{i-l}/\rho$. Applying Theorem D.3 with $V_z = W_{H_i}(z) - W_{H_{i-1}}(z)$ and $u = \sqrt{\log \frac{m}{\delta}}$ gives

$$\mathbf{Pr}\left[\sup_{z \in B_2} V_z \geq \mathcal{O}\left(\sqrt{2^{i-l}/\rho \cdot \log \frac{m}{\delta}}\right)\right] \leq \frac{\delta}{\text{poly}(m)}.$$

Recall that we set $\rho = \frac{r^4}{\varepsilon^2} \log \frac{m}{\delta}$. Taking a union bound over the $l = \mathcal{O}(\log n)$ groups, we have

$$\sup_{z \in B_2} |W_{\widehat{H}}(z) - W_H(z)| \leq \mathcal{O}\left(\sum_{i=1}^{l} \sqrt{2^{i-l}/\rho \cdot \log \frac{m}{\delta}}\right) = \mathcal{O}\left(\frac{\varepsilon}{r^2}\right),$$

with probability $1 - \frac{\delta}{\text{poly}(m)}$. Taking a union bound over $m$ arrivals in the stream, the same bound still holds with probability $\frac{\delta}{\text{poly}(m)}$. Thus, we show the correctness of our algorithm. $\qquad\square$

The next statement bounds the sample complexity, which mainly follows by the upper bound on the sum of online leverage scores.

**Lemma D.5.** *With probability* $1 - \frac{1}{\text{poly}(m)}$, *Algorithm 6 samples* $\mathcal{O}\left(\frac{nr^4}{\varepsilon^2} \log n \log \frac{m}{\delta}\right)$ *hyperedges. In addition, it uses* $\mathcal{O}\left(n \log^2 n \log m\right)$ *bits of working memory and* $\text{poly}(n)$ *update time.*

*Proof.* It suffices to upper bound the expected number of samples $\sum_{e \in E(H)} p_e$. By our definition of $p_e$, it is upper bounded by

$$\sum_{e \in E(H)} \rho \cdot \max_{(u,v) \in e} \tau^{\mathsf{OL}}_{uv} \leq \rho \cdot \sum_{(u,v) \in E(G)} \tau^{\mathsf{OL}}(\mathbf{a}_{uv}),$$

where $\mathbf{A}$ is the incidence matrix of the associated graph $G$. By Theorem B.10, the sum of the online leverage score is bounded by $\mathcal{O}(n \log \kappa)$, where $\kappa = \|\mathbf{A}\|_2 \cdot \max_{i \in [n]} \|\mathbf{A}_i^{-1}\|_2$ is the online condition number of $\mathbf{A}$. Therefore, the expected number of samples is $\mathcal{O}(\rho n \log \kappa) = \mathcal{O}\left(\frac{nr^4}{\varepsilon^2} \log \kappa \log \frac{m}{\delta}\right)$. Note that the inner randomness of sampling each hyperedge with probability $p_e$ is independent, then by standard concentration inequalities, the number of sampled hyperedges is $\mathcal{O}\left(\frac{nr^4}{\varepsilon^2} \log \kappa \log \frac{m}{\delta}\right)$ with probability $1 - \frac{1}{\text{poly}(m)}$. Suppose that the condition of Fact C.6 is satisfied, then we have $\log \kappa = \mathcal{O}(\log n)$.

In addition, by the guarantee of online row sampling in Theorem B.11, it suffices to sample $\mathcal{O}(n \log n \log m)$ rows in the incidence matrix $\mathbf{A}$ to construct the 2-approximation $\widehat{L_G(t)}$ to the graph Laplacian $L_G(t) = \mathbf{A}^\top \mathbf{A}$. Since we only need to store $\widehat{L_G(t)}$ to compute the sampling probabilities, Algorithm 6 uses $\mathcal{O}\left(n \log^2 n \log m\right)$ bits of working memory. Moreover, the calculation of the sampling probabilities only requires $\text{poly}(n)$ time. $\qquad\square$

With Lemma D.4 and Lemma D.5, we have the following result for the online hypergraph spectral sparsifier.

**Theorem D.6** (Online hypergraph spectral sparsifier)**.** *Given a hypergraph* $H = (V, E, w)$ *with* $n$ *vertices and rank* $r$, *there exists an online algorithm that constructs a* $(1+\varepsilon)$-*spectral sparsifier with probability* $1 - \frac{\delta}{\text{poly}(m)}$ *by sampling* $\mathcal{O}\left(\frac{nr^4}{\varepsilon^2} \log n \log \frac{m}{\delta}\right)$ *hyperedges, using* $\mathcal{O}\left(n \log^2 n \log m\right)$ *bits of working memory and* $\text{poly}(n)$ *update time.*

Now, we specify the number of bits needed to store each sampled hyperedge.

**Remark D.7** (Bits of precision)**.** *Suppose that all hyperedge weights are integers from* $[\text{poly}(n)]$, *we need* $\mathcal{O}(r \log n)$ *bits to store each sampled hyperedge.*

*Proof.* First, we note that it requires $r \log n$ bits to store the nodes included in the hyperedge. Second, for each reweighted hyperedge sampled by our algorithm, we round its weight $w(e)$ to the nearest power of $(1 + \mathcal{O}(\varepsilon))$. Recall that the energy of a vector $x$ in hypergraph $H$ is an additive function: $Q_H(x) = \sum_{e \in E} w(e) \cdot \max_{(u,v)} (x_u - x_v)^2$, therefore perturbing each $w(e)$ by a $(1 + \mathcal{O}(\varepsilon))$-fraction only has an additional $(1 + \mathcal{O}(\varepsilon))$-multiplicative error in our approximation guarantee. Note that we sample each hyperedge with probability $p$ proportional to $w(e_t) \cdot (\chi_u - \chi_v)^\top \cdot \widehat{L_G(t)}^{-1} \cdot (\chi_u - \chi_v)$, where $\widehat{L_G(t)}$ is a 2-spectral sparsifier of the associated graph $G$ of the hypergraph $H$, and we rescale each sampled hyperedge by $\frac{1}{p}$. Thus, assuming that all hyperedge weights are integers from $[\text{poly}(n)]$, all eigenvalues in $L_G(t)$ are at most $\text{poly}(m)$, and so the sampling probability $p$ is at least $\frac{1}{\text{poly}(m)}$ for each hyperedge. The rescaling factor $\frac{1}{p}$ is then at most $\text{poly}(m)$, so there are $\mathcal{O}\left(\frac{\log m}{\varepsilon}\right)$ choices of powers that we need to store. Therefore, we need $\mathcal{O}(\log \log m)$ bits to store each sampled hyperedge, assuming that $\frac{1}{\varepsilon} \leq \text{polylog}(m)$. Since $m \leq \binom{n}{r} \leq \left(\frac{en}{r}\right)^r$, then $\log \log m \leq \mathcal{O}(\log r + \log \log n) \leq \mathcal{O}(r \log n)$, and so the desired claim follows. $\square$

## E  STREAMING MODEL

In this section, we provide streaming algorithms for sparsification problems with nearly optimal space. We start by introducing the well-known merge-and-reduce approach in achieving $(1 + \varepsilon)$-coresets in a data stream. An *online $(1 + \varepsilon)$-coreset* for graph sparsification for a graph $G$ defined by a stream of edges $e_1, \ldots, e_m$ is a subset $\widehat{G}$ of weighted edges of $G$ such that for any $x \in \mathbb{R}^n$ and any $t \in [m]$, we have $(1 - \varepsilon)Q_{G_t}(x) \leq Q_{\widehat{G_t}}(x) \leq (1 + \varepsilon)Q_{G_t}(x)$, where $G_t$ is the set of hyperedges of $G$ that have arrived by time $t$.

Let $S(\cdot) \to \mathbb{R}_{>0}$ denote a mapping from input parameters to the sample complexity of an online algorithm. The online coresets for graph sparsification sample $S(n, \log m, \varepsilon, \delta)$ edges for an input stream of length $m$ on a graph with $n$ nodes, accuracy $\varepsilon \in (0, 1)$, and failure probability $\delta$, with high probability. The merge-and-reduce approach partitions the stream into blocks of size $S\left(n, \log m, \frac{\varepsilon}{2 \log(mn)}, \frac{\delta}{\text{poly}(mn)}\right)$ and builds a $\left(1 + \frac{\varepsilon}{2 \log(mn)}\right)$-coreset for each block, so that each coreset can be interpreted as the leaves of a binary tree with height at most $\log(mn)$, as the binary tree is balanced and has at most $m$ leaves corresponding to the edges that arrive in the data stream. For each node in the binary tree, a coreset of size $S\left(n, \log m, \frac{\varepsilon}{2 \log(mn)}, \frac{\delta}{\text{poly}(mn)}\right)$ is built from the coresets representing the two children of the node. Assuming that the coreset construction admits a merging procedure, i.e., by taking the graph consisting of the union of the weighted edges in each of the coresets, then the root of the tree represents a coreset for the entire stream with distortion at most $\left(1 + \frac{\varepsilon}{2 \log(mn)}\right)^{\log(mn)} \leq (1 + \varepsilon)$ and failure probability $\delta$.

Cohen-Addad et al. (2023) improved the above framework by adding an online sampling procedure ahead of the merge-and-reduce approach. Suppose that the online sampling procedure is nearly optimal. Then the input stream of the merge-and-reduce approach is significantly shorter. In the graph sparsification problem, there is an online algorithm that samples $\mathcal{O}\left(\frac{n}{\varepsilon^2} \log n\right)$ edges, so the coresets only have size $S\left(n, \log \log n, \frac{\varepsilon}{2 \log \log(n)}, \frac{\delta}{\text{polylog}(n)}\right)$. This turns $\text{polylog}(n)$ factors into $\text{polylog} \log(n)$ factors, which is more space-efficient for huge graphs. We summarize this framework in Algorithm 7 and provide an illustration of the merge-and-reduce framework in Figure 4.

### E.1  GRAPH SPECTRAL SPARSIFIER

We show that the streaming framework produces space-optimal streaming algorithms for constructing graph and hypergraph sparsifiers. First, in the offline setting, an efficient way to construct $(1 + \varepsilon)$-coresets for graph sparsifier is given by Batson et al. (2014), using only $\mathcal{O}\left(\frac{n}{\varepsilon^2}\right)$ edges. We provide the formal statement below.

**Theorem E.1** (Offline algorithm for graph spectral sparsifier). *Batson et al. (2014) Given a graph $G = (V, E, w)$ with $n$ vertices, there exists an algorithm that constructs a $(1 + \varepsilon)$-spectral sparsifier with probability $1 - \frac{1}{\text{poly}(n)}$ using $\mathcal{O}\left(\frac{n}{\varepsilon^2}\right)$ edges in $\text{poly}(n)$ time.*

---

**Algorithm 7** Streaming framework of (Cohen-Addad et al., 2023), using online sampling and merge-and-reduce (restatement of Algorithm 3)

---

1: **Require:** Stream $\mathcal{S}$, online sampling procedure for $\mathcal{S}$, merge-and-reduce procedure
2: **Ensure:** Coreset on $\mathcal{S}$
3: **for** each update $s_t$ in the stream $\mathcal{S}$ **do**
4:    **if** $s_t$ is sampled by online sampling **then**
5:       Add the corresponding update to $\mathcal{S}'$
6:    **end if**
7:    Run merge-and-reduce on $\mathcal{S}'$
8: **end for**

---

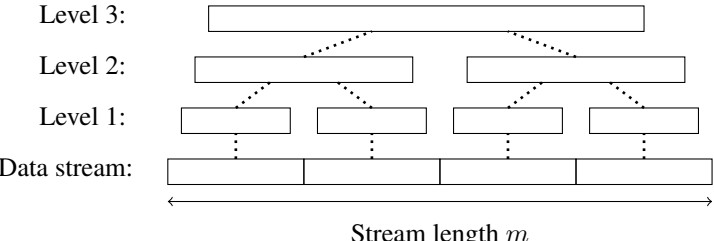

Fig. 4: Merge and reduce framework on a stream of length $m$. The coresets at level 1 are precisely the hyperedges in the block, while the coresets at level $\ell > 1$ are $\left(1 + \mathcal{O}\left(\frac{\varepsilon}{2\log(mn)}\right)\right)$-coresets of the hyperedges contained in the coresets of the children nodes in level $\ell - 1$.

We then state the result for the online graph spectral sparsifier in Cohen et al. (2020).

**Theorem E.2** (Online algorithm for graph spectral sparsifier). *Cohen et al. (2020) Given a graph $G = (V, E, w)$ with $n$ vertices and $m$ edges defined by an insertion-only stream, there exists an online algorithm that constructs a $(1+\varepsilon)$-spectral sparsifier with probability $1 - \frac{1}{\text{poly}(n)}$. The algorithm samples $\mathcal{O}\left(\frac{n}{\varepsilon^2}\log^2 n\right)$ edges and uses $\mathcal{O}\left(n\log^2 n\right)$ words of working memory and $\text{poly}(n)$ update time.*

Using the above subroutines, we show that our streaming algorithm has an optimal space up to a $\text{poly}(\log\log n)$ factor, assuming $m = \text{poly}(n)$ in the graph.

**Theorem E.3** (Streaming algorithm for graph spectral sparsifier). *Given a graph $G = (V, E, w)$ with $n$ vertices and $m$ edges defined by an insertion-only stream, there exists an algorithm that constructs a $(1 + \varepsilon)$-spectral sparsifier with probability $1 - \frac{1}{\text{poly}(n)}$. The algorithm stores $\frac{n}{\varepsilon^2} \cdot \text{poly}(\log\log n)$ edges, i.e., $\frac{n}{\varepsilon^2} \cdot \log n \, \text{poly}(\log\log n)$ bits, and has $\text{poly}(n)$ update time.*

*Proof.* The proof relies on the decomposability of the coresets, i.e., if $\widehat{G_1}$ is an $\varepsilon$-sparsifier of $G_1$ and $\widehat{G_2}$ is a $\varepsilon'$-sparsifier of $G_2$, then $\widehat{G_1} \cup \widehat{G_2}$ is a $\varepsilon'$-sparsifier of $G_1 \cup G_2$. This is simply because the energy of a vector $x$ on the graph $G$ is the sum of the energy of all edges in $G$: $Q_G(x) = \sum_{e \in G} Q_e(x)$. Then, the edges in the stream at level $l+1$ in the merge-and-reduce binary tree construct a $\varepsilon'$-sparsifier of the edges in the stream at level $l$. Thus, the accumulative error of the coreset on the roof of a tree of height $h$ is $\varepsilon' h$.

In our online framework Algorithm 7, the hyperedges at the bottom level are the output $\mathcal{S}'$ of the online algorithm, which contains $|\mathcal{S}'| = \mathcal{O}\left(\frac{n}{\varepsilon'^2}\log^2 n\right)$ edges by Theorem E.2. Thus, the height of the merge-and-reduce data structure is $h = \mathcal{O}\left(\log\frac{|\mathcal{S}'|}{|C|}\right)$, where $|C|$ is the size of each coreset in the merge-and-reduce structure. We define $C$ by Theorem E.1, which selects $\Theta(\frac{n}{\varepsilon'^2})$ hyperedges. Hence, $h = \mathcal{O}\left(\log\log n\right)$. Therefore, taking $\varepsilon' = \mathcal{O}\left(\frac{\varepsilon}{\log\log n}\right)$ achieves an accumulative error of at most $\mathcal{O}\left(\varepsilon' h = \varepsilon\right)$, which implies a $(1+\varepsilon)$-sparsifier. Then, $|C| = \mathcal{O}\left(\frac{n}{\varepsilon'^2}\right) = \mathcal{O}\left(\frac{n}{\varepsilon^2}\log\log^2 n\right)$. So, we need at most $\mathcal{O}\left(|C| \cdot h\right) = \mathcal{O}\left(\frac{n}{\varepsilon^2}\log\log^3 n\right)$ words of memory in total. The update time of our streaming algorithm directly follows from the results from Theorem E.2 and Theorem E.1.

Last, we remark that the online row sampling procedure in Theorem E.2 requires a sketch matrix of $\mathcal{O}\left(n \log^2 n\right)$ rows to define the online sampling probabilities, which exceeds our desired space complexity. We note that it suffices to use the matrix consisting of the edges in the roof of our merge-and-reduce tree structure, i.e., the output of our streaming algorithm. This gives the same guarantees for correctness and complexity as in Theorem E.2 since our output is always a 2-spectral approximation at each step $i$. We defer the formal proof to Theorem E.7. Then, since we only need to store our streaming output to define the online sampling probabilities, which has $\frac{n}{\varepsilon^2} \cdot \text{poly}(\log \log(n))$ edges, we use $\frac{n}{\varepsilon^2} \cdot \log n \, \text{poly}(\log \log(n))$ bits of working memory. $\qquad\square$

**Optimizing the Working Memory.** We introduce the method to optimize the size of the sketch matrix. In online row sampling, when a row $\mathbf{a}_i$ arrives, the algorithm samples $\mathbf{a}_i$ with probability proportional to the "$\lambda$-ridge leverage score":

$$\ell_i := \mathbf{a}_{i-1}^T \left(\mathbf{A}_{i-1}^T \mathbf{A}_{i-1} + \lambda \mathbf{I}\right)^{-1} \mathbf{a}_i,$$

where $\mathbf{A}_{i-1}$ is the prefix matrix containing all the rows arrived before $\mathbf{a}_i$ and $\lambda$ is some parameter defined in advance. However, storing each $\mathbf{A}_i$ explicitly requires a prohibitively large working memory. Instead, it suffices to use a 2-spectral approximation to $\mathbf{A}_i$ to define the sampling probability, so Cohen et al. (2020) used $\widetilde{\mathbf{A}}_i$, which is the approximation given by their online algorithm (see Algorithm 1 in Cohen et al. (2020)). However, $\widetilde{\mathbf{A}}_i$ still has $\mathcal{O}\left(n \log^2 n\right)$ rows, which is prohibitively large for our purpose, so we replace it by the matrix consisting of the edges in the roof of our merge-and-reduce tree structure, i.e., the output of our streaming algorithm. Next, we show that this modified online sampling scheme still gives the guaranties in Theorem E.2. First, we state that oversampling gives a valid graph sparsifier, i.e., the online sampling probabilities only need to be defined by $\widetilde{\ell}_i \geq \ell_i$.

**Theorem E.4** (Online Sampling Probability, Cohen et al. (2020)). *Let $\mathbf{A} \in \mathbb{R}^{m \times n}$ be an incidence matrix of a (multi-)graph with $n$ vertices and $m$ edges, whose edge weights are integers in $[\text{poly}(n)]$, let $\mathbf{a}_i$ be the $i$-th row in $\mathbf{A}$, and let $\mathbf{A}_{i-1}$ be the prefix matrix containing all the rows arrived before $\mathbf{a}_i$. We define an online sampling scheme as follows. Let $\widetilde{\ell}_i$ satisfies*

$$\widetilde{\ell}_i \geq \mathbf{a}_i^T \left(\mathbf{A}_{i-1}^T \mathbf{A}_{i-1} + \lambda \mathbf{I}\right)^{-1} \mathbf{a}_i,$$

*where $\lambda = \frac{\varepsilon}{\text{poly}(n)} \leq \frac{\varepsilon}{\sigma_{\min}^2(\mathbf{A})}$. We set the sampling probability of row $\mathbf{a}_i$ as $p_i = \min\{c\widetilde{\ell}_i, 1\}$, where $c = \mathcal{O}\left(\frac{\log m}{\varepsilon^2}\right)$, and if $\mathbf{a}_i$ is sampled, we store the re-weighted row $\frac{\mathbf{a}_i}{\sqrt{p_i}}$. Then, let $\widetilde{\mathbf{A}}_i$ be the matrix outputted by our sampling scheme for each $i \in [m]$, with probability at least $\frac{1}{\text{poly}(m)}$ we have*

$$(1 - \varepsilon)\mathbf{A}_i^\top \mathbf{A}_i \preceq \widetilde{\mathbf{A}}_i^T \widetilde{\mathbf{A}}_i \preceq (1 + \varepsilon)\mathbf{A}_i^\top \mathbf{A}_i.$$

We remark that we set $c = \mathcal{O}\left(\frac{\log m}{\varepsilon^2}\right)$, instead of $c = \mathcal{O}\left(\frac{\log n}{\varepsilon^2}\right)$ in Algorithm 1 in Cohen et al. (2020), to achieve a high success probability $1 - \frac{1}{\text{poly}(m)}$, enabling us to condition on the correctness upon each arrival $\mathbf{a}_i$. Next, we state an upper bound on the sum of ridge leverage scores.

**Theorem E.5** (Sum of ridge leverage scores, Cohen et al. (2020)). *In the sampling schemed described in Theorem E.4, we define*

$$\ell_i = \mathbf{a}_i^T \left(\mathbf{A}_{i-1}^T \mathbf{A}_{i-1} + \lambda \mathbf{I}\right)^{-1} \mathbf{a}_i,$$

*where $\lambda = \frac{\varepsilon}{\text{poly}(n)} \leq \frac{\varepsilon}{\sigma_{\min}^2(\mathbf{A})}$. Then we have $\sum_{i \in [m]} \ell_i = \mathcal{O}\left(n \log n\right)$.*

The upper bound in Theorem E.5 gives an upper bound on the sum of sampling probabilities; however, we cannot apply standard concentration inequalities to bound the number of sampled rows. This is because in the online subroutine, the sample probability of the current $\mathbf{a}_i$ depends on whether the previous rows are sampled, which is still the case after we replace the sketch matrix in the online subroutine by our output from the merge-and-reduce process. Thus, we need to apply the following Freedman's inequality, which does not assume the independence of each sample.

**Theorem E.6** (Freedman's inequality, Freedman (1975))**.** *Let* $Y_0, Y_1, \ldots, Y_n$ *be a scalar martingale with difference sequence* $X_1, \ldots, X_n$, *i.e.,* $Y_0 = 0$ *and* $Y_t = Y_{t-1} + X_t$ *for all* $t \in [n]$. *Let* $|X_t| \leq R$ *for all* $t \in [n]$ *with high probability. We define the predictable quadratic variation process of the martingale by* $w_k := \sum_{t=1}^{k} \mathbb{E}_{t-1} \left[ X_t^2 \right]$, *for* $k \in [n]$. *Then for all* $\epsilon \geq 0$ *and* $\sigma^2 > 0$, *and every* $k \in [n]$,

$$\mathbf{Pr} \left[ \max_{t \in [k]} |Y_t| > \epsilon \text{ and } w_k \leq \sigma^2 \right] \leq 2 \exp \left( - \frac{\epsilon^2/2}{\sigma^2 + R\epsilon/3} \right).$$

Now, we show that the modified online subroutine satisfies the guaranties in Theorem E.2. Thus, we only need to store a sketch matrix of $\mathcal{O} \left( \frac{n}{\varepsilon^2} \operatorname{poly}(\log \log(n)) \right)$ rows to define the online sampling probabilities, which achieves the desired space complexity.

**Theorem E.7.** *Let graph* $G = (V, E, w)$ *with* $n$ *vertices and* $m$ *edges be defined by an insertion-only stream, let* $\mathbf{A}$ *be the incidence matrix of* $G$, *and let* $\mathbf{a}_i$ *be the* $i$-*th row in* $\mathbf{A}$. *We define an online sampling scheme as follows. Let* $\widetilde{\ell}_i$ *satisfies*

$$\widetilde{\ell}_i = \mathcal{O}\left(1\right) \cdot \mathbf{a}_i^T \left( \widehat{\mathbf{A}}_{i-1}^T \widehat{\mathbf{A}}_{i-1} + \lambda \mathbf{I} \right)^{-1} \mathbf{a}_i,$$

*where* $\lambda = \frac{\varepsilon}{\operatorname{poly}(n)} \leq \frac{\varepsilon}{\sigma_{\min}^2(\mathbf{A})}$ *and* $\widehat{\mathbf{A}}_{i-1}$ *is the output of our streaming algorithm the last round (see the description in Theorem E.3). We set the sampling probability of row* $\mathbf{a}_i$ *as* $p_i = \min\{c\widetilde{\ell}_i, 1\}$, *where* $c = \mathcal{O}\left( \frac{\log m}{\varepsilon^2} \right)$, *and if* $\mathbf{a}_i$ *is sampled, we store the re-weighted row* $\frac{\mathbf{a}_i}{\sqrt{p_i}}$. *Then, let* $\widetilde{\mathbf{A}}_i$ *be the matrix outputted by our sampling scheme for each* $i \in [m]$, *with probability at least* $\frac{1}{\operatorname{poly}(m)}$ *we have*

$$(1 - \varepsilon)\mathbf{A}_i^\top \mathbf{A}_i \preceq \widetilde{\mathbf{A}}_i^T \widetilde{\mathbf{A}}_i \preceq (1 + \varepsilon)\mathbf{A}_i^\top \mathbf{A}_i.$$

*The algorithm samples* $\mathcal{O}\left( \frac{n}{\varepsilon^2} \log m \log n \right)$ *edges and uses* $\operatorname{poly}(n)$ *update time.*

*Proof.* First, we prove the correctness of our algorithm by applying Theorem E.5 in an iterative way. For initial stage, since $\mathbf{a}_1$ is sampled by the online subroutine, $\widetilde{\mathbf{A}}_1$ is a $(1+\varepsilon)$-spectral approximation of $\mathbf{A}_1$, and $\widehat{\mathbf{A}}_1$ is a 2-spectral approximation of $\mathbf{A}_1$. Now, conditioning on that $\widehat{\mathbf{A}}_{i-1}$ is a 2-spectral approximation of $\mathbf{A}_{i-1}$. Thus, the eigenvalues of $\widehat{\mathbf{A}}_{i-1}^T \widehat{\mathbf{A}}_{i-1} + \lambda \mathbf{I}$ and $\mathbf{A}_{i-1}^T \mathbf{A}_{i-1} + \lambda \mathbf{I}$ are within a constant fraction. Then, by our construction, we have

$$\widetilde{\ell}_i = \mathcal{O}\left(1\right) \cdot \mathbf{a}_i^T \left( \widehat{\mathbf{A}}_{i-1}^T \widehat{\mathbf{A}}_{i-1} + \lambda \mathbf{I} \right)^{-1} \mathbf{a}_i \geq \mathbf{a}_i^T \left( \mathbf{A}_{i-1}^T \mathbf{A}_{i-1} + \lambda \mathbf{I} \right)^{-1} \mathbf{a}_i.$$

Now, consider the next arrival $\mathbf{a}_i$, by the guarantee of Theorem E.5, the output $\widetilde{\mathbf{A}}_i$ of the online procedure is still a $(1 + \varepsilon)$-approximation with probability $1 - \frac{1}{\operatorname{poly}(m)}$. Condition on this event, by the analysis in Theorem E.3, the output $\widehat{\mathbf{A}}_i$ of our streaming algorithm is a 2-approximation with probability $1 - \frac{1}{\operatorname{poly}(m)}$. Following this induction, we prove that $\widetilde{\mathbf{A}}_i$ is a $(1 + \varepsilon)$-approximation for each $i \in [m]$. Since for each step, the success probability is at least $1 - \frac{1}{\operatorname{poly}(m)}$, we can union bound across the $m$ arrivals.

Next, we bound the number of sampled rows. From our induction, we also prove that $\widehat{\mathbf{A}}_i$ is a $(1+\varepsilon)$-approximation for each $i \in [m]$ with probability $1 - \frac{1}{\operatorname{poly}(m)}$. By our construction of the scores $\widetilde{\ell}_i$, we have

$$\widetilde{\ell}_i = \mathcal{O}\left(1\right) \cdot \mathbf{a}_i^T \left( \widehat{\mathbf{A}}_{i-1}^T \widehat{\mathbf{A}}_{i-1} + \lambda \mathbf{I} \right)^{-1} \mathbf{a}_i = \mathcal{O}\left(1\right) \cdot \mathbf{a}_i^T \left( \mathbf{A}_{i-1}^T \mathbf{A}_{i-1} + \lambda \mathbf{I} \right)^{-1} \mathbf{a}_i.$$

Then, by the upper bound in Theorem E.5, we have $\sum_{i \in [m]} \widetilde{\ell}_i = \mathcal{O}\left(1\right) \cdot \sum_{i \in [m]} \widetilde{\ell}_i = \mathcal{O}\left(n \log n\right)$. Moreover, we have $\sum_{i \in [m]} p_i \leq c \cdot \sum_{i \in [m]} \widetilde{\ell}_i = \mathcal{O}\left( \frac{n}{\varepsilon^2} \log m \log n \right)$.

Next, for each row $\mathbf{a}_i$, we define $X_i$ as the indicator variable for $\mathbf{a}_i$ shifted by $p_i$, i.e., $X_i = 1 - p_i$ if $\mathbf{a}_i$ is sampled and $X_i = -p_i$ otherwise. Thus, $\mathbb{E}_{i-1} \left[ X_i \right] = 0$. Then we define the martingale as $Y_0 = 0$ and $Y_i = \sum_{j \in [i]} X_j$ for each $i \in [m]$, which is the difference between the number of sampled

rows and $\sum_{i \in [m]} p_i$. For each $i \in [m]$, we have $|X_i| \le 1$ and $\underset{i-1}{\mathbb{E}} \left[ X_i^2 \right] = p_i(1 - p_i) \le p_i$, so $w_i := \sum_{j \in [i]} \underset{j-1}{\mathbb{E}} \left[ X_j^2 \right] \le \sum_{j \in [i]} p_j$. Now, applying the Freedman's inequality (c.f. Theorem E.6) with $\epsilon = \log m \cdot \sqrt{\sum_{i \in [m]} p_i}$ and $\sigma^2 = \sum_{i \in [m]} p_i$, we have

$$\mathbf{Pr} \left[ |Y_m| > \log m \cdot \sqrt{\sum_{i \in [m]} p_i} \right] \le \frac{1}{\text{poly}(m)}.$$

That is, the number of sampled rows is $\mathcal{O}\left( \frac{n}{\varepsilon^2} \log m \log n \right)$ with high probability. $\qquad\square$

### E.2 Optimizing the Working Memory of Online Hypergraph Sparsification

In this section, we optimize the working memory required in our online hypergraph sparsifier, which is applied as a black-box in our streaming algorithm for hypergraph sparsification in Algorithm 7. Recall that both algorithms in Theorem C.7 and Theorem D.6 store a sketch matrix obtained by online row sampling the associated graph, which is used to define the sampling probabilities of the hyperedges. We show that the sketch matrix can be replaced by the output of the streaming algorithm introduced in Appendix E.1 with fewer rows. The construction is based on the subroutine in Theorem E.7. The following is the improved statement of Theorem C.7.

**Theorem E.8** (Online hypergraph spectral sparsifier). *Given a hypergraph $H = (V, E, w)$ with $n$ vertices, $m$ hyperedges and rank $r$, there exists an online algorithm with $nr \log n \cdot \text{poly}(\log \log m)$ bits of working memory that constructs a $(1 + \varepsilon)$-spectral sparsifier with probability $1 - \frac{1}{\text{poly}(m)}$ by sampling $\mathcal{O}\left( \frac{n}{\varepsilon^2} \log n \log m \log r \right)$ hyperedges.*

*Proof.* Recall that we maintain a matrix $\mathbf{M}$ that is a 2-approximate spectral approximation of the re-weighted incidence matrix $\mathbf{Z}_t^{1/2} \mathbf{A}_t$ in Algorithm 4. We slightly change the procedure to define $\mathbf{M}$: when a hyperedge $e_{t+1}$ arrives, we run GetWeightAssignment$(\mathbf{M}, e)$ to obtain the weight assignment vector $z_{t+1}$; Then, for each edge $uv$ in the clique of $e_{t+1}$, we sample $\mathbf{a}_{uv} \cdot \sqrt{z_{t+1,uv}}$ by online row sampling and push it to the merge-and-reduce data structure if sampled (see Figure 4); then we set $\mathbf{M}$ to be the weighted matrix obtained by the merge-and-reduce procedure.

By the guarantee of Theorem E.3, sampling $n \, \text{poly}(\log \log m)$ rows to $\mathbf{M}$ by the merge-and-reduce process still gives a 2-approximation to $\mathbf{Z}_t^{1/2} \mathbf{A}_t$. Moreover, we can optimize the working memory by the subroutine in Theorem E.7. That is, we can use $\mathbf{M}$ itself to define the online sampling probability for the next row in matrix $\mathbf{Z}^{1/2} \mathbf{A}$.

Then, it suffices to show that the updated procedure still satisfies the independence guarantees in the conditions of Theorem C.1. The analysis follows from the decoupling technique in Lemma C.3. The above procedure that constructs the sketch matrix $\mathbf{M}$, which includes the online row sampling process and the merge-and-reduce process, and the sampling of the hyperedges are separate procedures with independent inner randomness. So, if we fix the inner randomness in the online row sampling process and the merge-and-reduce process in the construction of $\mathbf{M}$, then the sampling probabilities are fixed. Then, given these fixed sampling probabilities, the sampling of the hyperedges can be viewed as an offline procedure, and for each $t$, the sampling of $e_t$ is independent of whether the previous hyperedges are sampled. $\qquad\square$

We next show the improved statement of Theorem D.6.

**Theorem E.9** (Online hypergraph spectral sparsifier). *Given a hypergraph $H = (V, E, w)$ with $n$ vertices and rank $r$, there exists an online algorithm that constructs a $(1 + \varepsilon)$-spectral sparsifier with probability $1 - \frac{\delta}{\text{poly}(m)}$ by sampling $\mathcal{O}\left( \frac{nr^4}{\varepsilon^2} \log n \log \frac{m}{\delta} \right)$ hyperedges, using $nr \log n \cdot \text{poly}(\log \log m)$ bits of working memory and $\text{poly}(n)$ update time.*

*Proof.* We use the same construction in Theorem E.8. Recall that we do online row sampling on the incidence matrix $\mathbf{A}$ and obtain a 2-approximation $\widehat{L_G(t)}$ to the Laplacian $L_G(t) = \mathbf{A}^T \mathbf{A}$ of the associated graph. Now, when an edge $\mathbf{a}_i$ is sampled by the online row sampling procedure, we again

push it to the merge-and-reduce structure, and use the output to define the next online sampling probability. By Theorem E.3 and Theorem E.7, the output is still a 2-approximation to $\mathbf{A}$, and hence it suffices for our analysis in Appendix D. □

### E.3 HYPERGRAPH SPECTRAL SPARSIFIER

For the hypergraph sparsification problem, we apply the offline algorithms introduced by (Jambulapati et al., 2023; Lee, 2023) to merge the coresets in the two child nodes.

**Theorem E.10** (Offline algorithm for hypergraph spectral sparsifier). *Jambulapati et al. (2023); Lee (2023) Given a hypergraph $H = (V, E, w)$ with $n$ vertices, $m$ hyperedges and rank $r$, there exists an algorithm that constructs a $(1 + \varepsilon)$-spectral sparsifier with probability $1 - \frac{1}{\text{poly}(m)}$ using $\mathcal{O}\left(\frac{n}{\varepsilon^2} \log n \log r\right)$ hyperedges in $\tilde{\mathcal{O}}(mr)$ time.*

Notice that there is no guarantee that we can find a $(\gamma, e)$-balanced weight assignment in Definition C.8 in polynomial time, so using the online algorithm given by Theorem E.8 as a subroutine does not imply a fast update time. Therefore, we instead apply Theorem E.9 with $\text{poly}(n)$ update time as the online sampling subroutine. This will lose a $\text{poly}(r)$ factor in the online sample complexity; however, it is acceptable since the streaming framework reduces it to $\text{polylog}(r)$. We also include the result by applying Theorem E.8 in our statement, which has a better space bound.

**Theorem E.11** (Streaming algorithm for hypergraph spectral sparsifier). *Given a hypergraph $H = (V, E, w)$ with $n$ vertices, $m$ edges and rank $r$ defined by an insertion-only stream, there exists an algorithm that with probability $1 - \frac{1}{\text{poly}(m)}$, constructs a $(1 + \varepsilon)$-spectral sparsifier, storing $\frac{n}{\varepsilon^2} \log n \cdot \text{poly}(\log r, \log \log m)$ hyperedges, i.e., $\frac{rn}{\varepsilon^2} \log^2 n \cdot \text{poly}(\log r, \log \log m)$ bits, and using $\text{poly}(n)$ update time. There is also an algorithm that stores $\frac{n}{\varepsilon^2} \log n \log r \cdot \text{poly}(\log \log m)$ hyperedges, i.e. $\frac{rn}{\varepsilon^2} \log^2 n \log r \cdot \text{poly}(\log \log m)$ bits, and uses exponential update time.*

*Proof.* We have $|\mathcal{S}'| = \mathcal{O}\left(\frac{nr^4}{\varepsilon'^2} \log n \log m\right)$ by Theorem E.9 and $|C| = \Theta\left(\frac{n}{\varepsilon'^2} \log n \log r\right)$ by Theorem E.10. Thus, the height $h = \mathcal{O}\left(\log \frac{|\mathcal{S}'|}{|C|}\right) = \mathcal{O}(\log r + \log \log m)$. Taking $\varepsilon' = \varepsilon/h$, our total space usage is at most $\mathcal{O}(|C| \cdot h) = \mathcal{O}\left(\frac{n}{\varepsilon'^2} \log n \log r \cdot (\log r + \log \log m)^3\right)$ words.

Each time a hyperedge arrives, the online algorithm needs $\text{poly}(n)$ time to process by Theorem E.9. In addition, we at most need to merge $2h$ coresets each with $m' = |C|$ edges, which takes $\tilde{\mathcal{O}}(m'r) \cdot h = \text{poly}(n)$ update time by Theorem E.10. Thus, we need $\text{poly}(n)$ update time in total.

If we use Theorem E.8 instead, then $|S'| = \mathcal{O}\left(\frac{n}{\varepsilon^2} \log n \log m \log r\right)$ and $h = \mathcal{O}(\log \log m)$. Hence, we need $\mathcal{O}(|C| \cdot h) = \mathcal{O}\left(\frac{n}{\varepsilon^2} \log n \log r \cdot \log \log^3 m\right)$ words of memory in total.

Notice that the failure probabilities of both the offline and the online subroutines are $\frac{1}{\text{poly}(m)}$, thus, the failure probability of our streaming algorithm follows by a union bound across all times. □

### E.4 HYPERGRAPH SPECTRAL SPARSIFIER - HIGH PROBABILITY

In this section, we provide a streaming algorithm that succeeds with probability at least $1 - \frac{\delta}{\text{poly}(m)}$. In our online algorithm in Theorem D.6, we lose a $\log \frac{1}{\delta}$ factor in the sample complexity to boost the failure probability to $\delta$, which may be prohibitively large if $\delta = \frac{1}{c^n}$. With the streaming framework of (Cohen-Addad et al., 2023), we reduce it to a $\log \log \frac{1}{\delta}$ factor.

**Theorem E.12** (Small failure probability). *Given a hypergraph $H = (V, E, w)$ with $n$ vertices, $m$ hyperedges, and rank $r$ defined by an insertion-only stream, there exists an algorithm that constructs a $(1 + \varepsilon)$-spectral sparsifier with probability $1 - \frac{\delta}{\text{poly}(m)}$ storing $\frac{n}{\varepsilon^2} \log n \cdot \text{poly}(\log r, \log \log \frac{m}{\delta})$ hyperedges, i.e., $\frac{rn}{\varepsilon^2} \log^2 n \cdot \text{poly}(\log r, \log \log \frac{m}{\delta})$ bits.*

*Proof.* First, we describe a deterministic offline algorithm that constructs a $(1 + \varepsilon)$-hypergraph spectral sparsifier. It loses more $\text{poly}(n)$ factors at runtime while it does not fail. Since Theorem E.10 gives a randomized algorithm that finds the sparsifier with $\mathcal{O}\left(\frac{n}{\varepsilon^2} \log n \log r\right)$ hyperedges given any

hypergraph with non-zero probability, there must exist such a sparsifier. Let $m$ denote the total number of hyperedges. We simply traverse through all possible groups of $\mathcal{O}\left(\frac{n}{\varepsilon^2} \log n \log r\right)$ hyperedges, where there are $\binom{m}{\mathcal{O}\left(\frac{n}{\varepsilon^2} \log n \log r\right)}$ of them. For each group, we test it on the net of points $x \in \mathbb{R}^n$ given by the chaining argument in (Jambulapati et al., 2023) and report this group of hyperedges if it successfully approximates the energy of the hypergraph $Q_H(x)$.

Then, we use the online algorithm in Theorem E.9 with failure probability $\frac{\delta}{\text{poly}(m)}$ to construct the stream $\mathcal{S}'$ and the offline deterministic algorithm mentioned above to construct the coresets. The calculation of space complexity follows from the same argument in Theorem E.11. Here, we have $|\mathcal{S}'| = \mathcal{O}\left(\frac{nr^4}{\varepsilon'^2} \log n \log \frac{m}{\delta}\right)$ by Theorem E.9 and $|C| = \Theta\left(\frac{n}{\varepsilon'^2} \log n \log r\right)$ by Theorem E.10. Thus, the height $h = \mathcal{O}\left(\log \frac{|\mathcal{S}'|}{|C|}\right) = \mathcal{O}\left(\log r + \log \log \frac{m}{\delta}\right)$. Taking $\varepsilon' = \varepsilon/h$, our total space usage is at most $\mathcal{O}\left(|C| \cdot h\right) = \mathcal{O}\left(\frac{n}{\varepsilon'^2} \log n \log r \cdot (\log r + \log \log \frac{m}{\delta})^3\right)$ words. Unfortunately, we do not have the $\text{poly}(n)$ update time guarantee due to the offline deterministic algorithm.

Notice that the failure probability of the online subroutines is $\frac{\delta}{\text{poly}(m)}$, and the offline subroutine is deterministic, thus, the failure probability of our streaming algorithm follows by a union bound across all times. $\qquad\square$

### E.5 ADVERSARIALLY ROBUST HYPERGRAPH SPARSIFICATION

In this section, we apply the result in Appendix E.4 to achieve an adversarially robust streaming algorithm. The adversarially robust model can be captured by the following two-player game between a streaming algorithm $\mathcal{P}$ and an adversary $\mathcal{A}$ that produces adaptive inputs to $\mathcal{P}$. Given a query function $\mathcal{Q}$, the game proceeds over $m$ rounds, and in the $t$-th round:

(1) $\mathcal{A}$ determines an input $s_t$, which possibly depends on previous outputs from $\mathcal{P}$.

(2) $\mathcal{P}$ processes $s_t$ and outputs its answer $Z_t$ to the query function $\mathcal{Q}$.

(3) $\mathcal{A}$ receives and records the response $Z_t$.

The goal of $\mathcal{P}$ is to produce a correct answer $Z_t$ to the query function $\mathcal{Q}$ based on the previously arrived data stream $\{s_1, \ldots, s_t\}$ sent by the adaptive adversary $\mathcal{A}$, at all times $t \in [m]$. We now provide an adversarially robust streaming algorithm for hypergraph sparsification.

We begin with the definition of the $\varepsilon$-flip number, which upper bounds the number of multiplicative increments of the output of a streaming algorithm.

**Definition E.13** ($\varepsilon$-flip number)**.** *Let $\varepsilon \geq 0$ and let $y = (y_0, y_1, \ldots, y_m)$ be a sequence of real numbers. Then, the $\varepsilon$-flip number $\lambda_{\varepsilon,m}(y)$ of the sequence $y$ is the maximum $k$ for which there exists $0 \leq i_1 < \ldots < i_k \leq m$ so that $y_{i_{j-1}} \notin (1 \pm \varepsilon) y_{i_j}$ for every $j = 2, 3, \ldots, k$. In particular, for a function $g : \mathbb{R}^n \to \mathbb{R}$ and a class of data stream $\mathcal{S} \subset [n]^m$, the $(\varepsilon, m)$-flip number $\lambda_{\varepsilon,m}(g)$ of $g$ over $\mathcal{S}$ is the maximum $\varepsilon$-flip number of the sequence $\bar{y} = (y_0, y_1, \ldots, y_m)$ defined by $y_t = g(s_1, \ldots, s_t)$, over all choices of data streams $S = (s_1, \ldots, s_m) \in \mathcal{S}$.*

Ben-Eliezer et al. (2020) introduced a framework for vector-based problems, i.e., one computes a target function $g$ on the frequency vector induced by the data stream, which transforms any non-robust streaming algorithm to an adversarially robust streaming algorithm. The core idea is that we only change the estimate $\hat{g}$ when it increases by an $\varepsilon$-fraction, so if the $\varepsilon$-flip number is small, then the total number of input streams that we need to handle is also relatively small, and we can union bound across them by setting a sufficiently small failure probability $\delta$ for the non-robust streaming algorithm. We adapt their framework to solve the sparsification problems.

**Robust Graph and Hypergraph Sparsification.** We start by introducing an adversarially robust streaming algorithm for graph sparsification. A challenge in our problem is that our output is a sparsified graph that preserves the energy of all vectors, which is not a real number, so we need to define a proper way to change the output. Recall that the energy of a vector $x$ on $G$ is the quadratic form $x^T L_G x$, where $L_G$ is the graph Laplacian. Thus, we use the eigenvalues of $L_G$ to decide whether we change the output. We state the formal algorithm and its guarantees as follows.

**Theorem E.14** (Robust graph sparsification). *Given a (multi-)graph $G = (V, E, w)$ with $n$ vertices and $m$ edges defined by an insertion-only stream, there exists an adversarially robust algorithm that constructs a $(1+\varepsilon)$-spectral sparsifier with probability $1 - \frac{\delta}{\mathrm{poly}(m)}$ storing $\frac{n}{\varepsilon^2} \cdot \mathrm{poly}(\log n, \log \log \frac{m}{\delta})$ edges, i.e., $\frac{n}{\varepsilon^2} \cdot \mathrm{poly}(\log n, \log \log \frac{m}{\delta})$ bits.*

*Proof.* First, we bound the $\varepsilon$-flip number in our problem, which is the number of times that $L_{G'} \succeq (1+\varepsilon) \cdot L_G$, where $G'$ is a graph later in the data stream. Recall that our stream is insertion-only, so all the eigenvalues of $G$ are increasing. Note that the flip does not occur if none of the eigenvalues of $G$ increases by an $\varepsilon$-fraction. Here, without loss of generality, we assume that all eigenvalues are non-zero, since we can consider it a flip when the eigenvalue first becomes non-zero, and there are at most $n$ such flips. Assuming that all edge weights are within $\mathrm{poly}(n)$, the eigenvalues at the end of the stream are upper bounded by $m \, \mathrm{poly}(n)$, and so each eigenvalue can increases by an $\varepsilon$-fraction for at most $\frac{\log m}{\varepsilon}$ times. Thus, the $\varepsilon$-flip number is at most $\lambda_{\varepsilon,m}(G) = \frac{n \log m}{\varepsilon}$.

We next state the formal algorithm for robust graph sparsification. We run a non-robust streaming algorithm with parameters $\varepsilon' = \frac{\varepsilon}{8}$ and $\delta' = \frac{\delta}{\mathrm{poly}(m) \cdot \binom{m}{\lambda} T^\lambda}$, where $\lambda := \lambda_{\frac{\varepsilon}{8},m}(G)$ and $\log T$ are the bits of precision. For the sequence of outputs $\widehat{G}_1, \ldots, \widehat{G}_m$, let $\mathcal{G}_1 = \widehat{G}_1$, we set $\mathcal{G}_t = \mathcal{G}_{t-1}$ when all eigenvalues of $L_{\widehat{G}_t}$ are within a $(1 + \frac{\varepsilon}{8})$-fraction of that of $L_{\mathcal{G}_{t-1}}$, otherwise we set $\mathcal{G}_t = \widehat{G}_t$. The output to the adversary is the sequence $\mathcal{G}_1, \ldots \mathcal{G}_m$.

We prove the correctness of the algorithm as follows. Consider a fixed time $t$ when we change the output, and let $t'$ be any time after $t$ before the next output change. Note that $\widehat{G}_{t'}$ is a $\frac{\varepsilon}{8}$-sparsifier of $G_{t'}$ and all eigenvalues of $L_{\widehat{G}_{t'}}$ are within a $(1 + \frac{\varepsilon}{8})$-fraction of that of $L_{\widehat{G}_t}$, so all eigenvalues of $L_{G_{t'}}$ are within a $(1 + \frac{\varepsilon}{2})$-fraction of that of $L_{G_t}$. This implies that $L_{G_{t'}} \preceq (1 + \frac{\varepsilon}{2}) \cdot L_{G_t}$, and so $x^\top L_{G_{t'}} x \leq (1 + \frac{\varepsilon}{2}) \cdot x^\top L_{G_t} x$ for all vectors $x \in \mathbb{R}^n$. Therefore, $\mathcal{G}_{t'} = \widehat{G}_t$ is an $\varepsilon$-sparsifier of $G_{t'}$. Moreover, we can assume the adversary to be deterministic (see the proof of Lemma 3.5 in (Ben-Eliezer et al., 2020). Then, the number of output sequences is at most $\binom{m}{\lambda} T^\lambda$ and they at most determine $\binom{m}{\lambda} T^\lambda$ choices of input streams. Since we set the failure probability as $\delta' = \frac{\delta}{\mathrm{poly}(m) \cdot \binom{m}{\lambda} T^\lambda}$, we can union bound across all choices of input streams, ensuring the correctness of our algorithm.

By Theorem E.12, we have that the non-robust streaming algorithm with parameters $\varepsilon, \delta$ stores $\frac{n}{\varepsilon^2} \log n \cdot \mathrm{poly}(\log \log \frac{m}{\delta})$ edges. Therefore, since we require $\log T = \mathcal{O}(\log n + \log \log m)$ bits of precisions, the robust algorithm stores $\frac{n}{\varepsilon'^2} \log n \cdot \mathrm{poly}(\log \log \frac{m}{\delta'}) = \frac{n}{\varepsilon^2} \log^2 n \cdot \mathrm{poly}(\log \log \frac{m}{\delta})$ edges. $\square$

Next, we apply the above result to construct a robust hypergraph sparsification algorithm. Note that the energy of a vector $x$ on a hyperedge $e$ is defined as the maximum energy of $x$ on each edge $(u, v)$ in the clique of $e$, so we cannot directly define the $\varepsilon$-flip number by the sparsified hypergraph. Instead, we run a separated robust subroutine that constructs a sparsifier for the associated graph with higher accuracy, then we decide whether to change the output by the eigenvalues of the sparsified associated graph. We introduce the formal algorithm and analyze its guarantees as follows.

**Theorem E.15** (Robust hypergraph sparsification). *Given a graph $H = (V, E, w)$ with $n$ vertices, $m$ edges, and rank $r$ defined by an insertion-only stream, there exists an adversarially robust algorithm that constructs a $(1 + \varepsilon)$-spectral sparsifier with probability $1 - \frac{\delta}{\mathrm{poly}(m)}$ storing $\frac{n}{\varepsilon^2} \mathrm{poly}(\log n, \log r, \log \log \frac{m}{\delta})$ hyperedges and $\frac{n r^4}{\varepsilon^2} \cdot \mathrm{poly}(\log n, \log \log \frac{m}{\delta})$ edges in the associated graph, i.e., $\frac{n r^5}{\varepsilon^2} \cdot \mathrm{poly}(\log n, \log r, \log \log \frac{m}{\delta})$ bits in total.*

*Proof.* We first state the formal algorithm for robust hypergraph sparsification. We run the robust streaming graph sparsification algorithm in Theorem E.14 on the associated graph $G$ of the hypergraph $H$ defined by the stream with parameters $\tilde{\varepsilon} = \frac{\varepsilon}{8r^2}$ and $\tilde{\delta} = \delta$, and we have the outputs $\mathcal{G}_1, \ldots, \mathcal{G}_m$. Here, we note that $G$ is defined by the standard associated graph definition in Definition B.3, but not the balanced-weight version introduced in Section 2. We run a non-robust hypergraph sparsification algorithm separately with parameters $\varepsilon' = \frac{\varepsilon}{8}$ and $\delta' = \frac{\delta}{\mathrm{poly}(m) \cdot \binom{m}{\lambda} T^\lambda}$, where $\lambda := \lambda_{\frac{\varepsilon}{8r},m}(G)$ and $\log T$ are the bits of precision, and we have the outputs $\widehat{H}_1, \ldots, \widehat{H}_m$. Let

$\mathcal{H}_1 = \widehat{H}_1$, we set $\mathcal{H}_t = \mathcal{H}_{t-1}$ if $\mathcal{G}_t = \mathcal{G}_{t-1}$, otherwise we set $\mathcal{H}_t = \widehat{H}_t$. The output to the adversary is the sequence $\mathcal{H}_1, \ldots \mathcal{H}_m$.

We prove the correctness of the algorithm as follows. Consider a fixed time $t$ when we change the output, and let $t'$ be any time after $t$ before the next output change. We note that $t$ is also the time when the robust graph sparsification algorithm changes its output. Then, from the analysis in Theorem E.14, we have $L_{G_{t'}} \preceq (1 + \frac{\varepsilon}{2r^2}) \cdot L_{G_t}$ and $x^\top L_{G_{t'}} x \leq (1 + \frac{\varepsilon}{2r^2}) \cdot x^\top L_{G_t} x$ for all vectors $x \in \mathbb{R}^n$. Recall that the energy of $x$ on the hypergraph $H$ is $Q_H(x) = \sum_{e \in E} \max_{(u,v) \in e} x^\top L_{uv} x$, which is at least $\frac{1}{r^2} \cdot \sum_{e \in E} \sum_{(u,v) \in e} x^\top L_{uv} x = \frac{1}{r^2} \cdot x^\top L_G x$, since there are at most $\binom{r}{2}$ edges $(u,v) \in e$. Then, we have

$$Q_H(x) \leq Q_{H'}(x) + \frac{\varepsilon}{2r^2} \cdot x^\top L_{G_t} x \leq (1 + \frac{\varepsilon}{2}) \cdot Q_{H'}(x).$$

Therefore, $\mathcal{H}_{t'} = \widehat{H}_t$ is an $\varepsilon$-sparsifier of $H_{t'}$. Since we set the failure probability as $\delta' = \frac{\delta}{\text{poly}(m) \cdot \binom{m}{\lambda} T^\lambda}$, we can union bound across all $\binom{m}{\lambda} T^\lambda$ choices of input streams, ensuring the correctness of our algorithm.

By Theorem E.12, we have that the non-robust streaming algorithm with parameters $\varepsilon', \delta'$ stores $\frac{n}{\varepsilon^2} \log n \cdot \text{poly}(\log r, \log \log \frac{m}{\delta})$ edges. Note that the number of times that we change the output hypergraph is $\lambda = \mathcal{O}\left(\frac{nr \log(mr)}{\varepsilon}\right)$. Therefore, since we require $\log T = \mathcal{O}\left(r \log n + \log \log m\right)$ bits of precisions, our algorithm stores $\frac{n}{\varepsilon'^2} \log n \cdot \text{poly}(\log r \log \log \frac{m}{\delta'}) = \frac{n}{\varepsilon^2} \cdot \text{poly}(\log n, \log r, \log \log \frac{m}{\delta})$ hyperedges. In addition, by Theorem E.14 the robust graph sparsification algorithm for $G$ with parameters $\tilde{\varepsilon}, \tilde{\delta}$ stores $\frac{nr^4}{\varepsilon^2} \cdot \text{poly}(\log n, \log \log \frac{m}{\delta})$ edges. $\qquad \square$

## E.6 Graph Min-Cut Approximation

In this section, we apply the streaming framework to solve the graph min-cut approximation problem, which asks for a $(1 + \varepsilon)$-approximation to the size of the min-cut. First, we introduce a relaxation to the graph spectral sparsifier called the graph for-each spectral sparsifier, which is a graph $\widehat{G}$ that satisfies, for any given vector $x \in \mathbb{R}^n$, it preserves the energy of $x$ in the original graph $G$ with probability $\delta$.

**Theorem E.16** (Offline algorithm for graph for-each sparsifier). *(Ding et al., 2024) Given a graph $G = (V, E, w)$ with $n$ vertices and $m$ edges, there exists an algorithm that constructs a $(1 + \varepsilon)$-for-each spectral sparsifier with probability $1 - \frac{1}{\text{poly}(n)}$. It samples $\tilde{\mathcal{O}}\left(\frac{n}{\varepsilon}\right)$ edges and uses $\tilde{\mathcal{O}}\left(\frac{n}{\varepsilon}\right)$ words of working memory and $\tilde{\mathcal{O}}(m)$ update time.*

Ding et al. (2024) proposed an algorithm for min-cut approximation in the offline setting using the above procedure and graph for-all sparsifiers: We first utilize the graph spectral sparsifier to give a 2-approximation to all cuts. Then, let $c^*$ denote the minimum cut in the sparsifier, we select a set $S$ that contains all cuts within a factor of 4 of $c^*$, so $S$ contains the actual minimum cut. It is known that there are at most $n^{\mathcal{O}(C)}$ cuts that are within a $C$-factor to the min-cut (Karger, 2000) for any $C \geq 1$, which means that $S$ has at most $n^{\mathcal{O}(1)}$ items. Thus, we can obtain a $(1 + \varepsilon)$-graph for-each sparsifier and select the cut in $S$ that has the minimum energy on the for-each sparsifier. We can union bound over the failure events of the graph for-each sparsifier across each cut query $S$, so we obtain a $(1 + \varepsilon)$-estimation to the min-cut with high probability.

**Theorem E.17** (Offline algorithm for min-cut approximation). *(Ding et al., 2024) Given a graph $G = (V, E, w)$ with $n$ vertices and $m$ edges, there exists an algorithm that provides a $(1 + \varepsilon)$-approximation to the min-cut with probability $1 - \frac{1}{\text{poly}(n)}$. It samples $\tilde{\mathcal{O}}\left(\frac{n}{\varepsilon}\right)$ edges and uses $\tilde{\mathcal{O}}\left(\frac{n}{\varepsilon}\right)$ words of working memory and $\text{poly}(n)$ update time.*

Ding et al. (2024) applied merge-and-reduce directly to the input data stream to achieve a streaming algorithm, which loses an extra $\log n$ factor. We show that using the aforementioned framework, where we first obtain an online graph spectral sparsifier and then run the offline algorithm in each coreset, we can improve this $\log n$ factor to a $\log \frac{1}{\varepsilon}$ while maintaining a $\text{poly}(n)$ update time, achieving more efficient space.

**Theorem E.18** (Streaming algorithm for graph min-cut approximation). *Given a graph $G = (V, E, w)$ with $n$ vertices and $m$ edges defined by an insertion-only stream, there exists a streaming algorithm that provides a $(1+\varepsilon)$-approximation to the min-cut with probability $1 - \frac{1}{\text{poly}(n)}$. It stores $\frac{n}{\varepsilon} \cdot \text{polylog}(n, \frac{1}{\varepsilon})$ edges, i.e., $\frac{n}{\varepsilon} \cdot \text{polylog}(n, \frac{1}{\varepsilon})$ bits, and uses $\text{poly}(n)$ update time.*

*Proof.* We use the online algorithm in Theorem E.2 to generate the stream $\mathcal{S}'$. It produces an online for-all sparsifier, which generalizes an online cut sparsifier for the graph. We use the offline algorithm for for-each sparsifier in Theorem E.16 to construct the coresets. Then, the coreset on the roof of the binary tree is a $(1 + \varepsilon)$-for-each sparsifier by setting $\varepsilon' = \varepsilon/h$. Then, we can use the same method in Theorem E.17 to select a set $S$ that contains the min-cut and output the cut with the minimum energy on the $(1 + \varepsilon)$-for-each sparsifier. The union bound argument in Theorem E.17 gives the correctness of our streaming algorithm.

The calculation of the space complexity and the update time is similar to that of Theorem E.11. From Theorem E.16 and Theorem E.2, we have $|\mathcal{S}'| = \mathcal{O}\left(\frac{n}{\varepsilon^2} \log^2 n\right)$ and $|C| = \Theta\left(\frac{n}{\varepsilon} \text{polylog} \frac{n}{\varepsilon}\right)$, so the height of the merge-and-reduce data structure is $h = \log \frac{|\mathcal{S}'|}{|C|} = \mathcal{O}\left(\log \frac{1}{\varepsilon}\right)$. Thus, we require $\mathcal{O}\left(|C| \cdot h\right) = \frac{n}{\varepsilon} \cdot \text{polylog}(n, \frac{1}{\varepsilon})$ words of space in total. The update time of the streaming algorithm is upper bounded by the bounds in Theorem E.2 and Theorem E.16. $\square$

## F  SLIDING WINDOW MODEL

In this section, we consider the sliding window model. Although the merge-and-reduce procedure produces a coreset for an insertion-only stream in a straightforward way, it fails for the sliding window model due to the expiration of elements at the beginning of the data stream by the sliding window. Since coresets at earlier blocks of the streams are no longer valid, then the coreset at the root of the stream would no longer be accurate. To resolve this issue, Woodruff et al. (2023b) observed that we can once again partition the stream into blocks consisting of $S\left(n, \log m, \frac{\varepsilon}{2\log(mn)}, \frac{\delta}{\text{poly}(mn)}\right)$ hyperedges. However, instead of creating an offline coreset for the hyperedges in each block of updates, we create an online coreset for the elements in the reverse order of their arrival. Specifically, as the hyperedges in each block and each coreset are explicitly stored, we can create a synthetic data stream that consists of the hyperedges in the reverse order and then we can feed the synthetic stream as input to the online coreset construction. In this way, this effectively reverses the stream, so that the sliding window always corresponds to the beginning of the stream. Crucially, the online coreset construction implies correctness over any prefix of the reversed stream, which translates to correctness over any suffix of the input stream, including the sliding window. For the sake of completeness, we present this approach in Algorithm 8.

The following proof shows the correctness of the framework in Algorithm 8. It uses induction and is entirely standard, adapting the approach in (Woodruff et al., 2023b).

**Theorem F.1.** *Let $e_1, \ldots, e_m$ be a stream of hyperedges, let $\varepsilon \in (0, 1)$ be an approximation parameter, and let $H = \{e_{m-W+1}, \ldots, e_m\}$ be a hypergraph defined by the $W$ most recent hyperedges. Suppose there exists a randomized algorithm that with probability at least $1 - \delta$, outputs an online coreset algorithm for hypergraph sparsification using $S(n, \log m, \varepsilon, \delta)$ hyperedges. Then there exists a randomized algorithm that with probability at least $1 - \delta$, outputs a $(1 + \varepsilon)$-hypergraph sparsification in the sliding window model, using $\mathcal{O}\left(S\left(n, \log m, \frac{\varepsilon}{2\log(mn)}, \frac{\delta}{(mn)^2}\right) \log(mn)\right)$ hyperedges.*

*Proof.* Consider Algorithm 8, where $\text{CORESET}(H, \varepsilon, \delta)$ is a randomized algorithm that, with probability at least $1 - \delta$, computes a $(1 + \varepsilon)$-approximate online coreset for hypergraph sparsification on an input hypergraph $H$ that has $n$ nodes and $m$ hyperedges.

We first claim that for each index $i$, $C_i$ is a $\left(1 + \frac{\varepsilon}{2\log(mn)}\right)^i$ online coreset for hypergraph sparsification for $2^{i-1}M$ hyperedges. Indeed, note that $C_i$ can only be non-empty if at some time, the coreset $C_0$ contains $M$ hyperedges and the coresets $C_1, \ldots, C_{i-1}$ are all non-empty. Then by the correctness of the subroutine CORESET, $C_i$ is a $\left(1 + \frac{\varepsilon}{2\log(mn)}\right)$ online coreset for the hyper-

---

**Algorithm 8** Algorithm for hypergraph sparsification in the sliding window model via merge-and-reduce and online coresets, adapted from (Woodruff et al., 2023b)

---

1: **Require:** Hyperedges $e_1, \ldots, e_m$ on $n$ vertices, accuracy parameter $\varepsilon \in (0, 1)$, failure probability $\delta \in (0, 1)$, and window size $W > 0$
2: **Ensure:** Hypergraph sparsification of the $W$ most recent hyperedges
3: Let $\text{CORESET}(H, \varepsilon, \delta)$ be an online coreset construction that samples $S(n, \log(mn), \varepsilon, \delta))$ hyperedges, where $H$ has $n$ vertices and $m$ hyperedges
4: $M \leftarrow \mathcal{O}\left(S\left(n, \log(mn), \frac{\varepsilon}{2\log(mn)}, \frac{\delta}{(mn)^2}\right)\right)$
5: Initialize coresets $C_0, C_1, \ldots, C_{\log(mn)} \leftarrow \emptyset$
6: **for** each hyperedge $e_t$ with $t \in [m]$ **do**
7:     **if** $C_0$ does not contain $M$ hyperedges **then**
8:         Prepend $e_t$ to $C_0$, i.e., $C_0 \leftarrow \{e_t\} \cup C_0$
9:     **else**
10:         Let $i$ be the smallest index such that $C_i = \emptyset$
11:         $C_i \leftarrow \text{CORESET}\left(\widetilde{H}, \frac{\varepsilon}{2\log(mn)}, \frac{\delta}{(mn)^2}\right)$, where $\widetilde{H} = C_0 \cup \ldots \cup C_{i-1}$
12:         $\{\widetilde{H}$ is an ordered set of weighted hyperedges$\}$
13:         For $j = 0$ to $j = i - 1$, reset $C_j \leftarrow \emptyset$
14:         $C_0 \leftarrow \{e_t\}$
15:     **end if**
16:     **Return** the ordered set $C_0 \cup \ldots \cup C_{\log(mn)}$, in reverse
17: **end for**

---

edges in $C_0 \cup \ldots \cup C_{i-1}$ at some point during the stream. It follows that by induction, $C_i$ is a $\left(1 + \frac{\varepsilon}{2\log(mn)}\right)\left(1 + \frac{\varepsilon}{2\log(mn)}\right)^{i-1} = \left(1 + \frac{\varepsilon}{2\log(mn)}\right)^i$ online coreset for $M + \sum_{j=1}^{i-1} 2^{j-1}M = 2^{i-1}M$ hyperedges.

Observe that Algorithm 8 inserts the latter hyperedges to the beginning of $C_0$. Hence, the stream is fed in reverse to the merge-and-reduce procedure. In other words, for any $W \in [2^{i-1}, 2^i)$, the reverse of $C_0 \cup \ldots \cup C_i$ provides a $(1 + \varepsilon)$-hypergraph sparsifier for the $W$ most recent hyperedges in the data stream.

Moreover, there are at most $m$ hyperedges in the data stream. For each hyperedge, there are at most $\log(mn)$ coresets constructed by the subroutine CORESET, corresponding to the height of the tree. Because each subroutine has failure probability at most $\frac{\delta}{(mn)^2}$, then the total failure probability is at most $\delta$ by a union bound. This completes the argument for correctness.

It remains to justify the space complexity. To that end, observe that there are at most $\mathcal{O}(\log(mn))$ online coreset constructions $C_0, \ldots, C_{\log(mn)}$ simultaneously stored by the algorithm. Since each online coreset construction samples $S\left(n, \log m, \frac{\varepsilon}{2\log(mn)}, \frac{\delta}{(mn)^2}\right)$ hyperedges, then the total number of stored hyperedges is $\mathcal{O}\left(S\left(n, \log m, \frac{\varepsilon}{2\log(mn)}, \frac{\delta}{(mn)^2}\right)\log(mn)\right)$, as claimed. $\qquad\square$

We give the following result for hypergraph sparsification in the sliding window using the above framework.

**Theorem F.2.** *Given a hypergraph $H = (V, E)$ with $n$ vertices, $m$ hyperedges, and rank $r$ defined by an insertion-only stream, there exists an algorithm that constructs a $(1+\varepsilon)$-spectral sparsifier in the sliding window model with probability $1 - \frac{1}{\text{poly}(n)}$. It stores $\frac{n}{\varepsilon^2}\,\text{polylog}(m, r)$ hyperedges, i.e., $\frac{rn}{\varepsilon^2}\log n\,\text{polylog}(m, r)$ bits.*

# G  EXPERIMENTS

In this section, we perform a number of empirical evaluations to complement our theoretical results.[1] The experiments are conducted using an Apple M2 CPU, with 16 GB RAM and 8 cores.

**Experiment setup.** We compare the performance between three algorithms: the online algorithm, the merge-and-reduce approach applying directly to the input stream, and our streaming algorithm. In each comparison, we set a budget for the number of sampled edges and compare the multiplicative error of the three algorithms. For each experiment, we iterate $5 - 10$ times and take the arithmetic mean. We first test synthetic graphs. Given inputs $n$ and $m$, we randomly generate a graph with $n$ vertices and $m$ edges: for each edge, we uniformly sample two vertices from $[n]$ and select its weight from the distribution $\mathcal{U}(1, 10)$. We allow multi-edges in the graph. We then test the Facebook ego social network from the Stanford Large Network Dataset Collection (SNAP) (McAuley & Leskovec, 2012). We choose the graph from user 107 with $n = 1034$ and $m = 53498$, and we also select its weight from the distribution $\mathcal{U}(1, 10)$ since the original graph is unweighted.

**Graph metric.** Let $L_G$ and $\widehat{L_G}$ be the graph Laplacian of the original graph and the sparsified graph, respectively. We note that the multiplicative error of the sparsifier is $\max_{x \in \mathbb{R}^n} \frac{x^T (L_G - \widehat{L_G}) x}{x^T L_G x}$, which is the generalized Rayleigh quotient of matrices $L_G$ and $\widehat{L_G}$. To measure this quantity, we solve the generalized eigenvalue problem $(L_G - \widehat{L_G}) x = \lambda \cdot L_G x$ and obtain the maximum eigenvalue, which equals to the multiplicative error by the properties of generalized Rayleigh quotient.

**Fine-tuning the parameters.** Given a budget $l$, we fine-tune the parameters for each method such that they output roughly $l$ edges, and thus ensure that they have the same space budget. In the online algorithm, we sample each edge $e_t$ with probability $\rho \cdot r_{e_t}$, where $r_{e_t}$ is the effective resistance of $e_t$ in the graph $L_G(t)$ constructed by the previously arrived edges. Since we do not have a fixed relationship between the sum of online leverage scores and $n$, we fine-tune the parameter $\rho$ such that the mean of 10 trials is within $l \pm 200$. Note that the second approach and our streaming algorithm includes implementing offline algorithm to construct the merge-and-reduce coresets. In the offline algorithm, we sample each edge $e$ with probability $\rho \cdot r_e$, where $r_e$ is its effective resistance. Due to the equivalence between the effective resistance and the leverage score of the incidence matrix, we have $\sum_{e \in G} r_e = n$, and so the expected number of sampled edges is $\rho n$. Thus, we set $\rho = l/n$ such that we sample $l$ edges in expectation, and the actual number of samples only varies a little due to the concentration. Then, we use the same parameter $\rho$ for the offline algorithm in the second approach and our streaming algorithm to ensure fair comparison. In our streaming algorithm, recall that we run an online algorithm to obtain the prefix substream $\mathcal{S}'$ (which is not stored), we tune the length of $\mathcal{S}'$ to have the optimal error bound. We explicitly list our choice of parameters below.

We adapt the code for efficiently estimating the effective resistances in the experiments of Chen et al. (2024). In the online algorithm, we do a batch implementation to save time. We partition the stream into batches with 100 edges and use the same $L_G(t)$ to compute the effective resistances for a batch, so we do not have to re-construct the Laplacian within a batch.

**Experimental parameters.** We provide the choices of the hyperparameters of our experiments. We use $C_{\text{ol}}$, $C_{\text{of}}$, and $C_{\text{ol\_str}}$ to denote the constant factors of the online algorithm, the offline algorithm that constructs the coresets, and the prefix online substream in our streaming algorithm, respectively. The values of them are shown in Table 1.

We also include the (amortized) run-time of our algorithm in Table 2, showing that our algorithm is time-efficient in practice.

**Results and discussion.** In the first experiment, we fix the number of vertices and edges in the graph and compare the performance under different budgets $l$. For the synthetic graph, we set $n = 100, m = 50000$, and $l \in \{500, 1000, 1500, 2000, 2500, 3000\}$. For the Facebook graph, we set $l \in \{10000, 15000, 20000, 25000\}$. The results are displayed in Figure 5.

For both graphs, the merge-and-reduce methods obtain $\sim 0.3$ multiplicative error using $< 50\%$ budget, and our streaming algorithm (blue line) has the lowest multiplicative error under various budgets. This occurs because our algorithm applies the merge-and-reduce method to the online

---

[1]The code is available at https://github.com/XieShenghao16/graph_sparsification_iclr_2026.

| Synthetic Graph ($n = 100$, $m = 50000$) | | | | | | |
|---|---|---|---|---|---|---|
| Budget | 500 | 1000 | 1500 | 2000 | 2500 | 3000 |
| $C_{\text{ol}}$ | 0.001 | 0.01 | 0.15 | 0.35 | 0.55 | 0.75 |
| $C_{\text{off}}$ | 1.1 | 2.2 | 3.3 | 4.4 | 5.5 | 6.6 |
| $C_{\text{ol\_str}}$ | 2.0 | 2.5 | 5.5 | 8.0 | 11.5 | 15.0 |

| Facebook Graph ($n = 1034$, $m = 53498$) | | | | |
|---|---|---|---|---|
| Budget | 10000 | 15000 | 20000 | 25000 |
| $C_{\text{ol}}$ | 0.145 | 0.35 | 0.8 | 1.3 |
| $C_{\text{off}}$ | 1.333 | 2.333 | 3.333 | 4.5 |
| $C_{\text{ol\_str}}$ | 1.0 | 1.8 | 4 | 5.5 |

| Comparison under Different $m$, Synthetic Graph ($n = 100$, Budget=1500) | | | | | | | | | |
|---|---|---|---|---|---|---|---|---|---|
| m | 10000 | 20000 | 30000 | 40000 | 50000 | 60000 | 70000 | 80000 | 90000 | 100000 |
| $C_{\text{ol}}$ | 0.3 | 0.2 | 0.15 | 0.1 | 0.08 | 0.05 | 0.04 | 0.02 | 0.005 | 0.002 |
| $C_{\text{off}}$ | 3.3 | 3.3 | 3.3 | 3.3 | 3.3 | 3.3 | 3.3 | 3.3 | 3.3 | 3.3 |
| $C_{\text{ol\_str}}$ | 15.0 | 7.0 | 5.5 | 4.8 | 4.3 | 4.05 | 3.75 | 3.75 | 3.7 | 3.2 |

Table 1: Constant factors ($C_{\text{ol}}$, $C_{\text{off}}$, $C_{\text{ol\_str}}$) for the three experiments.

| Note | $n$ | $m$ | Budget | Batched online | Merge-and-reduce | Our algorithm |
|---|---|---|---|---|---|---|
| Synthetic graph | 50 | 2500 | 750 | $2.0 \times 10^{-4}\,s$ | $2.9 \times 10^{-4}\,s$ | $4.9 \times 10^{-4}\,s$ |
| Synthetic graph | 100 | 10000 | 2500 | $6.1 \times 10^{-4}\,s$ | $3.0 \times 10^{-4}\,s$ | $5.6 \times 10^{-4}\,s$ |
| Synthetic graph | 200 | 40000 | 10000 | $7.6 \times 10^{-4}\,s$ | $4.7 \times 10^{-4}\,s$ | $6.1 \times 10^{-4}\,s$ |
| Facebook graph | 1034 | 53498 | 15000 | $4.0 \times 10^{-2}\,s$ | $2.3 \times 10^{-2}\,s$ | $9.6 \times 10^{-2}\,s$ |

Table 2: Comparison of running times in different experiments.

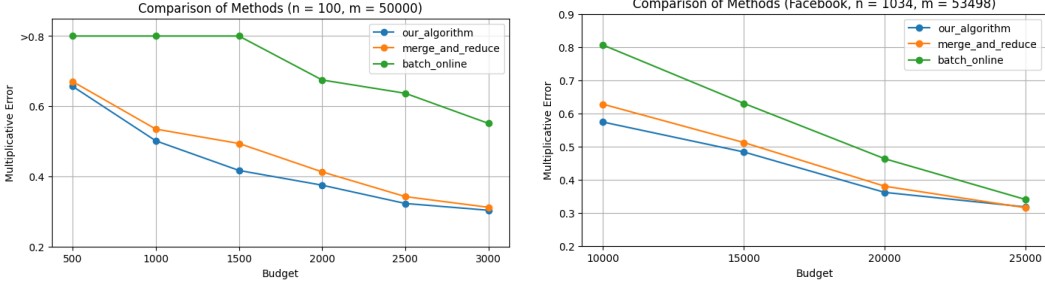

Fig. 5: Comparison under different budgets. The $x$-axis shows the space budget and the $y$-axis shows the multiplicative error. The left is the synthetic graph and the right is the Facebook graph. We include the result of the online algorithm as a baseline.

substream $\mathcal{S}'$, resulting in a shorter tree height compared to directly applying merge-and-reduce to the original stream. Since errors accumulate at each tree level, our algorithm generally outperforms others. However, its advantage diminishes as the budget increases. This is because with large budgets, we set large coreset sizes; and so the heights of the two methods are roughly the same, reducing the advantage of our algorithm. In addition, at extremely low budgets (e.g., $l = 500$ in the synthetic graph), our algorithm performs similarly to the merge-and-reduce algorithm. This implies a trade-off between the accuracy of the online substream and the tree height, when the budget is extremely low compared to $n$. If we set a small budget for $\mathcal{S}'$, the accuracy of the prefix online

algorithm is suboptimal; on the contrary, if we set a large budget for $\mathcal{S}'$, then it increases the tree height. Both situations hinder the performance of our algorithm.

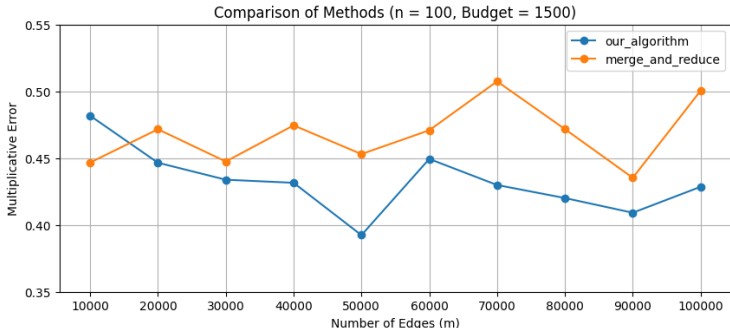

Fig. 6: Comparison under different numbers of edges

In the second experiment, we fix the budget and compare the performance under different numbers of edges $m$. We use the synthetic graph; and we set $n = 100$, $l = 1500$, and $m \in \{i \cdot 10000, i \in [10]\}$. The results are displayed in Figure 6, showing that our algorithm generally outperforms the merge-and-reduce method as $m$ increases. This advantage is because we set our budget as $l = 1500$, which is substantial compared to $n$, enabling the prefix online substream $\mathcal{S}'$ to have a high-quality approximation while maintaining a small tree height. Consequently, this enhances our algorithm's performance. In summary, our experiments demonstrate the optimality of our streaming algorithm under dense graphs and limited budgets, matching the theoretical guarantees.

