# OpenReview forum: "Nearly Space-Optimal Graph and Hypergraph Sparsification in Insertion-Only Data Streams"
_ICLR.cc/2026/Conference — ICLR 2026 Poster_

### Official Review · Reviewer_2k1R · 2025-10-29

**Soundness:** 2
**Presentation:** 2
**Contribution:** 2
**Rating:** 6
**Confidence:** 4

**Summary:**

The paper shows that in insertion-only streams one can build $(1\pm\varepsilon)$ spectral sparsifiers at essentially offline sample complexity: for graphs it stores $\tilde O(n/\varepsilon^{2})$ edges with $\mathrm{poly}(n)$ update time, and for hypergraphs it stores $O\big(n/\varepsilon^{2}\cdot \log n\cdot \mathrm{poly}(\log r,\log\log m)\big)$ hyperedges; it also gives a $(1\pm\varepsilon)$ min-cut estimator using $O\big(n/\varepsilon\cdot \mathrm{polylog}(n,1/\varepsilon)\big)$ space. The key idea is an online, for-all sparsification scheme combined with merge-and-reduce coresets, and the framework extends to sliding-window and adversarially robust settings.

**Strengths:**

The problem is important, and the paper delivers near–space-optimal $(1\pm\varepsilon)$ sparsification in insertion-only streams for graphs and hypergraphs, shaving prior $\log n$ factors and extending to min-cut, adversarially robust, and sliding-window settings—useful breadth for streaming theory and practice.

**Weaknesses:**

The problem is important, but the results feel incremental—they mainly tighten log factors and integrate known techniques, with the novelty lying more in careful refinement and synthesis than in new methods. Experiments are light: few/small datasets, no hypergraph or min-cut evaluations, no sliding-window/adversarial tests, limited comparisons, and few metrics ( no throughput or memory-per-update reporting).

**Questions:**

See weaknesses.

---

> ### Author Response · Authors · 2025-11-19
>
> Thank you for your feedback. Our replies are outlined as follows.
>
> > The problem is important, but the results feel incremental—they mainly tighten log factors and integrate known techniques, with the novelty lying more in careful refinement and synthesis than in new methods.
>
> We appreciate the reviewer’s concern regarding the scale of the improvements and would like to clarify both the significance of the results and the novelty of our techniques.
> For **graph sparsification**, while our improvement removes a $\log n$ factor, this yields a *near-optimal* online sparsifier: our sample complexity matches the offline bound up to a $\mathrm{poly}(\log\log n)$ factor.
> Achieving such a bound in the adversarial online setting is nontrivial, and our revised version highlights more clearly how our techniques accomplish this.
>
> For **hypergraph sparsification**, the gains are substantially larger.
> As noted in line 120 of the original version, prior online approaches incur additional $r$ and $\mathrm{polylog}(m)$ factors in their space bounds, which can be $\mathrm{poly}(n)$ in dense hypergraphs where $m = \Theta(2^n)$.
> Thus, improving $\log m$ factors actually translate to improving factors that are polynomial in $n$, which could otherwise be extra losses that are prohibitive in downstream applications.
> Our algorithm removes both the $r$ and $\log m$ dependencies, giving a significantly more space-efficient approach.
> We have emphasized this improvement more clearly in the revision.
>
> Regarding **technical novelty**, we summarize the key ideas here (with further detail in our response to Reviewer wsKb and in the expanded technical overview in Section 2 of the revised manuscript):
>
> - We introduce an *online* hypergraph sparsification framework that operates without prior knowledge of the incidence matrix, a central challenge in the streaming setting.
> - We design a local weight-assignment strategy that approximates the ideal sampling probabilities and integrates with online row sampling while retaining efficient memory usage.
> - We develop a decoupling technique that separates online row sampling from hyperedge sampling, ensuring independence and correctness of the sparsifier.
> - Together, these ideas yield the **first space-efficient adversarially robust hypergraph sparsification algorithm**.
>
> We hope these clarifications illustrate that the contributions go beyond tightening logarithmic factors and involve both conceptual and technical advances that may be of independent interest.
>
> > Experiments are light: few/small datasets, no hypergraph or min-cut evaluations, no sliding-window/adversarial tests, limited comparisons, and few metrics ( no throughput or memory-per-update reporting).
>
> We appreciate the reviewer’s feedback regarding the experimental scope.
> Our primary aim in this section was to provide a proof-of-concept of our main streaming framework, showing that combining an online algorithm with merge-and-reduce leads to meaningful space improvements compared to standard merge-and-reduce (Figure 1).
> To keep the focus on evaluating the framework itself, we focused our experiments to graph sparsification. We have also added runtime comparisons (Table 2), which demonstrate that our method remains efficient in practice.
>
> We agree that broader evaluations would be valuable.
> However, implementing hypergraph sparsification in a practical setting presents substantial engineering challenges: computing online sampling probabilities is computationally intensive, and there is currently no standard or efficient methodology for evaluating the multiplicative error of a hypergraph sparsifier.
> For these reasons, we view a full empirical study of hypergraph sparsification as an interesting and important direction for future work.

---

### Official Review · Reviewer_Tgfe · 2025-10-30

**Soundness:** 3
**Presentation:** 3
**Contribution:** 3
**Rating:** 8
**Confidence:** 3

**Summary:**

The paper considers the problem of graph and hypergraph sparsification in a streaming setting. The input is a (hyper)graph where the edges arrive online in a stream, and the goal of the algorithm is to maintain a spectral sparsifier of the graph with few edges. For graph sparsification, they obtain an algorithm that gives an $(1+\varepsilon)$ sparsifier with high probability in $n$ storing only $n/\varepsilon^2\cdot poly(\log\log(n))$ edges with $poly(n)$ update time (Theorem 1.1). This improves by several logarithmic factors over what one can obtain using algorithms for  offline spectral sparsification combined with the merge-and-reduce framework and is in fact optimal within $\log\log n$ factors. Additionally, they provide an algorithm for streaming min-cut which stores $n/\varepsilon\cdot poly(\log(n),1/\varepsilon)$ edges and returns a $(1+\varepsilon)$ approximation to the min-cut improving by a $\log n$ factor on previous work. I don't know how impressive this improvement is, since it's not quite clear to me on page 2, how big a polynomial dependence on $\log n$ past work (Ding et al., 2024) incurred.

Next, the paper considers algorithms for hypergraph sparsification. The number of edges needed to store in past work had factors of either $\log m$, where $m$ is the number of edges, or $r$, where $r$ is the rank of the hypergraph. Both of these are unpleasant, since $m$ could be exponential in $n$ and $r$ could be as large as $n$. In contrast, the present paper provides a streaming algorithm that only stores $n/\varepsilon^2 poly(\log n,\log r\log\log m)$, removing these unpleasant dependencies.
The paper provides a related algorithm in the online setting where edges have to be either included or not irrevocably upon arrival.

Finally, the paper provides related results in the setting where the edge arrivals are adversarial and can be based on past decisions of the algorithm, and results in the well-motivated sliding window model, where the algorithm only cares about maintaining a sparsifier of the graph defined by the edge arrivals in a past window of time in the past, and all updates that occurred before the window, should be disregarded.

The techniques seem to be quite sophisticated modifications of past work that aim to sample edges according to effective resistances in the final graph (which turns out to be the right thing to do when constructing spectral sparsifiers). I am quite unfamiliar with these techniques and got quite lost in the matrix algebra, so while the paper seems to be mathematically precisely written, I can't vouch for correctness.

**Strengths:**

Spectral sparsification of graphs and hypergraphs are important problems from both a theoretical and practical perspective and studying these problems in streaming and online settings seem quite natural. It is impressive that the paper gets improved algorithms for so many of these problems. Especially Theorem  1.1 (nearly tight bounds) and 1.3 (removing dependencies on $\log m$ and $r$) seem nice. The paper provides experiments that demonstrate that a version of their algorithm performs quite favorably in on real and synthetic data sets.

**Weaknesses:**

The paper might be difficult to read as the techniques are quite technical. It was somewhat difficult for me to assess the novelty of the ideas due to this technical opacity. It is however quite possible that people stronger in linear algebra and spectral methods find it more understandable.

Another weakness is that the polynomial update time in $n$ might be a little disappointing, but I don't think this is a big issue.

**Questions:**

What is the update time of your algorithm? I'm aware that it's $poly(n)$, but do you have a bound on the degree of the polynomial? Do previous works also have polynomial update time?

Is it always the case that you can construct a spectral sparsifier by sampling edges, or do you also need to reweight them? I was a little unsure about this, and it might be useful to the reader to discuss this.

---

> ### Author Response · Authors · 2025-11-19
>
> We appreciate your thorough review. Our responses are provided below.
>
> > The paper might be difficult to read as the techniques are quite technical. It was somewhat difficult for me to assess the novelty of the ideas due to this technical opacity. It is however quite possible that people stronger in linear algebra and spectral methods find it more understandable.
>
> We agree that the paper contains several technical components.
> To better highlight the novelty of our contributions, we have reorganized the manuscript and added a comprehensive technical overview in Section 2.
> Our key innovations include the development of an online local weight assignment strategy with efficient memory and a rigorous decoupling technique necessary to ensure the correctness of sparsification when the incidence matrix is revealed sequentially.
> We also use these online subroutines to obtain the first space-efficient, adversarially robust algorithm for hypergraph sparsification.
> (Please also see our detailed response to Reviewer wsKb regarding novelty).
>
> > Another weakness is that the polynomial update time in $n$ might be a little disappointing, but I don't think this is a big issue.
> > What is the update time of your algorithm? I'm aware that it's $\mathrm{poly}(n), but do you have a bound on the degree of the polynomial? Do previous works also have polynomial update time?
>
> Thank you for raising the question about our update time.
> The update time of our streaming algorithm is dominated by the calculation of the online sampling probability, which requires matrix-multiplication time $\tilde{O}(n^\omega)$ (omitting $\varepsilon$ and polylogarithmic factors), i.e., $\omega$ is the exponent for matrix multiplication runtime.
> While applying the standard merge-and-reduce directly offers faster $\tilde{O}(r)$ update time (where $r$ is the rank), it incurs a prohibitive $polylog(m)$ space dependency.
> Our focus is optimizing space, which is often the primary bottleneck in streaming.
> Achieving both near-optimal space and fast (near-linear) update time remains an important open question.
>
> > Is it always the case that you can construct a spectral sparsifier by sampling edges, or do you also need to reweight them? I was a little unsure about this, and it might be useful to the reader to discuss this.
>
> Yes, reweighting is essential.
> When an edge $e$ is sampled with probability $p_e$, it must be reweighted, e.g., by $1/p_e$ (as shown in Algorithm 1, line 14).
> If sampled edges are not reweighted, the resulting energy of the resulting graph is significantly smaller than the original graph unless a prohibitively large number of edges is sampled.
> Thus a valid sparsifier with a small number of edges requires some amount of reweighting.
> We have added this discussion in the updated version.

---

### Official Review · Reviewer_wsKb · 2025-11-01

**Soundness:** 2
**Presentation:** 1
**Contribution:** 2
**Rating:** 2
**Confidence:** 4

**Summary:**

This work proposes an improved algorithm for both cut and spectral sparsification of hypergraphs in the insertion-only data stream model. In particular, they remove the $\log m$ factors that can grow up to $n$ for hypergraphs of arbitrarily high rank. They achieve this by proposing an online weight assignment following the strategy of Jambulapati et al. (2023), and improve the computational cost of the sampling process by leveraging the online row sampling idea of Cohen et al. 2020. They also employ their techniques to obtain better bounds (in terms of number of edges stored, or overall space complexity) for other settings, such as online, sliding window model, and for robust hypergraph sparsification. They also demonstrate the performance of their algorithm through empirical evaluations.

**Strengths:**

The paper improves the bound for the insertion-only streaming model by a multiplicative factor that can grow up to $n$, the number of vertices.

They also propose better bounds for several other models, including resolving an open question of Soma et al. (2024) in the online hypergraph case.

**Weaknesses:**

The paper is structured poorly and difficult to read. The main results are not presented clearly. The comparison with other works (Appendix A, Fig. 3) should have been presented in the main paper.

The absence of a technical overview or a clear presentation of the main technical challenges and proposed solutions makes it difficult to comprehend the core ideas of the paper.

The comparison with Khanna et al. 2025a seems a bit unfair, given that their algorithm is for the dynamic setting, although this has been addressed in the concurrent and independent work section.

The improvements demonstrated in the experimental section, even in the limited resources setting, do not seem significant. Although this is not a major issue, given the theoretical nature of the work.

**Questions:**

1. What are the technical challenges and novel approaches involved in the work? Possibly provide a technical overview.

2. The sliding window model result (Theorem 1.6) seems to have higher space complexity than the streaming model, which is a weaker model. Am I missing something here?

Remark/Typos:

3. Fix the “additive $\log n$ factor” to “additional (multiplicative) $\log n$ factor”

4. It seems the reference in Line 119 should be Khanna et al. 2025a.

**Details Of Ethics Concerns:**

What are the technical challenges and novel approaches involved in the work? Possibly provide a technical overview.

The sliding window model result (Theorem 1.6) seems to have higher space complexity than the streaming model, which is a weaker model. Am I missing something here?

Remark/Typos:

Fix the “additive $\log n$ factor” to “additional (multiplicative) $\log n$ factor”

It seems the reference in Line 119 should be Khanna et al. 2025a.

---

> ### Author Response · Authors · 2025-11-19
>
> Thank you for the review. Our responses are as follows.
>
> > The paper is structured poorly and difficult to read. The main results are not presented clearly. The comparison with other works (Appendix A, Fig. 3) should have been presented in the main paper.
>
> We appreciate the reviewer's suggestion on our presentation.
> We have stated our main results in Theorems 1.1-1.6 in the original version, though we have reorganized the introduction in the revised version to make the structure and flow clearer.
> In particular, we moved the comparison with prior work into the main text and highlighted our improvements more prominently to improve readability.
>
> > The comparison with Khanna et al. 2025a seems a bit unfair, given that their algorithm is for the dynamic setting, although this has been addressed in the concurrent and independent work section.
>
> While Khanna et al. (2025a) address the more general dynamic setting, their work also provided the state-of-the-art bound for the insertion-only streaming model.
> Our key contribution here is achieving the first nearly-optimal space bound for streaming hypergraph sparsification, crucially removing the dependence on $\text{polylog}(m)$.
> This is a significant advancement, especially for dense hypergraphs where $m$ can be exponential in $n$.
>
> > The improvements demonstrated in the experimental section, even in the limited resources setting, do not seem significant. Although this is not a major issue, given the theoretical nature of the work.
>
> Yes, we view our central contributions as the design of new streaming algorithms for graph and hypergraph sparsification, as well as the corresponding analysis.
> The experiments serve as a proof-of-concept that complements our theoretical guarantees and empirically demonstrates the efficacy of our streaming framework over the standard merge-and-reduce approach.
> In particular, Figure 1 shows that our algorithm consistently achieves lower error for the same space usage.
>
> > What are the technical challenges and novel approaches involved in the work? Possibly provide a technical overview.
>
> Our main techniques are discussed in Sections 2 and 3 in the original version.
> We apologize for any confusion caused by the organization and presentation.
> We reorganized the original version to provide a technical overview in the revised version and summarized the technical contributions as follows:
>
> - **Novel subroutines in our online hypergraph sparsifier:**
> Prior (offline) work assigns weights $\mathbf{Z}$ to each edge in the hypergraph's associated graph to balance the effective resistance within the clique of each hyperedge, which defines their sampling probabilities.
> However, in the online setting, the hypergraph is not fully known in advance.
> Therefore, as mentioned in lines 280-284 (in the original version, same as below), the key challenge lies in defining the online sampling probabilities to satisfy the requirements of the chaining argument.
> Our novel techniques are stated in lines 285-313: we propose a local weight assignment algorithm that defines the weights $z_t$ for hyperedge $e_t$ based solely on the previously arrived rows and combine the algorithm with online row sampling to achieve efficient working memory.
> Furthermore, we decouple the construction of approximation matrix (by the online row sampling procedure) from sampling the hyperedges, ensuring that the sampling probability of $e_t$ is defined independently of whether the previous hyperedges are sampled to the sparsifier, which is essential for the correctness.
>
> - **Optimizing the working memory of our streaming algorithms:**
> We apply our online algorithms to achieve near-optimal space bounds in the streaming setting.
> By pre-filtering the input stream with an online subroutine and passing the sampled edges to the merge-and-reduce framework, we reduce the polylogarithmic factors in several critical problems while maintaining a $\mathrm{poly}(n)$ update time.
> The main technical challenge here is to optimize the memory for the sketch matrix to define the next online sampling probability, as we cannot afford to store the online coreset.
> We show that the result at the previous step from the streaming algorithm suffices to approximate the online sampling probability for the next hyperedge.
>
> - **The first adversarially robust algorithm:**
> We apply the streaming framework to provide the first space-efficient adversarially robust algorithm for hypergraph sparsification, as discussed in lines 381-390.
> We observe that the failure probability of our streaming algorithm can be limited to $\frac{\delta}{\mathrm{poly}(m)}$ with only an additional $\log \log \frac{m}{\delta}$.
> We lift it to an adversarially robust algorithm using the computational path framework by carefully controlling the flip number.

---

> > ### Comment · Reviewer_wsKb · 2025-11-27
> >
> > While the updated version with the subheadings and comparison tables improves the readability of the work compared to the earlier version, it still remains difficult to read. In particular
> >
> > 1. It is not clear which parts of the techniques are due to existing works, and where the contribution of this work lies. I feel the presentation with respect to existing techniques needs to be improved.
> >
> > 2. The proofs in the appendix section are not mapped clearly with respect to the technical overview. It would be helpful if all relevant proofs in the appendix were clearly referred to within the technical overview, as has been done in line 345 for the ‘Online sampling probability’. In this context, the appendix section names also seem to be confusing. Section C.1 is named ‘ MISSING PROOFS IN SECTION 2.1’, while some of the proofs for Section 2.1 are in Section C.2.
> >
> > 3. There seems to be a trade-off between space and time complexity of the algorithm, particularly for the hypergraph sparsification problems. When comparing with the existing works, the update time should be included to present a more complete picture of this tradeoff.
> >
> > In summary, the presentation of the technical sections still requires a lot of work. Moreover, it is difficult to validate the technical contributions and appreciate the novelties of the work, given the overall presentation. While I appreciate the results stated in the paper and technical contributions as outlined in the rebuttal, it is difficult for me to judge these aspects given the presentation of the paper.
> >
> > The improvements in the experimental section, though consistent, are incremental. This is to be expected, given that it studies standard synthetic and real-world graphs, while the main theoretical contribution of the paper is improvements for hypergraphs of high rank. Given this, I do not think the experimental section contributes much to the presentation of the work. It would be good to include the technical challenges of validating the performance on high rank hypergraphs, as outlined by the authors in response to reviewer 2k1R.
> >
> > One minor comment is that the reference to Khanna2025b in line 142 (Earlier line 119, as highlighted in the review) still remains unchanged. The same reference is given in the table in Figure 3. Is this correct?

---

> ### Author Response · Authors · 2025-11-19
>
> > The sliding window model result (Theorem 1.6) seems to have higher space complexity than the streaming model, which is a weaker model. Am I missing something here?
>
> The reviewer is correct that the space bound is higher.
> However, the sliding window model is actually more general than the standard streaming model because it must maintain an accurate approximation of the most recent $W$ items, where $W$ is the window size and can be much smaller than the stream length.
> In the standard streaming model, a newly-arrived edge with low overall contribution can be discarded.
> By comparison, in the sliding window model, the same edge may be critical to the most recent $W$ items and must therefore be retained, which leads to a higher space complexity.
> Moreover, one can set the window size $W$ as the stream length, in which case the entire stream is in the active window.
> Thus, the sliding window model can be parameterized to subsume the streaming model.
> We added this clarification in the revised version.
>
>
> > Remark/Typos:
>
> > Fix the “additive $\log n$ factor” to “additional (multiplicative) $\log n$ factor”
>
> > It seems the reference in Line 119 should be Khanna et al. 2025a.
>
> We thank the reviewer for catching the typos. We have corrected them in the revised version.

---

> ### Author Response · Authors · 2025-12-03
>
> > 2. The proofs in the appendix section are not mapped clearly with respect to the technical overview. It would be helpful if all relevant proofs in the appendix were clearly referred to within the technical overview, as has been done in line 345 for the ‘Online sampling probability’. In this context, the appendix section names also seem to be confusing. Section C.1 is named ‘ MISSING PROOFS IN SECTION 2.1’, while some of the proofs for Section 2.1 are in Section C.2.
>
> Thanks for the comment. We have moved the relevant statement (Theorem C.2) in Appendix C.1, which now proves the statements in Section 2.1, assuming Theorem C.2. We then provide the full proof of Theorem C.2 in Appendix C.2.
>
> > 3. There seems to be a trade-off between space and time complexity of the algorithm, particularly for the hypergraph sparsification problems. When comparing with the existing works, the update time should be included to present a more complete picture of this tradeoff.
>
> In the streaming setting, space is typically the primary resource of interest, with update time playing a secondary role. Our work follows this convention by prioritizing improved space bounds over previous results. While we do not attempt to optimize update time, our algorithms currently run in roughly $O(n^{2.37})$ update time. In contrast, concurrent work attains linear update time but does not achieve sublinear space. We agree that understanding whether their techniques can be adapted to reduce our update time is an interesting future direction, though it lies beyond the focus of this work.
>
> > 1. It is not clear which parts of the techniques are due to existing works, and where the contribution of this work lies. I feel the presentation with respect to existing techniques needs to be improved.
>
> > In summary, the presentation of the technical sections still requires a lot of work. Moreover, it is difficult to validate the technical contributions and appreciate the novelties of the work, given the overall presentation. While I appreciate the results stated in the paper and technical contributions as outlined in the rebuttal, it is difficult for me to judge these aspects given the presentation of the paper.
>
> Thanks for the feedback. We have heavily reworked the text in the main body so that there is an overview in each of the technical sections (Section 2 and Section 3), providing a high-level description of our approach, including the relevant previous works and our technical novelties. The additional technical sections in the appendix also include overviews that describe previous approaches, why they fail, and our contributions to overcome these barriers.
>
> > The improvements in the experimental section, though consistent, are incremental. This is to be expected, given that it studies standard synthetic and real-world graphs, while the main theoretical contribution of the paper is improvements for hypergraphs of high rank. Given this, I do not think the experimental section contributes much to the presentation of the work. It would be good to include the technical challenges of validating the performance on high rank hypergraphs, as outlined by the authors in response to reviewer 2k1R.
>
> Thanks for the suggestion, we have moved the experiments to the appendix, so that the main body can focus on the theoretical contributions of the paper. In particular, the additional room has enabled us to significantly expand on the technical novelties of our approaches. We have incorporated a discussion on previous approaches, their shortcomings, and both our algorithmic and analytic novelties to achieve our guarantees where previous techniques could not.
>
> > One minor comment is that the reference to Khanna2025b in line 142 (Earlier line 119, as highlighted in the review) still remains unchanged. The same reference is given in the table in Figure 3. Is this correct?
>
> Thanks for the pointer, we have fixed the reference in the table.

---

### Author Response · Authors · 2025-11-19

We thank the reviewers for their careful evaluation and valuable suggestions that helped improve the paper.
Below is a summary of the main changes in the revised version:

- **Moved the comparison** previously in Appendix A (Figure 3) to the main paper. We split it into three separate tables (now Figures 1, 2, and 3) for better visualization of our results (response to Reviewer wsKb).
- **Clarified the update time** of our algorithms (new text in lines 156–161, response to Reviewer Tgfe).
- **Clarified the space complexity** in the sliding-window model (new text in lines 240–246, response to Reviewer wsKb).
- **Clarified the re-weighting procedure** of hypergraph sparsification (new text in lines 314-315, response to Reviewer Tgfe).
- **Reorganized the original Sections 2 and 3 into a new “Technical Overview” section** to better highlight our technical contributions (response to Reviewers wsKb, Tgfe, and 2k1R). Specifically:
  1. Added subtitles to each paragraph for better readability.
  2. Included a short summary in front of each subsection.
  3. Rewrote the description of our online sketch maintenance (lines 358–363) to more clearly emphasize the novel techniques.
  4. Added explanation of our method to optimize the working memory of our streaming algorithms (lines 428–435).
  5. Fixed all typos and minor issues pointed out by the reviewers.

The major changes are highlighted in blue to make them easy to spot.
We believe these revisions address the reviewers’ comments and significantly improve the clarity and presentation of the paper.

---

### Author Response · Authors · 2025-12-03

We would like to thank the reviewers again for their attentive reading and constructive comments. As we wrap up the rebuttal phase, we start by underscoring the key positive aspects of the paper noted by the reviewers, such as:
- The paper improves the bound for the insertion-only streaming model by a multiplicative factor that can grow up to $n$, the number of vertices (Reviewer wsKb)
- They also propose better bounds for several other models, including resolving an open question of Soma et al. (2024) in the online hypergraph case (Reviewer wsKb)
- Spectral sparsification of graphs and hypergraphs are important problems from both a theoretical and practical perspective and studying these problems in streaming and online settings seem quite natural (Reviewer Tgfe)
- It is impressive that the paper gets improved algorithms for so many of these problems. Especially Theorem 1.1 (nearly tight bounds) and 1.3 (removing dependencies on $\log m$ and $r$) seem nice (Reviewer Tgfe)
- The paper provides experiments that demonstrate that a version of their algorithm performs quite favorably in on real and synthetic data sets (Reviewer Tgfe)
- The problem is important, and the paper delivers near–space-optimal $(1\pm\varepsilon)$ sparsification in insertion-only streams for graphs and hypergraphs, shaving prior $\log n$ factors and extending to min-cut, adversarially robust, and sliding-window settings—useful breadth for streaming theory and practice (Reviewer 2k1R)

In response to the reviewers’ comments during the discussion phase, we have made several substantive revisions to improve clarity and organization. The key updates in the revised version are:

- **Moved all experiments** to be fully contained in Appendix G (response to Reviewer wsKb)
- **Reworked Sections 2 and 3** to better highlight our main ideas (response to Reviewers wsKb, Tgfe, and 2k1R). Specifically:
  1. Added technical overview to highlight our algorithmic and analytic novelties at the start of each subsection.
  2. Added descriptive subheadings to improve readability.
   3. Moved the theorem statement and the algorithm block of our novel local weight assignment method previously in Appendix C.2 to the main paper (new text in lines 359-369 and lines 420-448).
   4. Rewrote the description of our online sketch maintenance (new text in lines 378–385) to more clearly emphasize the novel techniques.
   5. Added explanation of our method to optimize the working memory in the streaming algorithms (new text in lines 496-526).
   6. Fixed all typos and minor issues pointed out by the reviewers.

- **Moved the comparison** previously in Appendix A (Figure 3) into the main paper, splitting it into three clearer tables (now Figures 1–3) for better visualization of our results (response to Reviewer wsKb).
- **Clarified the update time** of our algorithms (new text in lines 156–161, response to Reviewer Tgfe).
- **Clarified the space complexity** in the sliding-window model (new text in lines 240–246, response to Reviewer wsKb).
- **Clarified the re-weighting procedure** in our hypergraph sparsification algorithm (new text in lines 326–328, response to Reviewer Tgfe).

All major changes are highlighted in blue for easy reference. We believe these incorporate all reviewer suggestions and address the remaining concerns, while substantially enhancing the overall presentation.

---

### Meta-Review · Area_Chair_MK6g · 2026-01-06

**Summary:**

The authors propose a space-efficient algorithm for the streaming graph sparsification problem.

The only major concern by reviewers, as far as I can see, is about presentation. While the reviewers acknowledge that the results are novel, they found it difficult to assess which part of the proof is borrowed from the existing literature and which part is novel. A reviewer suggested that  "the novelty lies more in careful refinement and synthesis than in new methods," which might explain the confusion of the other two reviewers.

Although reviewers seem not to have checked for the correctness of the dense theoretical results, they don't seem to have a reason to suspect any issues.

**Reviewer Concerns:**

There is already some back and forth with the main critical reviewer Reviewer wsKb and the authors. The reviewer acknowledged the updates but still insists that they cannot assess technical novelty. However, given their initial review, and the back-and-forth between them and the authors, the current score of 2 is not justified.

**Reviewer Scores:**

Reviewer wsKb didn't seem like they would change their score but I don't think their review justifies their very low score.

---

### Decision · Program_Chairs · 2026-01-26

Accept (Poster)